# Model-Free Robust Reinforcement Learning with Sample Complexity Analysis

**Yudan Wang**[1] **Shaofeng Zou**[1,2] **Yue Wang**[3]

[1]Electrical Engineering, University at Buffalo
[2] Computer Science & Engineering, University at Buffalo
[3]Electrical and Computer Engineering, University of Central Florida

## Abstract

Distributionally Robust Reinforcement Learning (DR-RL) aims to derive a policy optimizing the worst-case performance within a predefined uncertainty set. Despite extensive research, previous DR-RL algorithms have predominantly favored model-based approaches, with limited availability of model-free methods offering convergence guarantees or sample complexities. This paper proposes a model-free DR-RL algorithm leveraging the Multi-level Monte Carlo (MLMC) technique to close such a gap. Our innovative approach integrates a threshold mechanism that ensures finite sample requirements for algorithmic implementation, a significant improvement than previous model-free algorithms. We develop algorithms for uncertainty sets defined by total variation, Chi-square divergence, and KL divergence, and provide finite sample analyses under all three cases. Remarkably, our algorithms represent the first model-free DR-RL approach featuring finite sample complexity for total variation and Chi-square divergence uncertainty sets, while also offering an improved sample complexity and broader applicability compared to existing model-free DR-RL algorithms for the KL divergence model. The complexities of our method establish the tightest results for all three uncertainty models in model-free DR-RL, underscoring the effectiveness and efficiency of our algorithm, and highlighting its potential for practical applications.

## 1 INTRODUCTION

Reinforcement learning (RL)[Sutton and Barto, 2018] aims to find the optimal policy that maximizes cumulative rewards through interactions with the environment and has

witnessed demonstrated success in real applications, including robotics[Kober et al., 2013], finance, and computer vision. However, in more practical scenarios, direct interaction with the true environment is often unfeasible due to concerns such as safety, resource constraints, and ethical considerations. Consequently, a policy is initially learned within a simulated environment and subsequently transferred to the real environment. Yet due to reasons including unexpected external perturbations and adversarial attacks, a model mismatch between the simulation and the real environment exists, meaning the simulation may not be identical to the real environment. This model mismatch further leads to a degradation in performance when attempting to directly apply the learned policy in the real environment [Zhao et al., 2020].

One promising framework to address this issue is the DR-RL [Iyengar, 2005, Nilim and El Ghaoui, 2004]. Unlike conventional RL which optimizes performance under a specific environment, DR-RL constructs an uncertainty set of environments and aims to optimize the worst-case performance within this set. If the uncertainty set is designed to encompass the true environment, DR-RL can learn a policy robust to the model uncertainty and provide an optimized lower bound on the true performance.

Numerous algorithms have been studied and proposed for DR-RL, which can be broadly categorized into two groups: model-based methods and model-free methods. Model-based approaches, e.g., [Shi et al., 2023, Panaganti and Kalathil, 2022, Yang et al., 2022, Wang et al., 2023d], involve the collection of samples from a simulation environment to estimate an empirical robust MDP. Subsequently, robust dynamic programming [Iyengar, 2005] is employed on the empirical MDP to derive the optimal policy. In contrast, model-free methods [Wang and Zou, 2021, Liu et al., 2022, Wang et al., 2023c, Liang et al., 2023] directly learn the policy while collecting samples, bypassing the need for model estimation and storage.

While model-based methods generally require fewer sam-

*Accepted for the 40th Conference on Uncertainty in Artificial Intelligence* (UAI 2024).

ples to derive an optimal policy, storing the entire model becomes prohibitively expensive or impractical for large-scale problems. Conversely, model-free methods offer an efficient alternative that adapts without the need to store the model, facilitating more practical applications. Despite extensive research on model-based methods, the model-free DR-RL approaches remain relatively understudied. This is primarily attributed to the challenge of the distribution shift between the simulation that generates samples and the worst-case environment within the uncertainty set. The utilization of such biased samples introduces errors in each updating step and can accumulate deviations from the accurate values through the model-free bootstrapping algorithms, thus posing challenges in ensuring convergence and accurately quantifying algorithmic complexity.

To address the challenge of biased estimated updating, [Liu et al., 2022, Wang et al., 2023c] propose a Multi-level Monte Carlo (MLMC) operator, renowned for its unbiased estimation of worst-case performance, leading to asymptotically convergent model-free algorithms. However, implementing the MLMC estimator in these works necessitates an infinite number of samples. Specifically, to construct the vanilla MLMC estimator, the learner first generates a random level number $N$ following a geometric distribution and then generates $2^{N+1}$ samples. To ensure algorithm convergence, the parameter of the geometric distribution is set to be less than $\frac{1}{2}$, resulting in an infinite expected total number of samples required. Subsequently, in [Wang et al., 2023a], a modified MLMC algorithm is introduced, requiring finite samples for implementation under the KL divergence uncertainty set. Nevertheless, their findings are constrained by a restrictive assumption, limiting their applicability. In this paper, we present a novel MLMC-based DR-RL algorithm by incorporating a threshold design, referred to as the threshold-MLMC (T-MLMC) algorithm. This design ensures our implementation demands only a finite number of samples for any general uncertainty set models. Furthermore, we provide complexity analysis for our T-MLMC algorithm under three uncertainty sets without relying on any restrictive assumptions. Our contributions are outlined as follows.

## 1.1 MAJOR CONTRIBUTIONS

**We introduce a model-free T-MLMC algorithm for DR-RL with guaranteed implementation and convergence.** Unlike previous MLMC algorithms, which typically require an infinite number of samples for implementation, our approach incorporates a threshold design on the level number during the construction of our MLMC estimator. This design ensures that our estimator behaves similarly to the traditional MLMC estimator when the level number remains below the threshold. However, it adopts a simplified structure requiring fewer samples when the level number exceeds

the threshold. By implementing this threshold design, we ensure that only a finite number of samples is necessary to construct the estimator, albeit with the trade-off of introducing bias. Nevertheless, we demonstrate that our algorithm converges to a close approximation of the optimal robust value function, where the approximation error exponentially diminishes as the threshold value increases. By setting a suitable threshold value, our algorithm represents the first model-free DR-RL algorithm applicable to general uncertainty sets, providing assurances of both sample finiteness and convergence. This characteristic renders our algorithm practical for implementation and highlights its potential for diverse applications.

**We establish that our algorithm achieves the tightest sample complexity across three distinct uncertainty sets among model-free methods.** Adapting our algorithm to accommodate three uncertainty set models—defined by total variation, Chi-square divergence, and KL divergence, we ascertain their respective sample complexities. By fine-tuning the threshold, we strike a balance between bias and sample complexity, demonstrating that our algorithms effectively identify the optimal robust policy with minimal samples. Specifically, for both total variation and Chi-square divergence uncertainty sets, our algorithms achieve $\epsilon$-optimality with $\widetilde{\mathcal{O}}\left(\frac{|\mathcal{S}||\mathcal{A}|}{(1-\gamma)^5 \epsilon^2}\right)$ samples, where $|\mathcal{S}|$ and $|\mathcal{A}|$ denote the cardinality of the state and action space, respectively, and $\gamma$ represents the discount factor. For the KL divergence uncertainty set, our algorithm exhibits a sample complexity of $\widetilde{\mathcal{O}}\left(\frac{|\mathcal{S}||\mathcal{A}|}{(1-\gamma)^5 \epsilon^2 p_\wedge^2}\right)$, where $p_\wedge$ signifies the minimal non-zero entry of the nominal transition kernel. Notably, all our results boast the most stringent parameter dependencies, marking the first model-free complexity results for the total variation and Chi-square divergence models, while significantly enhancing previous findings for the KL divergence model. Furthermore, our analysis requires no restrictive assumptions, underscoring the practical applicability of our model-free algorithms. A comprehensive comparison of our results with prior ones is presented in tables Tables 1 to 3[1]. Evidently, across all three uncertainty sets, our outcomes achieve the most favorable sample complexity among model-free methods.

## 1.2 RELATED WORKS

**Model-based Methods for DR-RL** When the environment is fully known by the learner, robust dynamic programming can be applied to obtain the optimal policy [Iyengar, 2005, Nilim and El Ghaoui, 2004], which is shown to converge exponentially. When the environment is unknown, the learner can first use samples obtained to construct an empirical tran-

---

[1]Due to space limitations, we only list part of the complexity results from [Shi et al., 2023] for comparison. The complete results can be found in Table 1 therein.

| REFERENCE | MODEL-FREE | SAMPLE SIZE |
|---|---|---|
| [PANAGANTI AND KALATHIL, 2022] | × | $\widetilde{\mathcal{O}}\left(\frac{|\mathcal{S}|^2|\mathcal{A}|}{(1-\gamma)^4\epsilon^2}\right)$ |
| [YANG ET AL., 2022] | × | $\widetilde{\mathcal{O}}\left(\frac{|\mathcal{S}|^2|\mathcal{A}|}{(1-\gamma)^4\epsilon^2}\right)$ |
| [CLAVIER ET AL., 2023] | × | $\widetilde{\mathcal{O}}\left(\frac{|\mathcal{S}||\mathcal{A}|}{(1-\gamma)^4\epsilon^2}\right)$ |
| [SHI ET AL., 2023] | × | $\widetilde{\mathcal{O}}\left(\frac{|\mathcal{S}||\mathcal{A}|}{(1-\gamma)^3\epsilon^2}\right)$ |
| [WANG ET AL., 2023C] | √ | ASYMPTOTIC |
| OUR WORK | √ | $\widetilde{\mathcal{O}}\left(\frac{|\mathcal{S}||\mathcal{A}|}{(1-\gamma)^5\epsilon^2}\right)$ |

Table 1: Sample Complexity under TV Uncertainty Set

| REFERENCE | MODEL-FREE | SAMPLE SIZE |
|---|---|---|
| [PANAGANTI AND KALATHIL, 2022] | × | $\widetilde{\mathcal{O}}\left(\frac{|\mathcal{S}|^2|\mathcal{A}|}{(1-\gamma)^4\epsilon^2}\right)$ |
| [YANG ET AL., 2022] | × | $\widetilde{\mathcal{O}}\left(\frac{|\mathcal{S}|^2|\mathcal{A}|}{(1-\gamma)^4\epsilon^2}\right)$ |
| [SHI ET AL., 2023] | × | $\widetilde{\mathcal{O}}\left(\frac{|\mathcal{S}||\mathcal{A}|}{(1-\gamma)^4\epsilon^2}\right)$ |
| [WANG ET AL., 2023C] | √ | ASYMPTOTIC |
| OUR WORK | √ | $\widetilde{\mathcal{O}}\left(\frac{|\mathcal{S}||\mathcal{A}|}{(1-\gamma)^5\epsilon^2}\right)$ |

Table 2: Sample Complexity under Chi-square Uncertainty Set

| REFERENCE | MODEL-FREE | SAMPLE SIZE |
|---|---|---|
| [PANAGANTI AND KALATHIL, 2022] | × | $\widetilde{\mathcal{O}}\left(\frac{|\mathcal{S}|^2|\mathcal{A}|e^{\frac{1}{1-\gamma}}}{(1-\gamma)^4\epsilon^2}\right)$ |
| [YANG ET AL., 2022] | × | $\widetilde{\mathcal{O}}\left(\frac{|\mathcal{S}|^2|\mathcal{A}|}{(1-\gamma)^4p_\wedge^2\epsilon^2}\right)$ |
| [WANG ET AL., 2023C] | √ | ASYMPTOTIC |
| [LIANG ET AL., 2023] | √ | ASYMPTOTIC |
| [LIU ET AL., 2022] | √ | ASYMPTOTIC |
| [WANG ET AL., 2023A] | √ | $\widetilde{\mathcal{O}}\left(\frac{|\mathcal{S}||\mathcal{A}|}{p_\wedge^6(1-\gamma)^5\epsilon^2}\right)$ |
| [WANG ET AL., 2023B] | √ | $\widetilde{\mathcal{O}}\left(\frac{|\mathcal{S}||\mathcal{A}|}{p_\wedge^3(1-\gamma)^5\epsilon^2}\right)$ |
| [WANG ET AL., 2023B] (VR) | √ | $\widetilde{\mathcal{O}}\left(\frac{|\mathcal{S}||\mathcal{A}|}{p_\wedge^3(1-\gamma)^4\epsilon^2}\right)$ |
| OUR WORK | √ | $\widetilde{\mathcal{O}}\left(\frac{|\mathcal{S}||\mathcal{A}|}{p_\wedge^2(1-\gamma)^5\epsilon^2}\right)$ |

Table 3: Sample Complexity under KL Uncertainty Set. VR denotes the result obtained with variance reduce technique.

sition kernel and an empirical uncertainty set, and then apply robust dynamic programming on this empirical model, e.g., [Panaganti and Kalathil, 2022, Yang et al., 2022, Shi et al., 2023, Clavier et al., 2023, Zhou et al., 2021]. Although model-based methods generally are more data efficient, they require large memory space to store the data and model, becoming impractical for large-scale problems.

**Model-free Methods for DR-RL** Model-free methods, which learn the optimal robust policy while gathering samples, have been investigated in the context of DR-RL. In [Wang and Zou, 2021], a model-free algorithm for a contamination uncertainty set is devised, subsequently extended to other uncertainty sets in [Liu et al., 2022, Wang et al.,

2023c] through the introduction and application of a multi-level Monte Carlo (MLMC) estimator. Despite exhibiting asymptotic convergence, these algorithms necessitate an infinite number of samples to construct the MLMC estimator, thus lacking a quantified sample complexity. In [Wang et al., 2023a], it is demonstrated that a finite sample complexity for the MLMC algorithm for the KL divergence uncertainty set can be attained under a restrictive assumption, limiting the applicability of their findings. Under a similar assumption, a variance reduction-based algorithm is proposed in [Wang et al., 2023b] for the KL divergence model, and sample complexity is obtained. On the other hand, [Liang et al., 2023] introduces a stochastic approximation-based model-free algorithm, achieving asymptotic convergence without assurances on sample complexity. Despite all these works, designing a model-free DR-RL algorithm with finite sample complexity under minimal assumptions remains an open question. In this paper, we present a model-free DR-RL algorithm, providing finite sample analysis under various uncertainty set models without imposing additional assumptions.

## 2 PRELIMINARIES AND PROBLEM FORMULATIONS

### 2.1 MARKOV DECISION PROCESSES

A Markov decision processes (MDPs) is specified by $\mathcal{M} = (\mathcal{S}, \mathcal{A}, R, \gamma, \mathbf{R}_0, \mathbf{P}_0)$, where $\mathcal{S}$ and $\mathcal{A}$ denote the state and action spaces. $R \subset [0, r_{\max}]$ is a finite set of possible rewards; $\mathbf{P}_0 = \{p_{s,a} \in \Delta(\mathcal{S}) : (s,a) \in \mathcal{S} \times \mathcal{A}\}$ is the transition kernel, where $p_{s,a} \in \Delta(\mathcal{S})$. $\mathbf{R}_0 = \{\mu_{s,a} \in \Delta(R) : (s,a) \in \mathcal{S} \times \mathcal{A}\}$ is the reward distribution. At each time step, the agent starts from state $s_t$ and takes an action $a_t$. The environment transits to the next state $s_{t+1}$ according to the transition kernel $p_{s_t,a_t}$, and provides a reward signal $r(s_t, a_t) \sim \mu_{s_t,a_t}$ to the agent.

A policy $\pi : \mathcal{S} \to \Delta(\mathcal{A})$[2] denotes the probability of taking actions under different state and represents the strategy of the agent. The value function of a policy $\pi$ is defined as the expected cumulative reward the agent received by following the policy starting from $s$:

$$V_{\mathbf{P}_0,\mathbf{R}_0}^\pi(s) = \mathbb{E}\left[\sum_{t=0}^\infty \gamma^t r_t | s_0 = s, \mathbf{P}_0, \mathbf{R}_0\right].$$

The $Q$-function is defined as the cumulative reward starting from $s$ and action $a$:

$$Q_{\mathbf{P}_0,\mathbf{R}_0}^\pi(s,a) = \mathbb{E}\left[\sum_{t=0}^\infty \gamma^t r_t | s_0 = s, a_0 = a, \mathbf{P}_0, \mathbf{R}_0\right].$$

---

[2]When $\pi$ is a deterministic policy, i.e., $\pi(\cdot|s)$ is a 0-1 distribution for all $s$, we denote the deterministic action chosen at state $s$ by $\pi(s)$.

The optimal $Q$-function $Q^*$ is defined as

$$Q^*_{\mathbf{P}_0, \mathbf{R}_0}(s, a) = \max_\pi Q^\pi_{\mathbf{P}_0, \mathbf{R}_0}(s, a), \quad (1)$$

and it satisfies the Bellman equation:

$$Q^*_{\mathbf{P}_0, \mathbf{R}_0}(s, a) = \mathbb{E}\left[r_{s,a} + \gamma \max_{a' \in \mathcal{A}} Q^*_{\mathbf{P}_0, \mathbf{R}_0}(s', a')\right].$$

Moreover, the optimal policy $\pi^*_{\mathbf{P}_0, \mathbf{R}_0} = \arg\max_\pi Q^\pi_{\mathbf{P}_0, \mathbf{R}_0}$ can be obtained from the optimal $Q$-function: $\pi^*_{\mathbf{P}_0, \mathbf{R}_0}(s) = \arg\max_{a \in \mathcal{A}} Q^*_{\mathbf{P}_0, \mathbf{R}_0}(s, a)$.

## 2.2 DISTRIBUTIONALLY ROBUST MDPS

In the formulation of distributionally robust MDPs, both transition kernel and reward distribution belong to $(s, a)$-rectangular uncertainty sets $\mathcal{P}^\rho(\sigma) = \bigotimes_{s,a} \mathcal{P}^\rho_{s,a}(\sigma)$ and $\mathcal{R}^\rho(\sigma) = \bigotimes_{s,a} \mathcal{R}^\rho_{s,a}(\sigma)$. Namely, a robust MDP can be specified as $(\mathcal{S}, \mathcal{A}, R, \gamma, \mathcal{R}^\rho(\sigma), \mathcal{P}^\rho(\sigma))$, where $\mathcal{P}^\rho_{s,a}(\sigma) = \{q \in \Delta(\mathcal{S}) : \rho(q, p_{s,a}) \leq \sigma\}$, and $\mathcal{R}^\rho_{s,a}(\sigma) = \{\nu \in \Delta(R) : \rho(\nu, \mu_{s,a}) \leq \sigma\}$. Here, $\rho$ denotes any distance or divergence between two distributions, and $\sigma$ denotes the uncertainty level. The centers of the uncertainty sets, $p_{s,a}$ and $\mu_{s,a}$, are called nominal distributions.

We consider three functions that can be used to define an uncertainty set, total variation, Chi-square divergence, and KL divergence. For two distributions $p, q$, the total variation between them is defined as $\rho_{TV}(q, p) := \frac{1}{2}\|q - p\|_1$; The Chi-square divergence is defined as $\rho_{\chi^2}(q, p) = \mathbb{E}_p\left[\left(1 - \frac{q(\cdot)}{p(\cdot)}\right)^2\right]$; And the KL divergence is defined as $\rho_{KL}(q, p) = \mathbb{E}_p\left[\log \frac{q(\cdot)}{p(\cdot)}\right]$.

DR-RL aims to optimize the worst-case performance among the uncertainty sets, i.e., to optimize the robust value function:

$$\pi^{*, \rho(\sigma)} = \arg\max_\pi V^{\pi, \rho(\sigma)}$$
$$= \arg\max_\pi \inf_{q \in \mathcal{P}^\rho(\sigma), \nu \in \mathcal{R}^\rho(\sigma)} V^\pi_{q, \nu}. \quad (2)$$

It is also convenient to use notations of the robust state-action value functions:

$$Q^{\pi, \rho(\sigma)}(s, a) = \inf_{q \in \mathcal{P}^\rho(\sigma), \nu \in \mathcal{R}^\rho(\sigma)} Q^\pi_{q, \nu}(s, a), \quad (3)$$

and the optimal robust policy can also be derived from it: $\pi^{*, \rho(\sigma)} = \arg\max_\pi Q^{\pi, \rho(\sigma)}$.

The optimal robust state-action value function is hence denoted as

$$Q^{*, \rho(\sigma)}(s, a) = \sup_\pi Q^{\pi, \rho(\sigma)}(s), \quad (4)$$

and the optimal robust policy can be directly obtained from it: $\pi^{*, \rho(\sigma)}(s) = \arg\max_a Q^{*, \rho(\sigma)}(s, a)$.

It is further shown in [Iyengar, 2005] that the optimal robust $Q$-function satisfies the following robust Bellman equation:

$$Q^{*, \rho(\sigma)}(s, a) = \inf_{\nu \in \mathcal{R}^\rho(\sigma)} \mathbb{E}_\nu\left[r_{s,a}\right] \quad (5)$$
$$+ \gamma \inf_{q \in \mathcal{P}^\rho(\sigma)} \mathbb{E}_q\left[\max_{a' \in \mathcal{A}} Q^{*, \rho(\sigma)}(s', a')\right].$$

Hence DR-RL aims to find the optimal robust policy, or equivalently to solve the robust Bellman equation Equation (5).

## 2.3 STRONG DUALITY

For a general uncertainty set $\mathcal{P}$, directly computing $\inf_{p \in \mathcal{P}} p^\top V$ for any vector $V$ is computationally expensive due to the set containing an infinite number of feasible distributions. However, this optimization problem can be equivalently solved using its dual form, which is a convex optimization [Iyengar, 2005, Hu and Hong, 2013]. These results play a crucial role in our algorithm design, therefore, we introduce the dual forms corresponding to the three uncertainty sets as follows.

**Lemma 2.1** (Total variation distance)**.** *[Iyengar, 2005] The optimization problem:*

$$minimize \quad \mathbb{E}_q[v(x)]$$
$$subject \quad to \quad q \in \{\rho_{TV}(q, p) \leq \sigma, q \in \Delta(\mathcal{X})\}, \quad (6)$$

*is equivalent to*

$$\max_{u \geq 0}\left\{\mathbb{E}_p\left[v(x) - u(x)\right] - \frac{\sigma}{2} Span(v - u),\right\}. \quad (7)$$

*where $Span(X) = \max_i X(i) - \min_i X(i)$. If moreover set*

$$(v(x))_\alpha = \begin{cases} v(x) & v(x) \leq \alpha \\ \alpha & v(x) > \alpha, \end{cases}$$

*then, the optimization problem is also equivalent to*

$$\max_{\alpha \geq 0}\left\{\mathbb{E}_p\left[(v(x))_\alpha\right] - \frac{\sigma}{2}\left(\alpha - \min_x v(x)\right)\right\}. \quad (8)$$

**Lemma 2.2** (Chi-square)**.** *[Iyengar, 2005] The optimization problem:*

$$minimize \quad \mathbb{E}_q[v(x)]$$
$$subject \quad to \quad q \in \left\{\rho_{\chi^2}(q, p) \leq \sigma, q \in \Delta(\mathcal{X})\right\},$$

*is equivalent to*

$$\max_{u \geq 0}\left\{\mathbb{E}_p\left[v(x) - u(x)\right] - \sqrt{\sigma \mathbf{Var}_p\left[v(x) - u(x)\right]}\right\},$$
$$= \max_{\alpha \geq 0}\left\{\mathbb{E}_p\left[(v(x)_\alpha\right] - \sqrt{\sigma \mathbf{Var}_p\left[(v(x))_\alpha\right]}\right\}. \quad (9)$$

**Lemma 2.3** (KL divergence). *[Iyengar, 2005] The optimization problem*

$$minimize \quad \mathbb{E}_q[v(x)]$$
$$subject \quad to \quad q \in \{\rho_{KL}(q,p) \le \sigma, q \in \Delta(\mathcal{X})\},$$

*is equivalent to*

$$\max_{\alpha \ge 0} \left\{ -\alpha \log \left( \mathbb{E}_p \left[ exp \left( -\frac{v(x)}{\alpha} \right) \right] \right) - \alpha\sigma \right\}. \quad (10)$$

*Remark* 2.4. For convenience, we denote the objective functions in the dual forms by $f^{\rho(\sigma)}(p, \alpha, v)$, i.e.,

$$f^{\rho_{TV}(\sigma)}(p, \alpha, v) = \mathbb{E}_p[(v(x))_\alpha] - \frac{\sigma}{2}\left(\alpha - \min_x v(x)\right);$$

$$f^{\rho_{\chi^2}(\sigma)}(p, \alpha, v) = \mathbb{E}_p[(v(x)_\alpha] - \sqrt{\sigma \mathbf{Var}_p[(v(x))_\alpha]};$$

$$f^{\rho_{KL}(\sigma)}(p, \alpha, v) = -\alpha \log\left(\mathbb{E}_p\left[\exp\left(-\frac{v(x)}{\alpha}\right)\right]\right) - \alpha\sigma.$$

We note that these objective functions correspond to the second term of (5); For the first term, we similarly denote their dual-form objective functions by $g^{\rho(\sigma)}(\mu, \alpha)$, whose specific definition can be found in Appendix B.

## 3 MODEL-FREE THRESHOLD-MLMC ALGORITHM

In this section, we present our design of the T-MLMC algorithm. Our algorithm assumes a generative model, which can generative i.i.d. samples following the nominal kernels under arbitrary state-action pair $(s, a) \in \mathcal{S} \times \mathcal{A}$:

$$r_{s,a}^i \overset{i.i.d}{\sim} \mu_{s,a}, s_{s,a}^i \overset{i.i.d}{\sim} p_{s,a}, i = 1, ..., N. \quad (11)$$

In robust dynamic programming, one needs to update the estimation of the robust value function by applying the robust Bellman operator:

$$Q(s, a) \leftarrow \mathcal{T}^{\rho(\sigma)}(Q)(s, a)$$
$$= \inf_{\nu \in \mathcal{R}^\rho(\sigma)} \mathbb{E}_\nu[r_{s,a}] + \gamma \inf_{q \in \mathcal{P}^\rho(\sigma)} \mathbb{E}_q[Q(s', a')],$$

which is shown to converge to the optimal robust value function. In our setting, we need to estimate the two worst-case terms with the samples from the nominal distributions. However, due to the distribution shift between the nominal kernel and the worst-case kernel, estimating them is challenging. One potential approach is to first obtain an empirical nominal distribution $\hat{p}$, and construct an uncertainty set centered on it using the same function $\rho$ and uncertainty radius $\sigma$: $\hat{\mathcal{P}} = \{q : \rho(q, \hat{p}) \le \sigma\}$. However, unlike the non-robust case, where $\hat{p}^\top V$ is an unbiased estimator of the expectation $\mathbb{E}_p[V]$, the term $\min_{p \in \mathcal{P}}(p^\top V)$ is non-linear in the nominal kernel, resulting in $\min_{p \in \hat{\mathcal{P}}}(p^\top V)$ being a biased empirical estimator [Wang et al., 2023c].

To address this issue, a multi-level Monte Carlo approach is proposed in [Liu et al., 2022, Wang et al., 2023c], which is inspired by the MLMC method in statistical inference from, e.g., [Blanchet and Glynn, 2015, Blanchet et al., 2019, Wang and Wang, 2022]. Specifically, MLMC first randomly generates a level number $N$ following a geometry distribution GEO($\psi$), and then generative $2^{N+1}$ samples. Using the these samples, an estimated operator $\widetilde{\mathcal{T}}_N$ of level $N$ is further constructed, and it is shown that $\mathbb{E}_N[\widetilde{\mathcal{T}}_N(V)] = \min_{p \in \mathcal{P}}(p^\top V)$ is unbiased. Hence by replacing the robust Bellman operator with the MLMC estimator, we obtain an unbiased updating rule and the algorithm is shown to converge to the optimal policy [Liu et al., 2022, Wang et al., 2023c].

Although the MLMC algorithms are shown to asymptotically converge in these works, the parameter $\psi$ of the geometry distribution is set to be $\psi < \frac{1}{2}$, which results in an infinite expected number of samples required to construct the MLMC estimator ($\mathbb{E}_{N \sim \text{GEO}(\psi)}[2^{N+1}] = \infty$). To address this issue, we modify the MLMC by designing a threshold on the level number, to avoid numerous sample requirements when the level number is large.

Specifically, we similarly set a fixed parameter $\psi$, and sample two level numbers $N_1, N_2 \sim \text{GEO}(\psi)$. Instead of directly sampling $2^{N_i+1}$ samples, we add a threshold $N_{\max}$ when generating samples. If $N_i \le N_{\max}$, then we generate $1 + 2^{N_i+1}$ i.i.d. samples; And if $N_i > N_{\max}$, we only generate 1 samples instead. Our design ensures that the number of samples required at each time step is less than $1 + 2^{N_{\max}+1}$ and hence finite. Specifically, if $N_1 \le N_{\max}$, we independently draw $2^{N_1+1} + 1$ samples $r_{s,a,i} \sim \mu_{s,a}, i = 0, 1, ..., 2^{N_1+1}$; And when $N_1 > N_{\max}$, we draw one sample $r_{s,a,0} \sim \mu_{s,a}$. Similarly, if $N_2 \le N_{\max}$, we independently draw $2^{N_2+1} + 1$ samples $s'_{s,a,i} \sim p_{s,a}, i = 0, 1, ..., 2^{N_2+1}$. And when $N_2 > N_{\max}$, we only draw one sample $s'_{s,a,0} \sim p_{s,a}$.

We then combine this scheme with the MLMC estimator to construct our estimation of the worst-case value as follows.

For the worst-case reward term, we set

$$\widehat{r}^{\rho(\sigma)}(s, a) := r_{s,a,0} + \frac{\delta_{s,a,N_1}^{r,\rho(\sigma)}}{P_{N_1}}, \quad (12)$$

where $P_{N_1} = \psi(1-\psi)^{N_1}$ and

$$\delta_{s,a,N_1}^{r,\rho(\sigma)} := \sup_{\alpha \ge 0}\left\{g^{\rho(\sigma)}(\widehat{\mu}_{s,a,2^{N_1+1}}, \alpha)\right\}$$
$$- \frac{1}{2}\sup_{\alpha \ge 0}\left\{g^{\rho(\sigma)}(\widehat{\mu}_{s,a,2^{N_1}}^E, \alpha)\right\}$$
$$- \frac{1}{2}\sup_{\alpha \ge 0}\left\{g^{\rho(\sigma)}(\widehat{\mu}_{s,a,2^{N_1}}^O, \alpha)\right\}$$

when $N_1 \le N_{\max}$, and when $N_1 > N_{\max}, \delta_{s,a,N_1}^{r,\rho(\sigma)} = 0$. Here, $\widehat{\mu}_{s,a,2^{N_1+1}}$ denotes the empirical reward distribution obtained from the $1 + 2^{N_1+1}$ samples $\{r_{s,a,i} : i =$

$0, 1, ..., 2^{N_1+1}\}$; And denote by $\widehat{\mu}^O_{s,a,2^{N_1}}$ and $\widehat{\mu}^E_{s,a,2^{N_1}}$ the empirical reward distribution estimated from the samples with odd and even indexes.

Similarly, for the worst-case value function term, we set

$$\widehat{v}^{\rho(\sigma)}(Q(s,a)) := V(s'_{s,a,0}) + \frac{\delta^{\rho(\sigma)}_{s,a,N_2}(Q)}{P_{N_2}}, \qquad (13)$$

where $V(s) = \max_{a'} Q(s,a')$ and $P_{N_2} = \psi(1-\psi)^{N_2}$. When $N_2 > N_{\max}$, set $\delta^{\rho(\sigma)}_{s,a,N_2}(Q) = 0$; Otherwise when $N_2 \leq N_{\max}$, $\delta^{\rho(\sigma)}_{s,a,N_2}(Q)$ is defined as:

$$\begin{aligned}
\delta^{\rho(\sigma)}_{s,a,N_2}(Q) := &\sup_{\alpha \geq 0} \left\{ f^{\rho(\sigma)}(\widehat{p}_{s,a,2^{N_2+1}}, \alpha, V) \right\} \\
&- \frac{1}{2} \sup_{\alpha \geq 0} \left\{ f^{\rho(\sigma)}(\widehat{p}^E_{2^{N_2}}, \alpha, V) \right\} \\
&- \frac{1}{2} \sup_{\alpha \geq 0} \left\{ f^{\rho(\sigma)}(\widehat{p}^O_{2^{N_2}}, \alpha, V) \right\}, \quad (14)
\end{aligned}$$

where we similarly denote the empirical transition kernel obtained from all samples $\{s'_{s,a,i}, i = 0, 1, ..., 2^{N_2+1}\}$, samples with odd indexes, and samples with even indexes by $\widehat{p}_{s,a,2^{N_2+1}}$, $\widehat{p}^O_{s,a,2^{N_2}}$ and $\widehat{p}^E_{s,a,2^{N_2}}$.

Combine the two terms above together, we obtain the estimated robust Bellman operator through our T-MLMC framework:

$$\widehat{\mathcal{T}}^{\rho(\sigma)}_{N_{\max}}(Q)(s,a) = \widehat{r}^{\rho(\sigma)}(s,a) + \gamma \widehat{v}^{\rho(\sigma)}(Q(s,a)).$$

With this estimator, we propose our model-free T-MLMC algorithm as in Algorithm 1.

Note that due to the threshold $N_{\max}$, the resulting T-MLMC estimator becomes biased. However, as we will show in the next section, the bias can be bounded and inversely depends on $N_{\max}$. That is, with the increase of $N_{\max}$, the bias term tends to 0, and we recover the MLMC estimator from the T-MLMC estimator, which however will result in increasing sample complexity. We show in the following section that by carefully designing the threshold $N_{\max}$ to balance the bias-complexity trade-off, our T-MLMC algorithm converges to the optimal robust policy with finite sample complexity.

## 4 SAMPLE COMPLEXITY

In this section, we present the sample complexity results of our T-MLMC algorithms under different uncertainty sets.

As discussed above, the estimator $\widehat{\mathcal{T}}^{\rho(\sigma)}_{N_{\max}}$ we constructed is a biased estimation of the robust Bellman operator. However, in the following result, we show that the operator reduces to the vanilla MLMC estimator and the bias diminishes if $N_{\max} \to \infty$, and hence can be controlled by setting a larger threshold.

---

**Algorithm 1** T-MLMC Algorithm

---

**Input:** Parameter $\psi = \frac{1}{2}$, stepsizes $\beta_t$, iteration number $T$
**Initialize:** $\widehat{Q}^{\rho(\sigma)}_0 = 0$
**for** $t = 0$ **to** $T - 1$ **do**
  **for** every $s \in \mathcal{S}$ **do**
    Set $\widehat{V}^{\rho(\sigma)}_t(s) = \max_a \widehat{Q}^{\rho(\sigma)}_t(s,a)$
    Set $\pi_t(s) = \arg\max_a \widehat{Q}^{\rho(\sigma)}_t(s,a)$
  **end for**
  **for** every $(s,a) \in \mathcal{S} \times \mathcal{A}$ **do**
    Independently sample $N_1, N_2 \sim \text{GEO}(\psi)$
    Compute total sample sizes:
      $\mathcal{N}_1 = 1 + 2^{N_1+1}\mathbf{1}_{(N_1 \leq N_{\max})}$
      $\mathcal{N}_2 = 1 + 2^{N_2+1}\mathbf{1}_{(N_2 \leq N_{\max})}$
    Independently draw $\mathcal{N}_1$ samples $r_{s,a,i} \sim \mu_{s,a}$
    Compute $\widehat{r}^{\rho(\sigma)}(s,a)$ by Equation (12)
    Independently draw $\mathcal{N}_2$ samples $s_{s,a,i} \sim p_{s,a}$
    Compute $\widehat{v}^{\rho(\sigma)}(\widehat{Q}^{\rho(\sigma)}_t(s,a))$ by Equation (13)
    Update synchronous $Q$-table: $\widehat{Q}^{\rho(\sigma)}_{t+1}(s,a) = (1 - \beta_t)\widehat{Q}^{\rho(\sigma)}_t(s,a) + \beta_t \widehat{\mathcal{T}}^{\rho(\sigma)}_{N_{\max}}(\widehat{Q}^{\rho(\sigma)}_t)(s,a)$
  **end for**
**end for**
**Output:** $Q^{\rho(\sigma)}_T(s,a)$

---

**Theorem 4.1.** *For any fixed* $Q \in \mathbb{R}^{|\mathcal{S}||\mathcal{A}|}, s \in \mathcal{S}, a \in \mathcal{A}$, *for three uncertainty sets we considered, the estimation bias can be bounded as:*

$$\sup_{s,a} \left| \mathbb{E}\left[ \widehat{\mathcal{T}}^{\rho(\sigma)}_{N_{\max}}(Q)(s,a) \right] - \mathcal{T}^{\rho(\sigma)}(Q)(s,a) \right| \qquad (15)$$
$$\leq \widetilde{\mathcal{O}}\left( N_{\max} 2^{-\frac{N_{\max}}{2}} \right);$$

*The variance can be bounded as:*

$$Var\left( \widehat{\mathcal{T}}^{\rho(\sigma)}_{N_{\max}}(Q)(s,a) \right) \leq \widetilde{\mathcal{O}}\left( N_{\max} \right). \qquad (16)$$

The result hence suggests to set a larger value of $N_{\max}$ to diminish the bias. However, we note that the number of samples required for constructing the estimator and the overall sample complexity increase as $N_{\max} \to \infty$. To balance the trade-off between the bias and sample complexity, we choose a suitable value of $N_{\max}$ and present our complexity results in the following sections.

### 4.1 TOTAL VARIATION DISTANCE

In this part, we provide the sample complexity analysis for the total variation uncertainty set.

Utilizing results in Theorem 4.1, we obtain the following sample complexity of our T-MLMC algorithm under the TV uncertainty set.

**Theorem 4.2** (Sample Complexity with TV Distance). *Set $\psi = \frac{1}{2}$, $N_{\max} = \frac{2\log T}{\log 2}$ and set the stepsize as $\beta_t = \frac{2\log T}{(1-\gamma)T}$. Then the output from Algorithm 1 satisfies that:*

$$\mathbb{E}\left[\left\|\widehat{Q}_T^{\rho_{TV}(\sigma)} - Q^{*\rho_{TV}(\sigma)}\right\|_\infty^2\right] \leq \widetilde{\mathcal{O}}\left(\frac{1}{(1-\gamma)^5 T}\right).$$

*To obtain $\epsilon$-optimality, i.e.,*

$$\mathbb{E}\left[\left\|\widehat{Q}_T^{\rho_{TV}(\sigma)} - Q^{*,\rho_{TV}(\sigma)}\right\|_\infty^2\right] \leq \epsilon^2,$$

*the expected sample complexity $N^{\rho_{TV}(\sigma)}(\epsilon)$ is*

$$N^{\rho_{TV}(\sigma)}(\epsilon) = |\mathcal{S}||\mathcal{A}|N_{\max}T \leq \widetilde{\mathcal{O}}\left(\frac{|\mathcal{S}||\mathcal{A}|}{(1-\gamma)^5\epsilon^2}\right).$$

Our result presents the first finite sample complexity for the model-free DR-RL algorithm for the total variation uncertainty set, indicating the effectiveness and efficiency of our T-MLMC algorithm. Compared to model-based DR-RL algorithms [Yang et al., 2022, Panaganti and Kalathil, 2022, Shi et al., 2023, Clavier et al., 2023], our algorithm results in a sample complexity with a higher dependence on $(1-\gamma)$. This aligns with findings from the non-robust setting [Li et al., 2020] that the vanilla model-free algorithms(without techniques including variance reduction) generally have larger sample complexity. Our result is also expected to be improved to align with the model-based complexity through standard techniques like variance reduction.

## 4.2 CHI-SQUARE DIVERGENCE

We then present our results for DR-RL with a Chi-square divergence uncertainty set.

**Theorem 4.3** (Sample Complexity with $\chi^2$ Distance). *Set $N_{\max} = \frac{2\log T}{\log 2}$ and the stepsize as $\beta_t = \frac{2\log T}{(1-\gamma)T}$. Then the output of Algorithm 1 satisfies that:*

$$\mathbb{E}\left[\left\|\widehat{Q}_T^{\rho_{\chi^2}(\sigma)} - Q^{*\rho_{\chi^2}(\sigma)}\right\|_\infty^2\right] \leq \widetilde{\mathcal{O}}\left(\frac{1}{(1-\gamma)^5 T}\right).$$

*To ensure*

$$\mathbb{E}\left[\left\|\widehat{Q}_T^{\rho_{\chi^2}(\sigma)} - Q^{*\rho_{\chi^2}(\sigma)}\right\|_\infty^2\right] \leq \epsilon^2,$$

*the expected total sample complexity $N^{\rho_{\chi^2}(\sigma)}(\epsilon)$ is,*

$$N^{\rho_{\chi^2}(\sigma)}(\epsilon) = |\mathcal{S}||\mathcal{A}|N_{\max}T \geq \widetilde{\mathcal{O}}\left(\frac{|\mathcal{S}||\mathcal{A}|}{(1-\gamma)^5\epsilon^2}\right).$$

Our result implies that our T-MLMC algorithm is the first model-free algorithm for DR-RL under the Chi-square divergence uncertainty set. Similarly, compared to the model-based methods, our complexity presents an additional $\mathcal{O}((1-\gamma)^{-1})$-order dependence.

## 4.3 KL DIVERGENCE

We then present our results for the KL divergence uncertainty set in this section.

**Theorem 4.4** (Sample Complexity with KL Distance). *If we set $\psi = \frac{1}{2}$, threshold*

$$N_{\max} = \max\left\{\frac{2\log T}{\log 2}, \frac{\log(1 + p_\wedge^2\log(2|\mathcal{S}|)\log T)}{\log 2}\right\},$$

*and the stepsize as $\beta_t = \frac{2\log T}{(1-\gamma)T}$. Then the output of Algorithm 1 satisfies that:*

$$\mathbb{E}\left[\left\|\widehat{Q}_T^{\rho_{KL}(\sigma)} - Q^{*\rho_{KL}(\sigma)}\right\|_\infty^2\right] \leq \widetilde{\mathcal{O}}\left(\frac{1}{p_\wedge^2(1-\gamma)^5 T}\right).$$

*To ensure*

$$\mathbb{E}\left[\left\|\widehat{Q}_T^{\rho_{KL}(\sigma)} - Q^{*\rho_{KL}(\sigma)}\right\|_\infty^2\right] \leq \epsilon^2,$$

*the expected total sample complexity $N^{\rho_{KL}(\sigma)}(\epsilon)$ is*

$$N^{\rho_{KL}(\sigma)}(\epsilon) = |\mathcal{S}||\mathcal{A}|N_{\max}T \geq \widetilde{\mathcal{O}}\left(\frac{|\mathcal{S}||\mathcal{A}|}{p_\wedge^2(1-\gamma)^5\epsilon^2}\right).$$

Our result implies that our T-MLMC algorithm also solves the DR-RL problem for KL divergence uncertainty sets effectively. Compared to other model-based methods [Shi and Chi, 2022], our results are $\mathcal{O}((1-\gamma)^{-1})$-order larger. We also note that there are several previous works on the sample complexity of model-free DR-RL approaches for the KL divergence model [Wang et al., 2023b,a], and we provide a discussion on the comparison of their works with ours as follows.

In both previous works, an assumption is made regarding the size of the uncertainty level $\sigma$, specifically, $p_\wedge \geq \mathcal{O}\left(1 - e^{-\sigma}\right)$ assuming the uncertainty set cannot be too large. This assumption significantly limits the applicability of their results, as in many scenarios, the uncertainty set must be designed relatively large to encompass a broader range of environments, particularly when the nominal environment is a low-fidelity model of the true environment. In contrast, our approach does not rely on such an assumption and can be applied to any uncertainty set.

On the other hand, our sample complexity result is less than those in [Wang et al., 2023a] and the first complexity in [Wang et al., 2023b]. In [Wang et al., 2023a], the sample complexity of the vanilla MLMC DR-RL algorithm is $\widetilde{\mathcal{O}}\left(\frac{|\mathcal{S}||\mathcal{A}|}{p_\wedge^6(1-\gamma)^5\epsilon^2}\right)$. Our result improves upon this by $\mathcal{O}(p_\wedge^{-4})$, and we attribute this improvement to the designing of the threshold. In [Wang et al., 2023b], a mini-batch type model-free DR-RL algorithm is introduced, with a demonstrated sample complexity of $\widetilde{\mathcal{O}}\left(\frac{|\mathcal{S}||\mathcal{A}|}{p_\wedge^3(1-\gamma)^5\epsilon^2}\right)$. Further enhancement is achieved through the use of variance

reduction (VR) technique, bringing the complexity down to $\widetilde{\mathcal{O}}\left(\frac{|\mathcal{S}||\mathcal{A}|}{p_\wedge^3(1-\gamma)^4\epsilon^2}\right)$. Notably, our result outperforms their initial vanilla algorithm by an order of $\mathcal{O}(p_\wedge^{-1})$. While the complexity with VR technique in [Wang et al., 2023b] exhibits a superior dependence on $1-\gamma$, it fares worse concerning $p_\wedge$. This enhancement in $1-\gamma$ can be attributed to the utilization of the VR technique, consistent with previous findings [Li et al., 2020]. We hence also anticipate further improvement in our complexity results through the application of VR technique, a direction left for future investigation. Consequently, our algorithm achieves superior sample complexity compared to previous vanilla model-free algorithms and is anticipated to surpass the results in [Wang et al., 2023b] with the VR technique.

## 5 PROOF SKETCH

In this section, we briefly discuss the proof sketch for our results under the TV uncertainty set. The proofs for the other two uncertainty sets can be similarly derived. For convenience, we only discuss the proof regarding the uncertainty set of the transition kernels, the proof regarding the reward uncertainty can also be similarly obtained.

Our proof can be divided into two main parts: We first conduct a sample complexity analysis to establish the convergence of Algorithm 1 to the fixed point of the T-MLMC estimator; Then we characterize the disparity between this fixed point and the optimal robust value function. Combine the two parts together, we quantify the sample complexity of our T-MLMC algorithm converging to the near approximation of the optimal robust value function. Specifically, we decompose the error as

$$
\left\|\widehat{Q}_T^{\rho(\sigma)} - Q^{*\rho(\sigma)}\right\|_\infty^2 \le 2\left\|\widehat{Q}_T^{\rho(\sigma)} - \widehat{Q}^{*\rho(\sigma)}\right\|_\infty^2 \\
+ 2\left\|\widehat{Q}^{*\rho(\sigma)} - Q^{*\rho(\sigma)}\right\|_\infty^2, \quad (17)
$$

where $\widehat{Q}^{*\rho(\sigma)}$ denotes the fixed point of the expected T-MLMC estimator: $\bar{\mathcal{T}}_{N_{\max}}^{\rho(\sigma)}(Q) = \mathbb{E}[\widehat{\mathcal{T}}_{N_{\max}}^{\rho(\sigma)}(Q)]$. The two steps in our proof correspond to bounding the two terms in (17).

For the first term in (17), we adapt the stochastic approximation scheme. From the definition, our T-MLMC operator is an unbiased estimator of $\bar{\mathcal{T}}_{N_{\max}}^{\rho(\sigma)}(Q)$, and it suffices to show the finite variance to ensure the asymptotic convergence to the fixed point $\widehat{Q}^{*\rho(\sigma)}$. Using the concrete construction of T-MLMC operator:

$$
\widehat{v}^{\rho(\sigma)}(Q(s,a)) := V(s'_{s,a,0}) + \frac{\delta_{s,a,N_2}^{\rho(\sigma)}(Q)}{P_{N_2}} \quad (18)
$$

and definition of $\delta_{s,a,N_2}^{\rho(\sigma)}$, we directly calculate the variance of it. We show that the variance is finite and can

be bounded by $\widetilde{\mathcal{O}}(N_{\max})$, as in Theorem 4.1. Hence according to stochastic approximation theory [Borkar, 2009], Algorithm 1 converges asymptotically to the fixed point $\widehat{Q}^{*\rho(\sigma)}$. We then adapt the analysis in stochastic approximation [Chen et al., 2022] to obtain the finite-time error bound on the convergence of Algorithm 1 to $\widehat{Q}^{*\rho(\sigma)}$, i.e., the first term in (17).

For the second term in (17), we show that the approximation error between $\widehat{Q}^{*\rho(\sigma)}$ and the optimal robust value function can be bounded by considering the disparity between the robust Bellman operator and our T-MLMC operator, i.e., the bias:

$$
\left\|\widehat{Q}^{*\rho(\sigma)} - Q^{*\rho(\sigma)}\right\| \\
\le \frac{1}{1-\gamma}\left\|\bar{\mathcal{T}}_{N_{\max}}^{\rho(\sigma)}\left(Q^{*\rho(\sigma)}\right) - \mathcal{T}^{\rho(\sigma)}\left(Q^{*\rho(\sigma)}\right)\right\|. \quad (19)
$$

We note that when the threshold is not met, the T-MLMC operator is an unbiased estimator of $\mathcal{T}^{\rho(\sigma)}$, in which case we bound the error using concentration inequalities as in conventional MLMC approaches; When the threshold is met, although the error bound between the two operator is large, we can set the threshold $N_{\max}$ larger such that the probability of $\text{GEO}(\psi) > N_{\max}$ is small, resulting in a smaller error bound due to its low probability. Combining the two cases together implies a tight bound on the bias, as the first part in Theorem 4.1, and further quantifies the approximation error introduced by our T-MLMC design.

Finally, combining the two parts, we derive the sample complexity for Algorithm 1 to converge to an approximation of the optimal robust value function with a quantifying approximation error. By setting the value of the threshold, we hence obtain the final sample complexity results.

## 6 CONCLUSION

In this paper, we introduce a novel model-free T-MLMC algorithm tailored for finding the optimal robust policy in the DR-RL problem. Our algorithm strikes a delicate balance between convergence guarantees and the expected total sample size, ensuring convergence within a finite sample size. We further conduct sample complexity analyses for our algorithm under three distinct uncertainty sets: total variation, Chi-square divergence, and KL divergence. Notably, our results mark the first complexity analyses for model-free DR-RL methods under the total variation and Chi-square divergence uncertainty sets, while also enhancing the complexity bounds and applicability of prior results for the KL divergence model. Our results achieve the tightest complexity bounds in the realm of model-free DR-RL methods, achieving state-of-the-art results under minimal assumptions.

## ACKNOWLEDGEMENT

The work of Yudan Wang and Shaofeng Zou is supported by the National Science Foundation under Grants CCF-2007783, CCF-2106560 and ECCS-2337375 (CAREER). Yue Wang is supported by UCF start-up funding.

This material is based upon work supported under the AI Research Institutes program by National Science Foundation and the Institute of Education Sciences, U.S. Department of Education through Award # 2229873 - National AI Institute for Exceptional Education. Any opinions, findings and conclusions or recommendations expressed in this material are those of the author(s) and do not necessarily reflect the views of the National Science Foundation, the Institute of Education Sciences, or the U.S. Department of Education.

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

# A NUMERICAL RESULT

In this section, we conduct numerical experiments to validate the convergence of our T-MLMC algorithm.

## A.1 CONVERGENCE AND OPTIMIALITY OF T-MLMC ALGORITHM

We adapt our algorithm under the Garnet problem $\mathcal{G}(15, 20)$ [Archibald et al., 1995]. There are 15 states and 20 actions. The transition kernel $\mathbf{P} = \{\mathbf{P}_s^a\}$ is randomly generated by a normal distribution: $\mathbf{P}_s^a \sim \mathcal{N}(\omega_s^a, \sigma_s^a)$ and then normalized, and the reward function $r(s, a) \sim \mathcal{N}(\nu_s^a, \psi_s^a)$, where $\omega_s^a, \sigma_s^a, \nu_s^a, \psi_s^a \sim \mathbf{Uniform}[0, 100]$. We implement our T-MLMC algorithm under three distinct uncertainty sets. In our experiment, the uncertainty level for each model are set to be 0.4, the step size are set to be $\beta = 0.01$, and $N_{\max} = 32$.

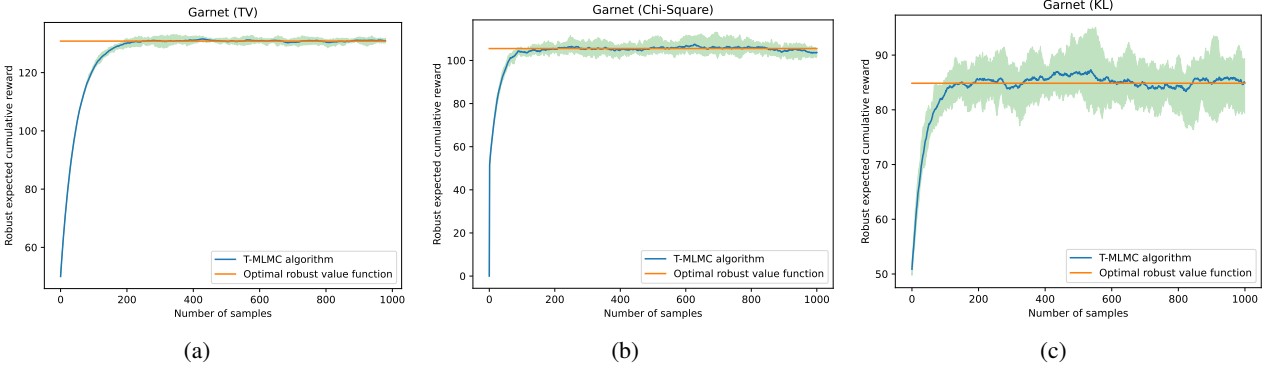

Figure 1: Garnet $\mathcal{G}(20, 15)$ (a)TV (b) $\chi^2$ (c) KL uncertainty set

We run algorithm under each uncertainty set for 20 times, and at each time step, we evaluate the worst-case performance of the greedy policy derived from the algorithm. We plot the average robust value function across the 20 runs, along with the 5th and 95th percentiles of the 20 runs as an envelope of variability. To establish a baseline, we compute the optimal robust value functions using robust dynamic programming.

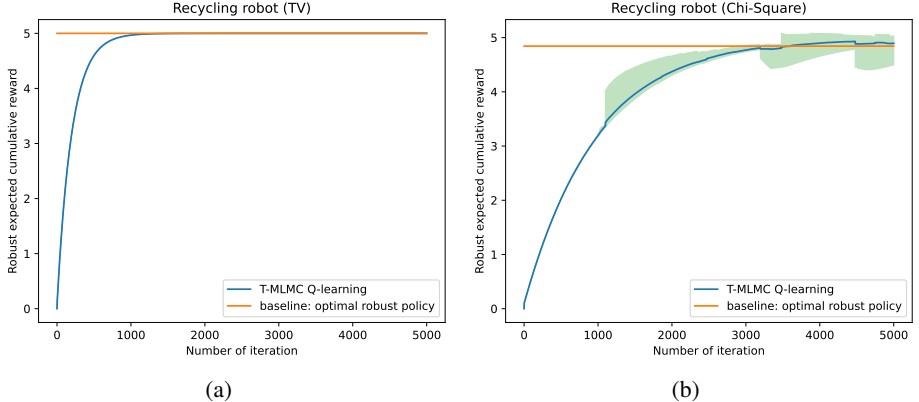

Figure 2: Recycling Robot (a)TV (b) $\chi^2$ uncertainty set

We further provide an experiment on a real-life problem: recycling robot problem (Example 3.3 [Sutton and Barto, 2018, Wang et al., 2023c]. A mobile robot running on a rechargeable battery aims to collect empty soda cans. It has 2 battery levels: low and high. The robot can either 1) search for empty cans; 2) remain stationary and wait for someone to bring it a can; 3) go back to its home base to recharge. Under low (high) battery level, the robot finds an empty can with probabilities $\alpha(\beta)$, and remains at the same battery level. If the robot goes out to search but finds nothing, it will run out of its battery and can only be carried back by human. We introduce model uncertainty to the probabilities $\alpha, \beta$ of finding an empty can if the robot chooses the action 'search'. We implement our algorithm under this problem. In our experiment, the uncertainty level for each model is set to be 0.2, the recycling system are set to be $\alpha = 0.5, \beta = 0.5$, the learning rate is set to be 0.01, and $N_{\max} = 32$. We run algorithm under each uncertainty set for 20 times, and at each time step, we evaluate the worst-case performance of the greedy policy derived from the algorithm. We plot the average robust value function across the 20 runs, along with the 5th and 95th percentiles of the 20 runs as an envelope of variability. To establish a baseline, we compute the optimal robust value functions using robust dynamic programming.

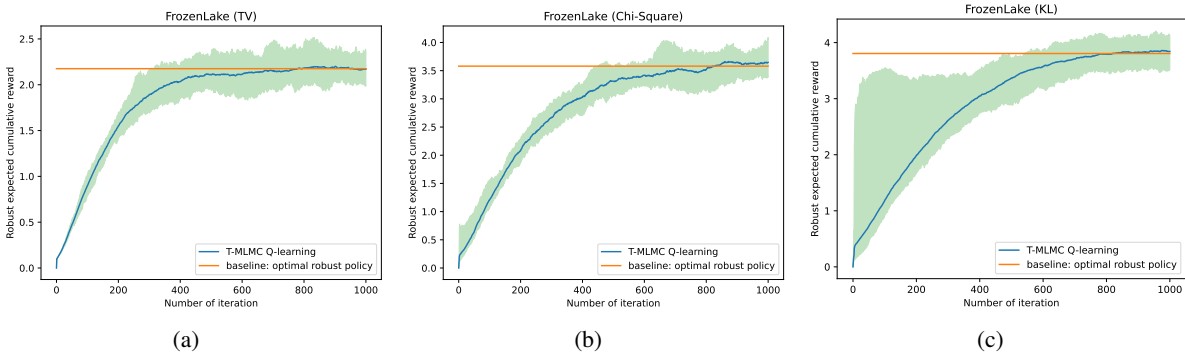

Figure 3: FrozenLake (a)TV (b) $\chi^2$ (c) KL uncertainty set

We further explore the theoretical FrozenLake environment. This simulation involves navigating from the starting point at [0,0] to the goal at [3,3] on a 4x4 grid of icy patches and holes. Players choose to move up, down, left, or right, but due to the ice's slipperiness, movement may not always follow the intended direction. We incorporate model uncertainty into movement probabilities to account for the ice's unpredictable nature, demanding strategic planning and robust algorithm implementation to safely reach the goal. In our experiment, the uncertainty level for each model is set to be 0.2, the learning rate is set to be 0.01, and $N_{\max} = 32$. We run algorithm under each uncertainty set for 20 times, and at each time step, we evaluate the worst-case performance of the greedy policy derived from the algorithm. We plot the average robust value function across the 20 runs, along with the 5th and 95th percentiles of the 20 runs as an envelope of variability. To establish a baseline, we compute the optimal robust value functions using robust dynamic programming.

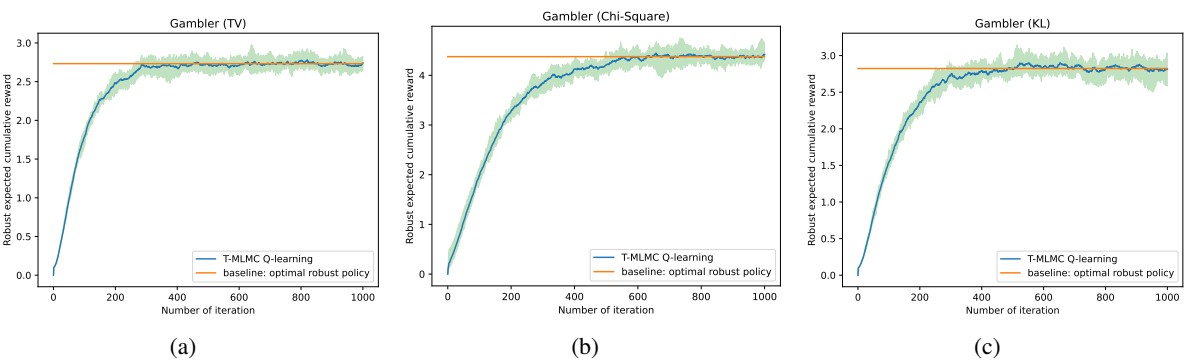

Figure 4: Gambler (a)TV (b) $\chi^2$ (c) KL uncertainty set

Besides, we validate our algorithm in the Gambler's Problem, featured [Zhou et al., 2021, Shi and Chi, 2022]. In this scenario, a gambler starts with an initial stake and bets on coin toss outcomes to reach a financial goal, such as turning 1 into 100. Each bet can lead to a gain or loss, dictated by probabilities $p$ and $1 - p$, respectively. The gambler's challenge is to devise a strategy that maximizes the odds of reaching the target without going bankrupt, considering they can bet any amount up to the lesser of their current capital or the amount needed to reach the goal. This problem emphasizes the development of optimal betting policies and the application of value iteration techniques to achieve desired outcomes in a risk-laden environment. In our experiment, the uncertainty level for each model is set to be 0.2, the parameter $p$ in system is set as $p = 0.6$, the learning rate is set to be 0.01, and $N_{\max} = 32$. We run algorithm under each uncertainty set for 20 times, and at each time step, we evaluate the worst-case performance of the greedy policy derived from the algorithm. We plot the average robust value function across the 20 runs, along with the 5th and 95th percentiles of the 20 runs as an envelope of variability. To establish a baseline, we compute the optimal robust value functions using robust dynamic programming.

As the results show, our algorithm converges to the optimal robust value functions under all uncertainty sets, indicating the algorithm's capacity to derive the optimal robust policy effectively. The experimental findings thus corroborate our theoretical assertions, affirming the convergence of our model-free T-MLMC algorithm.

## A.2 COMPARISON WITH VANILLA MLMC ALGORITHM

Furthermore, we compare our T-MLMC algorithm with the vanilla MLMC algorithm using the recycling robot problem as a test case. We run both algorithms with same parameters, and plot the robust value functions of the learned policy v.s. the number of samples. Our algorithm learned the optimal policy with a much fewer number of samples, demonstrating that our T-MLMC algorithm exhibits better sample complexity performance than the vanilla MLMC algorithm.

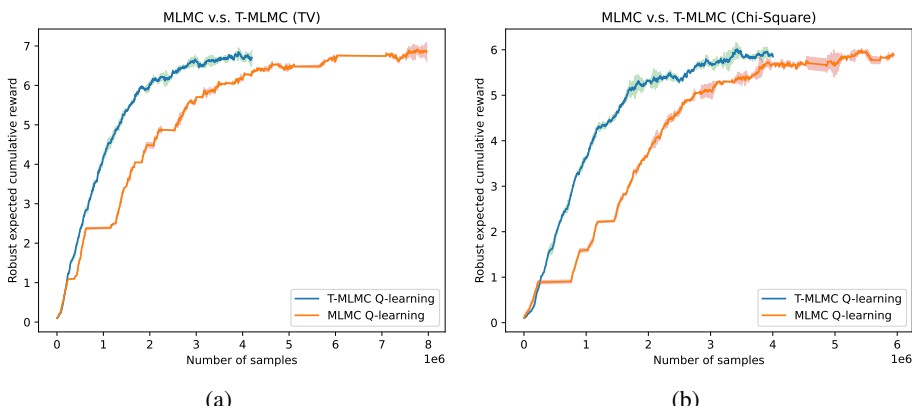

(a)                          (b)

Figure 5: T-MLMC v.s. MLMC (a)TV (b) $\chi^2$ uncertainty set

# B NOTATIONS AND LEMMAS

In this section, we present the necessary notations and lemmas which are later used in the proofs. The proofs of these lemmas can be found in Appendix F.

Recall that for the reward uncertainty set, we have

$$g^{\rho_{TV}(\sigma)}(\mu, \alpha) = \mathbb{E}_\mu[(x)_\alpha] - \frac{\sigma}{2}(\alpha - \min x);$$

$$g^{\rho_{\chi^2}(\sigma)}(\mu, \alpha) = \mathbb{E}_\mu[(x)_\alpha] - \sqrt{\mathbf{Var}_\mu[(x)_\alpha]};$$

$$g^{\rho_{KL}(\sigma)}(\mu, \alpha) = -\alpha \log\left(\mathbb{E}_p\left[\exp\left(-\frac{x}{\alpha}\right)\right]\right) - \alpha\sigma.$$

Specifically, the definition is as follows

$$g^{\rho_{TV}(\sigma)}(\mu_{s,a}, \alpha, r_{s,a}) = \mathbb{E}_{\mu_{s,a}}[(r_{s,a})_\alpha] - \frac{\sigma}{2}(\alpha - \min r_{s,a});$$

$$g^{\rho_{\chi^2}(\sigma)}(\mu_{s,a}, \alpha, r_{s,a}) = \mathbb{E}_{\mu_{s,a}}[(r_{s,a})_\alpha] - \sqrt{\sigma \mathbf{Var}_{\mu_{s,a}}[(r_{s,a})_\alpha]};$$

$$g^{\rho_{KL}(\sigma)}(\mu_{s,a}, \alpha, r_{s,a}) = -\alpha \log\left(\mathbb{E}_{\mu_{s,a}}\left[\exp\left(-\frac{r_{s,a}}{\alpha}\right)\right]\right) - \alpha\sigma. \tag{20}$$

For the transition kernel uncertainty set, the detailed definition is as follows

$$f^{\rho_{TV}(\sigma)}(p_{s,a}, \alpha, V) = \mathbb{E}_{p_{s,a}}\left[(V(s'_{s,a}))_\alpha\right] - \frac{\sigma}{2}\left(\alpha - \min_{s'_{s,a}} V(s'_{s,a})\right);$$

$$f^{\rho_{\chi^2}(\sigma)}(p_{s,a}, \alpha, V) = \mathbb{E}_{p_{s,a}}\left[(V(s'_{s,a}))_\alpha\right] - \sqrt{\sigma \mathbf{Var}_{p_{s,a}}\left[(V(s'_{s,a}))_\alpha\right]};$$

$$f^{\rho_{KL}(\sigma)}(p_{s,a}, \alpha, V) = -\alpha \log\left(\mathbb{E}_{p_{s,a}}\left[\exp\left(-\frac{V(s'_{s,a})}{\alpha}\right)\right]\right) - \alpha\sigma. \tag{21}$$

We present the analysis of propositions and theorems proof of the T-MLMC algorithm. To simplify the proof process, we just provide the analysis of the transition kernel uncertainty set, which is easy to extend to the reward uncertainty set.

Firstly, to define the surrogate $Q$-table $\widehat{Q}^{*\rho(\sigma)}$, we define the expected biased estimation of dual value as follows:

**Definition B.1** (Biased estimation). Draw $n$ samples from nominal distribution $s'_{s,a,i} \sim p_{s,a}, i = 0, 1, .., n-1$ and get the empirical distributio $\widehat{p}_{s,a,n}$. We define (resp. $\mu_{s,a}, \hat{\mu}_{s,a,n}$)

$$f^{*\rho(\sigma)}(\hat{p}_{s,a,n}, V) := \sup_{\alpha \geq 0} \left\{ f^{\rho(\sigma)}(\hat{p}_{s,a,n}, \alpha, V) \right\}.$$

The estimation of the robust Bellman operator is biased and the bias depends on empirical distribution sample sizes $n$, which referred to as

$$\left| \mathbb{E}\left[ f^{*\rho(\sigma)}(\hat{p}_{s,a,n}, V) \right] - f^{*\rho(\sigma)}(p_{s,a}, V) \right|.$$

We first show that when including the threshold $N_{\max}$ in our algorithm, the bias of the robust Bellman operator is equal to the bias when applying the model-based algorithm with sample size $2^{N_{\max}+1}$. Here, we describe the condition by the following proposition.

**Proposition B.2** (Threshold MLMC). *We recall that $\widehat{\mathcal{T}}_{N_{\max}}^{\rho(\sigma)}(Q)(s,a) = \widehat{r}^{\rho(\sigma)}(s,a) + \gamma \widehat{v}^{\rho(\sigma)}(Q(s,a))$. The robust Bellman estimator $\widehat{v}^{\rho(\sigma)}(Q(s,a))$ (resp. $\widehat{r}^{\rho(\sigma)}(s,a)$) satisfies that*

$$\mathbb{E}\left[ \widehat{r}^{\rho(\sigma)}(s,a) \right] = \mathbb{E}\left[ r_{s,a,0} + \frac{\delta_{s,a,N_1}^{\rho(\sigma),r}}{P_{N_1}} \right] = \mathbb{E}\left[ g^{*\rho(\sigma)}(\hat{\mu}_{s,a,2^{N_{\max}+1}}, r_{s,a}) \right],$$

$$\mathbb{E}\left[ \widehat{v}^{\rho(\sigma)}(Q(s,a)) \right] = \mathbb{E}\left[ V(s'_{s,a,0}) + \frac{\delta_{s,a,N_2}^{\rho(\sigma)}(Q)}{P_{N_2}} \right] = \mathbb{E}\left[ f^{*\rho(\sigma)}(\hat{p}_{s,a,2^{N_{\max}+1}}, V) \right].$$

*Thus, the estimated robust Bellman operator $\widehat{\mathcal{T}}_{N_{\max}}^{\rho(\sigma)}$ satisfies*

$$\begin{aligned}
\mathbb{E}\left[ \widehat{\mathcal{T}}_{N_{\max}}^{\rho(\sigma)}(Q)(s,a) \right] &= \mathbb{E}\left[ \widehat{r}^{\rho(\sigma)}(s,a) + \gamma \widehat{v}^{\rho(\sigma)}(Q(s,a)) \right] \\
&= \mathbb{E}\left[ g^{*\rho(\sigma)}(\hat{\mu}_{s,a,2^{N_{\max}+1}}, r_{s,a}) + \gamma f^{*\rho(\sigma)}(\hat{p}_{s,a,2^{N_{\max}+1}}, V) \right].
\end{aligned} \tag{22}$$

The Proposition B.2 shows the fact that the estimation biases are equal when drawing $2^{N_{\max}+1}$ samples to estimate the dual value directly and when setting the $N_{\max}$-threshold MLMC algorithm to estimate the dual value.

Based on Proposition B.2, for $\rho$ distance and uncertainty level $\sigma$, we define

$$\mathbb{E}\left[ \widehat{\mathcal{T}}_{N_{\max}}^{\rho(\sigma)}(Q)(s,a) \right] = \bar{\mathcal{T}}_{N_{\max}}^{\rho(\sigma)}(Q)(s,a), \tag{23}$$

where $\bar{\mathcal{T}}_{N_{\max}}^{\rho(\sigma)}$ is the surrogate robust operator being the expectation of our T-MLMC estimator.

**Proposition B.3.** *Given statistical distance $\rho$ and uncertainty level $\sigma$, estimated robust Bellman operator $\widehat{\mathcal{T}}^{\rho(\sigma)}$ is $\gamma$-contraction w.r.t. the infinity norm:*

$$\left\| \bar{\mathcal{T}}_{N_{\max}}^{\rho(\sigma)}(Q) - \bar{\mathcal{T}}_{N_{\max}}^{\rho(\sigma)}(Q') \right\|_{\infty} \leq \gamma \left\| Q - Q' \right\|_{\infty}. \tag{24}$$

We denote its unique fixed point as $\widehat{Q}^{*\rho(\sigma)}$, i.e.

$$\bar{\mathcal{T}}_{N_{\max}}^{\rho(\sigma)}\left( \widehat{Q}^{*\rho(\sigma)} \right) = \widehat{Q}^{*\rho(\sigma)}. \tag{25}$$

It hence holds that

$$\left\| \widehat{Q}_T^{\rho(\sigma)} - Q^{*\rho(\sigma)} \right\|_{\infty}^2 \leq 2\left\| \widehat{Q}_T^{\rho(\sigma)} - \widehat{Q}^{*\rho(\sigma)} \right\|_{\infty}^2 + 2\left\| \widehat{Q}^{*\rho(\sigma)} - Q^{*\rho(\sigma)} \right\|_{\infty}^2. \tag{26}$$

**Lemma B.4.** *The optimal robust Q-function and estimated optimal robust Q-function can be bounded as follows:*

$$\left\| \widehat{Q}^{*\rho(\sigma)} - Q^{*\rho(\sigma)} \right\|_\infty \leq \frac{1}{1-\gamma} \left\| \bar{\boldsymbol{\mathcal{T}}}_{N_{\max}}^{\rho(\sigma)} \left( Q^{*\rho(\sigma)} \right) - \mathcal{T}^{\rho(\sigma)} \left( Q^{*\rho(\sigma)} \right) \right\|_\infty. \tag{27}$$

Combined with Proposition B.2 and Lemma B.4, this term can be bounded specifically for different uncertainty set (TV, $\chi^2$ and KL) in the following sections.

We then aim to bound the first term in Equation (26) $\left\| \widehat{Q}_T^{\rho(\sigma)} - \widehat{Q}^{*\rho(\sigma)} \right\|_\infty$. The error between surrogate $Q$-table and the optimal robust $Q$-table, $\left\| Q^{*\rho(\sigma)} - \widehat{Q}^{*\rho(\sigma)} \right\|_\infty^2$ can be bounded following lemma.

**Lemma B.5** ([Chen et al., 2022] Theorem 2.1 & Corollary 2.1.2)**.** *For the following stochastic iteration,*

$$\theta_{k+1} = \theta_k + \beta_k \left( \mathcal{H}(\theta_k) - \theta_k + w_k \right), \tag{28}$$

*where $\theta \in \mathbb{R}^d$, $\beta_k$ is the stepsize. The fixed point $\theta^*$ satisfies that $\theta^* = \mathcal{H}(\theta^*)$. Define $\mathcal{F}_k = \{\theta_0, w_0, ..., \theta_{k-1}, w_{k-1}, \theta_k\}$. When*

$$\|\mathcal{H}(\theta) - \mathcal{H}(\theta')\|_\infty \leq \gamma \|\theta - \theta'\|_\infty, \tag{29}$$

*and*

$$(a).\mathbb{E}\left[w_k | \mathcal{F}_k\right] = 0; \qquad (b).\mathbb{E}\left[\|w_k\|_\infty^2 | \mathcal{F}_k\right] \leq A + B \|\theta_k\|_\infty^2, \tag{30}$$

*when $\beta_t \leq \frac{c_2}{c_3}$, we have*

$$\mathbb{E}\left[\|\theta_k - \theta^*\|_\infty^2\right]$$
$$\leq c_1 \|\theta_0 - \theta^*\|_\infty^2 \prod_{j=0}^{k-1}(1 - c_2\beta_j) + c_4 \left(A + 2B \|\theta^*\|_\infty^2\right) \sum_{i=0}^{k-1} \beta_i^2 \prod_{j=i+1}^{k-1}(1 - c_2\beta_j), \tag{31}$$

*where $c_1 = \frac{3}{2}$, $c_2 = \frac{1-\gamma}{2}$, $c_3 = \frac{32e(B+2)\log(d)}{1-\gamma}$, $c_4 = \frac{16e\log(d)}{1-\gamma}$.*

We consider the stochastic iteration that

$$\widehat{Q}_{t+1}^{\rho(\sigma)} = \widehat{Q}_t^{\rho(\sigma)} + \beta_t \left( \bar{\boldsymbol{\mathcal{T}}}_{N_{\max}}^{\rho(\sigma)} \left( Q_t^{\rho(\sigma)} \right) - \widehat{Q}_t^{\rho(\sigma)} + W_t \right), \tag{32}$$

where we define the filtration $\mathcal{F}_t = \left\{ Q_0^{\rho(\sigma)}, W_0, ..., Q_{t-1}^{\rho(\sigma)}, W_{t-1}, Q_t^{\rho(\sigma)} \right\}$. There are three requirements when applying the Lemma B.5:

**1)** $\bar{\boldsymbol{\mathcal{T}}}_{N_{\max}}^{\rho(\sigma)} = \mathbb{E}\left[ \widehat{T}_{N_{\max}}^{\rho(\sigma)}(Q) \right]$ **is $\gamma$ contraction operator (Proposition B.3).**

**2). Unbiased estimation:**

$$\mathbb{E}\left[ \widehat{\mathcal{T}}_{N_{\max}}^{\rho(\sigma)} \left( Q^{\rho_{TV}(\sigma)} \right) - \bar{\boldsymbol{\mathcal{T}}}_{N_{\max}}^{\rho_{TV}(\sigma)} \left( Q^{\rho_{TV}(\sigma)} \right) \Big| \mathcal{F}_t \right] = 0 \tag{33}$$

**3). Bounded infinite norm expectation:** Here, the boundary of the MLMC estimator infinite norm expectation $\mathbb{E}\left[ \left\| \widehat{\mathcal{T}}_{N_{\max}}^{\rho(\sigma)}(Q) \right\|_\infty^2 \right]$ is required to make sure the convergence of the algorithm. Take the expectation of $N \sim \text{Geo}(\psi)$, the expectation of infinite norm can be bounded by

$$\mathbb{E}\left[ \left\| \widehat{\mathcal{T}}_{N_{\max}}^{\rho(\sigma)}(Q) \right\|_\infty^2 \right] \overset{(a)}{\leq} 4r_{\max}^2 + 4\gamma^2 \frac{r_{\max}^2}{(1-\gamma)^2} + 4\mathbb{E}\left[ \sum_{N_1=0}^{N_{\max}} \sup_{s,a} \frac{\left(\delta_{s,a,N_1}^{r,\rho(\sigma)}\right)^2}{P_{N_1}} + \gamma^2 \sum_{N_2=0}^{N_{\max}} \sup_{s,a} \frac{\left(\delta_{s,a,N_2}^{\rho(\sigma)}(Q)\right)^2}{P_{N_2}} \right]$$

$$= 4r_{\max}^2 + 4\gamma^2 \frac{r_{\max}^2}{(1-\gamma)^2} + 4\mathbb{E}\left[ \sum_{N_1=0}^{N_{\max}} \sup_{s,a} \frac{\left(\delta_{s,a,N_1}^{r,\rho(\sigma)}\right)^2}{\psi(1-\psi)^{N_1}} + \gamma^2 \sum_{N_2=0}^{N_{\max}} \sup_{s,a} \frac{\left(\delta_{s,a,N_2}^{\rho(\sigma)}(Q)\right)^2}{\psi(1-\psi)^{N_2}} \right], \tag{34}$$

where $(a)$ follows from that $\sup_{s,a} r_{s,a} \le r_{\max}$, $\sup_{s,a} Q(s,a) \le \frac{r_{\max}}{1-\gamma}$ and the two estimators are independent.

Then, make the decomposition of the term $\sup_{s,a} \left| \delta^{\rho(\sigma)}_{s,a,N_2}(Q) \right|^2$,

$$
\sup_{s,a} \left| \delta^{\rho(\sigma)}_{s,a,N}(Q)(s,a) \right|^2 \le 3 \sup_{s,a} \left| \sup_{\alpha \ge 0} \left\{ f^{\rho(\sigma)}(\widehat{p}_{s,a,2^{N+1}}, \alpha, V) \right\} - \sup_{\alpha \ge 0} \left\{ f^{\rho(\sigma)}(p_{s,a}, \alpha, V) \right\} \right|^2
$$
$$
+ \frac{3}{4} \sup_{s,a} \left| \sup_{\alpha \ge 0} \left\{ f^{\rho(\sigma)}(\widehat{p}^{E}_{s,a,2^{N}}, \alpha, V) \right\} - \sup_{\alpha \ge 0} \left\{ f^{\rho(\sigma)}(p_{s,a}, \alpha, V) \right\} \right|^2
$$
$$
+ \frac{3}{4} \sup_{s,a} \left| \sup_{\alpha \ge 0} \left\{ f^{\rho(\sigma)}(\widehat{p}^{O}_{s,a,2^{N}}, \alpha, V) \right\} - \sup_{\alpha \ge 0} \left\{ f^{\rho(\sigma)}(p_{s,a}, \alpha, V) \right\} \right|^2. \tag{35}
$$

Then the terms in Equation (35) can be bounded specifically for different uncertainty sets (TV, $\chi^2$, and KL) and we can obtain the sample complexity.

## C TOTAL VARIATION UNCERTAINTY SET

In this part, we present the proof of propositions and theorems specifically for the TV-constrained uncertainty set.

**Theorem C.1** (Restatement of Theorem 4.1 specifically for TV distance)**.** *Consider the case of TV constraint uncertainty set with uncertainty level $\sigma$ i.e. $\mathcal{P}^{TV}(\sigma)$ and $\mathcal{R}^{TV}(\sigma)$, set $\psi = \frac{1}{2}$, for any $Q \in \mathbb{R}^{\mathcal{S} \times \mathcal{A}}, s \in \mathcal{S}, a \in \mathcal{A}$, the estimation bias can be bounded as:*

$$
\sup_{s,a} \left| \mathbb{E}\left[ \widehat{\mathcal{T}}^{\rho_{TV}(\sigma)}_{N_{\max}}(Q)(s,a) \right] - \mathcal{T}^{\rho_{TV}(\sigma)}(Q)(s,a) \right| \le \widetilde{\mathcal{O}}\left( N_{\max} 2^{-\frac{N_{\max}}{2}} \right),
$$

*The variance can be bounded as:*

$$
Var\left( \widehat{\mathcal{T}}^{\rho_{TV}(\sigma)}_{N_{\max}}(Q)(s,a) \right) \le \widetilde{\mathcal{O}}\left( N_{\max} \right). \tag{36}
$$

*Proof.* Firstly, we make error decomposition as follows:

$$
\sup_{s,a} \left| \mathbb{E}\left[ \widehat{\mathcal{T}}^{\rho_{TV}(\sigma)}_{N_{\max}}(Q)(s,a) \right] - \mathcal{T}^{\rho_{TV}(\sigma)}(Q)(s,a) \right|
$$
$$
\overset{(i)}{=} \sup_{s,a} \left| \mathbb{E}\left[ g^{*\rho_{TV}(\sigma)}(\hat{\mu}_{s,a,2^{N_{\max}+1}}, r_{s,a}) \right] + \gamma \mathbb{E}\left[ f^{*\rho_{TV}(\sigma)}(\hat{p}_{s,a,2^{N_{\max}+1}}, V) \right] \right.
$$
$$
\left. - g^{*\rho_{TV}(\sigma)}(\mu_{s,a}, r_{s,a}) - \gamma f^{*\rho_{TV}(\sigma)}(p_{s,a}, V) \right|
$$
$$
\le \sup_{s,a} \left| \mathbb{E}\left[ g^{*\rho_{TV}(\sigma)}(\hat{\mu}_{s,a,2^{N_{\max}+1}}, r_{s,a}) \right] - g^{*\rho_{TV}(\sigma)}(\mu_{s,a}, r_{s,a}) \right|
$$
$$
+ \gamma \sup_{s,a} \left| \mathbb{E}\left[ f^{*\rho_{TV}(\sigma)}(\hat{p}_{s,a,2^{N_{\max}+1}}, V) \right] - f^{*\rho_{TV}(\sigma)}(p_{s,a}, V) \right|
$$
$$
\le \mathbb{E}\left[ \sup_{s,a} \left| g^{*\rho_{TV}(\sigma)}(\hat{\mu}_{s,a,2^{N_{\max}+1}}, r_{s,a}) - g^{*\rho_{TV}(\sigma)}(\mu_{s,a}, r_{s,a}) \right| \right]
$$
$$
+ \gamma \mathbb{E}\left[ \sup_{s,a} \left| f^{*\rho_{TV}(\sigma)}(\hat{p}_{s,a,2^{N_{\max}+1}}, V) - f^{*\rho_{TV}(\sigma)}(p_{s,a}, V) \right| \right] \tag{37}
$$

where $(i)$ follows from Proposition B.2.

For convenience, we only bound the second term in Equation (37). The first term can be bounded similarly. By Lemma 2.1,

$$
\left| f^{*\rho_{TV}(\sigma)}(\hat{p}_{s,a,2^{N_{\max}+1}}, V) - f^{*\rho_{TV}(\sigma)}(p_{s,a}, V) \right|
$$

$$
= \left| \max_{\alpha \geq 0} \left\{ \mathbb{E}_{p_{s,a}} \left[ (V(s'_{s,a}))_\alpha \right] - \frac{\sigma}{2} \left( \alpha - \min_{s'_{s,a}} V(s'_{s,a}) \right) \right\} \right.
$$

$$
\left. - \max_{\alpha \geq 0} \left\{ \mathbb{E}_{\hat{p}_{s,a,2^{N_{\max}+1}}} \left[ (V(s'_{s,a}))_\alpha \right] - \frac{\sigma}{2} \left( \alpha - \min_{s'_{s,a}} V(s'_{s,a}) \right) \right\} \right|
$$

$$
\leq \max_{0 \leq \alpha \leq \max_{s'_{s,a}} V(s'_{s,a})} \left| \mathbb{E}_{p_{s,a}} \left[ (V(s'_{s,a}))_\alpha \right] - E_{\hat{p}_{s,a,2^{N_{\max}+1}}} \left[ (V(s'_{s,a}))_\alpha \right] \right|. \tag{38}
$$

Similarly to Lemma 9 in [Shi et al., 2023], we have the following lemma.

**Lemma C.2.** *Consider the case of TV constraint uncertainty set $\mathcal{P}^{TV}(\sigma)$ with uncertainty level $\sigma$, for any $\delta \in (0,1)$, one has with probability at least $1 - \delta$,*

$$
\max_{0 \leq \alpha \leq \max_{s'_{s,a}} V(s'_{s,a})} \left| \mathbb{E}_{p_{s,a}} \left[ (V(s'_{s,a}))_\alpha \right] - \mathbb{E}_{\hat{p}_{s,a,N}} \left[ (V(s'_{s,a}))_\alpha \right] \right| \leq 3 r_{\max} \sqrt{\frac{\log\left(\frac{18N}{\delta}\right)}{(1-\gamma)^2 N}}. \tag{39}
$$

According to Lemma C.2, it can be shown that with probability at least $1 - \frac{2^{-N_{\max}-1}}{|\mathcal{S}||\mathcal{A}|}$, we have

$$
\max_{0 \leq \alpha \leq \max_{s'_{s,a}} V(s'_{s,a})} \left| \mathbb{E}_{p_{s,a}} \left[ (V(s'_{s,a}))_\alpha \right] - E_{\hat{p}_{s,a,2^{N_{\max}+1}}} \left[ (V(s'_{s,a}))_\alpha \right] \right|
$$

$$
\leq 3 \sqrt{\frac{r_{\max}^2 \left( \log\left(18|\mathcal{S}||\mathcal{A}|\right) + 2(N_{\max}+1)\log 2 \right)}{(1-\gamma)^2 2^{N_{\max}+1}}}; \tag{40}
$$

Then, according to the Bernoulli's inequality, we have that

$$
\left( 1 - \frac{2^{-N_{\max}-1}}{|\mathcal{S}||\mathcal{A}|} \right)^{|\mathcal{S}||\mathcal{A}|} \geq 1 - 2^{-N_{\max}-1}. \tag{41}
$$

Therefore, with probability at least $1 - 2^{-N_{\max}-1}$, we have that

$$
\sup_{s,a} \left\{ \max_{0 \leq \alpha \leq \max_{s'_{s,a}} V(s'_{s,a})} \left| \mathbb{E}_{p_{s,a}} \left[ (V(s'_{s,a}))_\alpha \right] - E_{\hat{p}_{s,a,2^{N_{\max}+1}}} \left[ (V(s'_{s,a}))_\alpha \right] \right| \right\} \leq \frac{r_{\max} C_{TV}}{(1-\gamma) 2^{\frac{N_{\max}+1}{2}}}, \tag{42}
$$

where we set $C_{TV} = 3\sqrt{2(N_{\max}+1)\log 2 + \log(18|\mathcal{S}||\mathcal{A}|)}$.

With probability $2^{-N_{\max}-1}$, we directly have that

$$
\sup_{s,a} \left\{ \max_{0 \leq \alpha \leq \max_{s'_{s,a}} V(s'_{s,a})} \left| \mathbb{E}_{p_{s,a}} \left[ (V(s'_{s,a}))_\alpha \right] - E_{\hat{p}_{s,a,2^{N_{\max}+1}}} \left[ (V(s'_{s,a}))_\alpha \right] \right| \right\} \leq \sup_{s,a} \max_{s'_{s,a}} V(s'_{s,a}) \leq \frac{r_{\max}}{1-\gamma}. \tag{43}
$$

Hence, combining both cases together with Equation (38), we further have that

$$
\mathbb{E} \left[ \sup_{s,a} \left| f^{*\rho_{TV}(\sigma)}(\hat{p}_{s,a,2^{N_{\max}+1}}, V) - f^{*\rho_{TV}(\sigma)}(p_{s,a}, V) \right| \right]
$$

$$
\stackrel{(i)}{\leq} \frac{r_{\max} C_{TV}}{(1-\gamma)} 2^{-\frac{N_{\max}+1}{2}} + \frac{r_{\max}}{1-\gamma} 2^{-(N_{\max}+1)}
$$

$$
\leq \frac{r_{\max}}{1-\gamma} 2^{-\frac{N_{\max}+1}{2}} \left( 2^{-\frac{N_{\max}+1}{2}} + C_{TV} \right), \tag{44}
$$

where $(i)$ follows from that $1 - 2^{-(N_{\max}+1)} \leq 1$.

Similarly, we can get the bound

$$\mathbb{E}\left[\sup_{s,a}\left|g^{*\rho_{TV}(\sigma)}(\hat{\mu}_{s,a,2^{N_{\max}+1}}, r_{s,a}) - g^{*\rho_{TV}(\sigma)}(\mu_{s,a}, r_{s,a})\right|\right]$$
$$\leq r_{\max}2^{-\frac{N_{\max}+1}{2}}\left(2^{-\frac{N_{\max}+1}{2}} + C_{TV}\right). \tag{45}$$

Thus, we can get that

$$\sup_{s,a}\left|\mathbb{E}\left[\widehat{\mathcal{T}}_{N_{\max}}^{\rho_{TV}(\sigma)}(Q)(s,a)\right] - \mathcal{T}^{\rho_{TV}(\sigma)}(Q)(s,a)\right|$$
$$\leq \mathbb{E}\left[\sup_{s,a}\left|g^{*\rho_{TV}(\sigma)}(\hat{\mu}_{s,a,2^{N_{\max}+1}}, r_{s,a}) - g^{*\rho_{TV}(\sigma)}(\mu_{s,a}, r_{s,a})\right|\right]$$
$$+ \gamma\mathbb{E}\left[\sup_{s,a}\left|f^{*\rho_{TV}(\sigma)}(\hat{p}_{s,a,2^{N_{\max}+1}}, V) - f^{*\rho_{TV}(\sigma)}(p_{s,a}, V)\right|\right]$$
$$\leq \left(\frac{\gamma r_{\max}}{1-\gamma} + r_{\max}\right)2^{-\frac{N_{\max}+1}{2}}\left(2^{-\frac{N_{\max}+1}{2}} + C_{TV}\right). \tag{46}$$

We then consider the variance of the robust Bellman operator. Firstly, we make an error decomposition of the robust Bellman operator variance as

$$\mathrm{Var}\left(\widehat{\mathcal{T}}_{N_{\max}}^{\rho_{TV}(\sigma)}(Q)(s,a)\right) = \mathrm{Var}\left(\widehat{r}^{\rho_{TV}(\sigma)} + \gamma\widehat{v}^{\rho_{TV}(\sigma)}(Q)(s,a)\right)$$
$$= \mathrm{Var}\left(\widehat{r}^{\rho_{TV}(\sigma)}\right) + \gamma^2\mathrm{Var}\left(\widehat{v}^{\rho_{TV}(\sigma)}(Q)(s,a)\right), \tag{47}$$

which is due to the two estimators are independent.

For convenience, we analyze the second term in the above equation. The first term can be bounded similarly.

$$\mathrm{Var}\left(\widehat{v}^{\rho_{TV}(\sigma)}(Q)(s,a)\right) = \mathbb{E}\left[\left(\widehat{v}^{\rho_{TV}(\sigma)}(Q)(s,a)\right)^2\right] - \left(\mathbb{E}\left[\widehat{v}^{\rho_{TV}(\sigma)}(Q)(s,a)\right]\right)^2$$
$$\leq \mathbb{E}\left[\left(\widehat{v}^{\rho_{TV}(\sigma)}(Q)(s,a)\right)^2\right]. \tag{48}$$

Next, according to the Equations (13) and (14), the term above can be explicitly computed:

$$\mathbb{E}\left[\left(\widehat{v}^{\rho_{TV}(\sigma)}(Q)(s,a)\right)^2\right] = \mathbb{E}\left[\left(V(s'_{s,a,0}) + \frac{\delta_{s,a,N_2}^{\rho_{TV}(\sigma)}(Q)(s,a)}{P_{N_2}}\right)^2\right]$$
$$\leq 2\mathbb{E}\left[V(s'_{s,a,0})^2\right] + 2\mathbb{E}\left[\left(\frac{\delta_{s,a,N_2}^{\rho_{TV}(\sigma)}(Q)(s,a)}{P_{N_2}}\right)^2\right]$$
$$\leq \frac{2r_{\max}^2}{(1-\gamma)^2} + 2\sum_{N=0}^{N_{\max}}\mathbb{E}\left[\left(\frac{\delta_{s,a,N_2}^{\rho_{TV}(\sigma)}(Q)(s,a)}{P_{N_2}}|N_2 = N\right)^2\right]P_N$$
$$\leq \frac{2r_{\max}^2}{(1-\gamma)^2} + 2\sum_{N=0}^{N_{\max}}\frac{\mathbb{E}\left[(\delta_{s,a,N}^{\rho_{TV}(\sigma)}(Q)(s,a))^2\right]}{P_N}. \tag{49}$$

Next, we bound the term $\left|\delta_{s,a,N}^{\rho_{TV}(\sigma)}(Q)(s,a)\right|^2$, where

$$\left|\delta_{s,a,N}^{\rho_{TV}(\sigma)}(Q)(s,a)\right| = \left|\sup_{\alpha \geq 0}\left\{f^{\rho_{TV}(\sigma)}(\widehat{p}_{s,a,2^{N+1}},\alpha,V)\right\}\right.$$
$$\left. -\frac{1}{2}\sup_{\alpha \geq 0}\left\{f^{\rho_{TV}(\sigma)}(\widehat{p}_{s,a,2^N}^E,\alpha,V)\right\} - \frac{1}{2}\sup_{\alpha \geq 0}\left\{f^{\rho_{TV}(\sigma)}(\widehat{p}_{s,a,2^N}^O,\alpha,V)\right\}\right|. \tag{50}$$

We make an error decomposition as follows:

$$\left|\delta_{s,a,N}^{\rho_{TV}(\sigma)}(Q)(s,a)\right|^2 = \left|\sup_{\alpha \geq 0}\left\{f^{\rho_{TV}(\sigma)}(\widehat{p}_{s,a,2^{N+1}},\alpha,V)\right\}\right.$$
$$\left. -\frac{1}{2}\sup_{\alpha \geq 0}\left\{f^{\rho_{TV}(\sigma)}(\widehat{p}_{s,a,2^N}^E,\alpha,V)\right\} - \frac{1}{2}\sup_{\alpha \geq 0}\left\{f^{\rho_{TV}(\sigma)}(\widehat{p}_{s,a,2^N}^O,\alpha,V)\right\}\right|^2$$
$$\leq 3\left|\sup_{\alpha \geq 0}\left\{f^{\rho_{TV}(\sigma)}(\widehat{p}_{s,a,2^{N+1}},\alpha,V)\right\} - \sup_{\alpha \geq 0}\left\{f^{\rho_{TV}(\sigma)}(p_{s,a},\alpha,V)\right\}\right|^2$$
$$+\frac{3}{4}\left|\sup_{\alpha \geq 0}\left\{f^{\rho_{TV}(\sigma)}(\widehat{p}_{s,a,2^N}^E,\alpha,V)\right\} - \sup_{\alpha \geq 0}\left\{f^{\rho_{TV}(\sigma)}(p_{s,a},\alpha,V)\right\}\right|^2$$
$$+\frac{3}{4}\left|\sup_{\alpha \geq 0}\left\{f^{\rho_{TV}(\sigma)}(\widehat{p}_{s,a,2^N}^O,\alpha,V)\right\} - \sup_{\alpha \geq 0}\left\{f^{\rho_{TV}(\sigma)}(p_{s,a},\alpha,V)\right\}\right|^2. \tag{51}$$

Then, combined with the analysis in Equations (42) and (43) and the fact $\mathbb{P}(A \cap B \cap C) \geq 1 - \mathbb{P}(\neg A) - \mathbb{P}(\neg B) - \mathbb{P}(\neg C)$, we can conclude that with probability at least $1 - 3 * 2^{-N}$

$$\left|\delta_{s,a,N}^{\rho_{TV}(\sigma)}(Q)(s,a)\right|^2 \leq 3\left(C_{TV}\frac{r_{\max}}{1-\gamma}2^{-\frac{N+1}{2}}\right)^2 + \frac{3}{4}\left(C_{TV}\frac{r_{\max}}{1-\gamma}2^{-\frac{N}{2}}\right)^2 + \frac{3}{4}\left(C_{TV}\frac{r_{\max}}{1-\gamma}2^{-\frac{N}{2}}\right)^2$$
$$\leq 3\frac{C_{TV}^2 r_{\max}^2}{(1-\gamma)^2}2^{-(N+1)}, \tag{52}$$

Since $0 \leq \sup_{\alpha \geq 0}\left\{f^{\rho_{TV}(\sigma)}(q,\alpha,V)\right\} \leq \frac{r_{\max}}{1-\gamma}$ for any distribution $q$, with probability at most $3 * 2^{-N}$ we have that

$$\left|\delta_{s,a,N}^{\rho_{TV}(\sigma)}(Q)(s,a)\right|^2 \leq \left(\frac{r_{\max}}{1-\gamma}\right)^2. \tag{53}$$

Above all, we can get that

$$\mathbb{E}\left[\left|\delta_{s,a,N}^{\rho_{TV}(\sigma)}(Q)(s,a)\right|^2\right] \leq 3\frac{C_{TV}^2 r_{\max}^2}{(1-\gamma)^2}2^{-(N+1)} + \left(\frac{r_{\max}}{1-\gamma}\right)^2 3 * 2^{-N}$$
$$\leq \left(3C_{TV}^2 + 6\right)\left(\frac{r_{\max}}{1-\gamma}\right)^2 2^{-N-1}. \tag{54}$$

Then, plug Equation (54) in Equation (49), we can get the bound of variance of robust Bellman operator as follows:

$$\text{Var}\left(\widehat{v}^{\rho_{TV}(\sigma)}(Q)(s,a)\right) \leq \mathbb{E}\left[\left(\widehat{v}^{\rho_{TV}(\sigma)}(Q)(s,a)\right)^2\right]$$
$$\leq \frac{2r_{\max}^2}{(1-\gamma)^2} + 2\sum_{N=0}^{N_{\max}}\frac{\mathbb{E}\left[(\delta_{s,a,N}^{\rho_{TV}(\sigma)}(Q)(s,a))^2\right]}{P_N}$$
$$\leq \frac{2r_{\max}^2}{(1-\gamma)^2} + \frac{2r_{\max}^2}{(1-\gamma)^2}\left(3C_{TV}^2 + 6\right)\sum_{N=0}^{N_{\max}}\frac{2^{-N-1}}{P_N}$$

$$\overset{(i)}{\leq} \frac{2r_{\max}^2}{(1-\gamma)^2} \left(1 + \left(3C_{TV}^2 + 6\right)(N_{\max} + 1)\right), \tag{55}$$

where $(i)$ follows from $P_N = \psi(1-\psi)^N = (1/2)^{N+1}$.

Similarly, we can get the bound of the variance $\mathrm{Var}\left(\widehat{r}^{\rho_{TV}(\sigma)}\right)$ as follows:

$$\mathrm{Var}\left(\widehat{r}^{\rho_{TV}(\sigma)}\right) \leq \mathbb{E}\left[\left(\widehat{r}^{\rho_{TV}(\sigma)}\right)^2\right] \leq 2r_{\max}^2(1 + \left(3C_{TV}^2 + 6\right)(N_{\max} + 1)). \tag{56}$$

Hence, we can get the robust Bellman operator variance bound:

$$\mathrm{Var}\left(\widehat{\mathcal{T}}_{N_{\max}}^{\rho_{TV}(\sigma)}(Q)(s,a)\right) = \mathrm{Var}\left(\widehat{r}^{\rho_{TV}(\sigma)}\right) + \gamma^2 \mathrm{Var}\left(\widehat{v}^{\rho_{TV}(\sigma)}(Q)(s,a)\right)$$

$$\leq \left(2r_{\max}^2 + \gamma^2 \frac{2r_{\max}^2}{(1-\gamma)^2}\right)(1 + \left(3C_{TV}^2 + 6\right)(N_{\max} + 1)). \tag{57}$$

This completes the proof.

$\square$

**Lemma C.3.** *For any fixed $Q \in \mathbb{R}^{|\mathcal{S}||\mathcal{A}|}$, the infinite norm of robust Bellman operator can be bounded as:*

$$\mathbb{E}\left[\left\|\widehat{\mathcal{T}}_{N_{\max}}^{\rho_{TV}(\sigma)}(Q)\right\|_{\infty}^2\right] \leq \widetilde{\mathcal{O}}\left(N_{\max}\right). \tag{58}$$

*Proof of Lemma C.3.* We then consider the expectation of infinite norm of robust Bellman operator. Set $\psi = \frac{1}{2}$, from the construction of T-MLMC operator we directly have that

$$\mathbb{E}\left[\left\|\widehat{\mathcal{T}}_{N_{\max}}^{\rho_{TV}(\sigma)}(Q)\right\|_{\infty}^2\right] \leq 4r_{\max}^2 + 4\gamma^2 \frac{r_{\max}^2}{(1-\gamma)^2} + 4\mathbb{E}\left[\sum_{N_1=0}^{N_{\max}} \sup_{s,a} \frac{\left(\delta_{s,a,N_1}^{r,\rho(\sigma)}\right)^2}{2^{-N_1-1}} + \gamma^2 \sum_{N_2=0}^{N_{\max}} \sup_{s,a} \frac{\left(\delta_{s,a,N_2}^{\rho_{TV}(\sigma)}(Q)\right)^2}{2^{-N_2-1}}\right]. \tag{59}$$

For convenience, we analyze the last term in the above equation. Consider the term $\sup_{s,a}\left|\delta_{s,a,N}^{\rho_{TV}(\sigma)}(Q)(s,a)\right|^2$, we make an error decomposition as follows:

$$\sup_{s,a}\left|\delta_{s,a,N}^{\rho_{TV}(\sigma)}(Q)(s,a)\right|^2 = \sup_{s,a}\left|\sup_{\alpha\geq 0}\left\{f^{\rho_{TV}(\sigma)}(\widehat{p}_{s,a,2^{N+1}}, \alpha, V)\right\}\right.$$

$$\left. - \frac{1}{2}\sup_{\alpha\geq 0}\left\{f^{\rho_{TV}(\sigma)}(\widehat{p}_{s,a,2^N}^E, \alpha, V)\right\} - \frac{1}{2}\sup_{\alpha\geq 0}\left\{f^{\rho_{TV}(\sigma)}(\widehat{p}_{s,a,2^N}^O, \alpha, V)\right\}\right|^2$$

$$\leq 3\sup_{s,a}\left|\sup_{\alpha\geq 0}\left\{f^{\rho_{TV}(\sigma)}(\widehat{p}_{s,a,2^{N+1}}, \alpha, V)\right\} - \sup_{\alpha\geq 0}\left\{f^{\rho_{TV}(\sigma)}(p_{s,a}, \alpha, V)\right\}\right|^2$$

$$+ \frac{3}{4}\sup_{s,a}\left|\sup_{\alpha\geq 0}\left\{f^{\rho_{TV}(\sigma)}(\widehat{p}_{s,a,2^N}^E, \alpha, V)\right\} - \sup_{\alpha\geq 0}\left\{f^{\rho_{TV}(\sigma)}(p_{s,a}, \alpha, V)\right\}\right|^2$$

$$+ \frac{3}{4}\sup_{s,a}\left|\sup_{\alpha\geq 0}\left\{f^{\rho_{TV}(\sigma)}(\widehat{p}_{s,a,2^N}^O, \alpha, V)\right\} - \sup_{\alpha\geq 0}\left\{f^{\rho_{TV}(\sigma)}(p_{s,a}, \alpha, V)\right\}\right|^2. \tag{60}$$

Then, combined with the analysis in Equations (42) and (43) and the fact $\mathbb{P}(A \cap B \cap C) \geq 1 - \mathbb{P}(\neg A) - \mathbb{P}(\neg B) - \mathbb{P}(\neg C)$, we can conclude that for any $N \geq 0$, with probability at least $1 - 3*2^{-N}$

$$\sup_{s,a}\left|\delta_{s,a,N}^{\rho_{TV}(\sigma)}(Q)(s,a)\right|^2 \leq 3\left(C_{TV}\frac{r_{\max}}{1-\gamma}2^{-\frac{N+1}{2}}\right)^2 + \frac{3}{4}\left(C_{TV}\frac{r_{\max}}{1-\gamma}2^{-\frac{N}{2}}\right)^2 + \frac{3}{4}\left(C_{TV}\frac{r_{\max}}{1-\gamma}2^{-\frac{N}{2}}\right)^2$$

$$\leq 3\frac{C_{TV}^2 r_{\max}^2}{(1-\gamma)^2}2^{-(N+1)}, \tag{61}$$

Since $0 \leq \sup_{\alpha \geq 0} \left\{ f^{\rho_{TV}(\sigma)}(q, \alpha, V) \right\} \leq \frac{r_{\max}}{1-\gamma}$ for any distribution $q$, with probability at most $3 * 2^{-N}$ we have that

$$\sup_{s,a} \left| \delta_{s,a,N}^{\rho_{TV}(\sigma)}(Q)(s,a) \right|^2 \leq \left( \frac{r_{\max}}{1-\gamma} \right)^2. \tag{62}$$

Above all, we can get that

$$\mathbb{E}\left[ \sup_{s,a} \left| \delta_{s,a,N}^{\rho_{TV}(\sigma)}(Q)(s,a) \right|^2 \right] \leq 3 \frac{C_{TV}^2 r_{\max}^2}{(1-\gamma)^2} 2^{-(N+1)} + \left( \frac{r_{\max}}{1-\gamma} \right)^2 3 * 2^{-N}$$

$$\leq \left( 3C_{TV}^2 + 6 \right) \left( \frac{r_{\max}}{1-\gamma} \right)^2 2^{-N-1}. \tag{63}$$

Thus we have that

$$\mathbb{E}\left[ \sup_{s,a} \frac{\left| \delta_{s,a,N}^{\rho_{TV}(\sigma)}(Q)(s,a) \right|^2}{P_N} \right] = \left( 3C_{TV}^2 + 6 \right) \left( \frac{r_{\max}}{1-\gamma} \right)^2. \tag{64}$$

Then, plug Equation (64) in Equation (59), we can get the bound of expectation of infinite norm as follows:

$$\mathbb{E}\left[ \sum_{N_2=0}^{N_{\max}} \sup_{s,a} \frac{\left( \delta_{s,a,N_2}^{\rho_{TV}(\sigma)}(Q) \right)^2}{2^{-N_2-1}} \right] \leq \sum_{N_2=0}^{N_{\max}} \left( 3C_{TV}^2 + 6 \right) \left( \frac{r_{\max}}{1-\gamma} \right)^2 = \left( N_{\max} + 1 \right) \left( 3C_{TV}^2 + 6 \right) \left( \frac{r_{\max}}{1-\gamma} \right)^2. \tag{65}$$

Similarly, we can get the bound as follows:

$$\mathbb{E}\left[ \sum_{N_1=0}^{N_{\max}} \sup_{s,a} \frac{\left( \delta_{s,a,N_1}^{r,\rho(\sigma)} \right)^2}{2^{-N_1-1}} \right] \leq \sum_{N_1=0}^{N_{\max}} \left( 3C_{TV}^2 + 6 \right) r_{\max}^2 = \left( N_{\max} + 1 \right) \left( 3C_{TV}^2 + 6 \right) r_{\max}^2. \tag{66}$$

Hence, combining Equation (59) and the above equations, we can get the robust Bellman operator infinite norm bound:

$$\mathbb{E}\left[ \left\| \widehat{\mathcal{T}}_{N_{\max}}^{\rho_{TV}(\sigma)}(Q) \right\|_\infty^2 \right] \leq 4 \left( 1 + \left( N_{\max} + 1 \right) \left( 3C_{TV}^2 + 6 \right) \right) \left( r_{\max}^2 + \gamma^2 \left( \frac{r_{\max}}{1-\gamma} \right)^2 \right). \tag{67}$$

This completes the proof. $\qquad\square$

Next, we present the proof of Theorem 4.2

**Theorem C.4** (Restatement of Theorem 4.2). *Set $\psi = \frac{1}{2}$, and set the stepsize as*

$$\beta_t = \beta = \frac{2 \log T}{(1-\gamma)T}.$$

*Then the output from Algorithm 1 satisfies that:*

$$\mathbb{E}\left[ \left\| \widehat{Q}_T^{\rho_{TV}(\sigma)} - Q^{*\rho_{TV}(\sigma)} \right\|_\infty^2 \right] \leq \widetilde{\mathcal{O}}\left( \frac{1}{(1-\gamma)^5 T} \right).$$

*To obtain an $\epsilon$-optimal policy, i.e.,*

$$\mathbb{E}\left[ \left\| \widehat{Q}_T^{\rho_{TV}(\sigma)} - Q^{*,\rho_{TV}(\sigma)} \right\|_\infty^2 \right] \leq \epsilon^2,$$

*the expected sample complexity $N^{\rho_{TV}(\sigma)}(\epsilon)$ is*

$$N^{\rho_{TV}(\sigma)}(\epsilon) = |\mathcal{S}||\mathcal{A}|N_{\max}T \geq \widetilde{\mathcal{O}}\left( \frac{|\mathcal{S}||\mathcal{A}|}{(1-\gamma)^5 \epsilon^2} \right).$$

*Proof.* The update of our algorithm can be equivalently written as

$$\widehat{Q}_{t+1}^{\rho_{TV}(\sigma)} = \widehat{Q}_t^{\rho_{TV}(\sigma)} + \beta_t \left( \bar{\mathcal{T}}_{N_{\max}}^{\rho_{TV}(\sigma)} \left( Q_t^{\rho_{TV}(\sigma)} \right) - \widehat{Q}_t^{\rho_{TV}(\sigma)} + W_t \right), \tag{68}$$

where $W_t = \widehat{\mathcal{T}}_{N_{\max}}^{\rho_{TV}(\sigma)} \left( Q_t^{\rho_{TV}(\sigma)} \right) - \bar{\mathcal{T}}_{N_{\max}}^{\rho_{TV}(\sigma)} \left( Q_t^{\rho_{TV}(\sigma)} \right).$

Define the filtration $\mathcal{F}_t = \left\{ Q_0^{\rho_{TV}(\sigma)}, W_0, ..., Q_{t-1}^{\rho_{TV}(\sigma)}, W_{t-1}, Q_t^{\rho_{TV}(\sigma)} \right\}$. Note that by definition we have that

$$\mathbb{E}\left[W_t | \mathcal{F}_t\right] = 0, \tag{69}$$

and by Lemma C.3, we can get that

$$\mathbb{E}\left[ \|W_t\|_\infty^2 \, | \mathcal{F}_t \right] \leq \mathbb{E}\left[ \sup_{s,a} \left| \widehat{\mathcal{T}}_{N_{\max}}^{\rho_{TV}(\sigma)} \left( Q_t^{\rho_{TV}(\sigma)} \right)(s,a) - \bar{\mathcal{T}}_{N_{\max}}^{\rho_{TV}(\sigma)} \left( Q_t^{\rho_{TV}(\sigma)} \right)(s,a) \right|^2 | \mathcal{F}_t \right]$$

$$\leq 2\mathbb{E}\left[ \sup_{s,a} \left| \widehat{\mathcal{T}}_{N_{\max}}^{\rho_{TV}(\sigma)} \left( Q_t^{\rho_{TV}(\sigma)} \right)(s,a) \right|^2 + \sup_{s,a} \left| \bar{\mathcal{T}}_{N_{\max}}^{\rho_{TV}(\sigma)} \left( Q_t^{\rho_{TV}(\sigma)} \right)(s,a) \right|^2 | \mathcal{F}_t \right]$$

$$\overset{(a)}{\leq} 4\mathbb{E}\left[ \sup_{s,a} 2 \left| \widehat{\mathcal{T}}_{N_{\max}}^{\rho_{TV}(\sigma)} \left( Q_t^{\rho_{TV}(\sigma)} \right)(s,a) \right|^2 | \mathcal{F}_t \right]$$

$$\overset{(b)}{\leq} 16 \left( 1 + (N_{\max} + 1) \left( 3C_{TV}^2 + 6 \right) \right) \left( r_{\max}^2 + \gamma^2 \left( \frac{r_{\max}}{1-\gamma} \right)^2 \right), \tag{70}$$

where $(a)$ follows from that

$$\mathbb{E}\left[ \sup_{s,a} \left| \bar{\mathcal{T}}_{N_{\max}}^{\rho_{TV}(\sigma)} \left( Q_t^{\rho_{TV}(\sigma)} \right)(s,a) \right|^2 | \mathcal{F}_t \right] = \mathbb{E}\left[ \sup_{s,a} \left| \mathbb{E}\left[ \widehat{\mathcal{T}}_{N_{\max}}^{\rho_{TV}(\sigma)} \left( Q_t^{\rho_{TV}(\sigma)} \right)(s,a) \right] \right|^2 | \mathcal{F}_t \right]$$

$$\leq \mathbb{E}\left[ \sup_{s,a} \mathbb{E}\left[ \left| \widehat{\mathcal{T}}_{N_{\max}}^{\rho_{TV}(\sigma)} \left( Q_t^{\rho_{TV}(\sigma)} \right)(s,a) \right|^2 \right] | \mathcal{F}_t \right]$$

$$\leq \mathbb{E}\left[ \sup_{s,a} \left| \widehat{\mathcal{T}}_{N_{\max}}^{\rho_{TV}(\sigma)} \left( Q_t^{\rho_{TV}(\sigma)} \right)(s,a) \right|^2 | \mathcal{F}_t \right], \tag{71}$$

and $(b)$ follows from Lemma C.3.

According to Equation (23), we have that

$$\widehat{Q}^{*\rho_{TV}(\sigma)}(s,a) = \bar{\mathcal{T}}_{N_{\max}}^{\rho(\sigma)}(\widehat{Q}^{*\rho_{TV}(\sigma)})(s,a) = \mathbb{E}\left[ \widehat{\mathcal{T}}_{N_{\max}}^{\rho(\sigma)}(\widehat{Q}^{*\rho_{TV}(\sigma)})(s,a) \right].$$

Then, to apply Lemma B.5 [Chen et al., 2020], we set the constant stepsize $\beta_t = \beta = \frac{2\log T}{(1-\gamma)T}$. We note that as long as $T \geq \mathcal{O}\left( \frac{\log T}{(1-\gamma)^3} \right)$, it holds that

$$\beta = \frac{2\log T}{(1-\gamma)T} \leq \frac{(1-\gamma)^2}{128e \log(|\mathcal{S}||\mathcal{A}|)},$$

and the condition in Lemma B.5 are satisfied. We hence have that

$$\mathbb{E}\left[ \left\| \widehat{Q}_T^{\rho_{TV}(\sigma)} - \widehat{Q}^{*\rho_{TV}(\sigma)} \right\|_\infty^2 \right]$$

$$\overset{(i)}{\leq} \frac{3}{2} \left\| \widehat{Q}_0^{\rho_{TV}(\sigma)} - \widehat{Q}^{*\rho_{TV}(\sigma)} \right\|_\infty^2 \prod_{j=0}^{T-1} \left( 1 - \frac{1-\gamma}{2}\beta_t \right) + \frac{16e \log(|\mathcal{S}||\mathcal{A}|)}{1-\gamma} 16 \left( r_{\max}^2 + \gamma^2 \frac{r_{\max}^2}{(1-\gamma)^2} \right)$$

$$\cdot \left( 1 + (N_{\max} + 1) \left( 3C_{TV}^2 + 6 \right) \right) \sum_{i=0}^{T-1} \beta_i^2 \prod_{t=i+1}^{T-1} \left( 1 - \frac{1-\gamma}{2}\beta_t \right)$$

$$\overset{(ii)}{\leq} \frac{3}{2} \frac{r_{\max}^2}{(1-\gamma)^2} \frac{1}{T} + \frac{16e \log(|\mathcal{S}||\mathcal{A}|)}{1-\gamma} 16 \left( r_{\max}^2 + \gamma^2 \frac{r_{\max}^2}{(1-\gamma)^2} \right) \left( 1 + (N_{\max} + 1) \left( 3C_{TV}^2 + 6 \right) \right) \frac{4\log T}{(1-\gamma)^2 T}, \tag{72}$$

where $(i)$ follows from the Lemma B.5. $(ii)$ follows from $(1 - (1-\gamma)\beta/2)^T \leq \frac{1}{T}$.

We set $N_{\max} = \frac{2\log T}{\log 2}$, then the bound of $\mathbb{E}\left[\left\|\widehat{Q}_T^{\rho_{TV}(\sigma)} - Q^{*\rho_{TV}(\sigma)}\right\|_\infty^2\right]$ can be obtained as follows

$$
\begin{aligned}
&\mathbb{E}\left[\left\|\widehat{Q}_T^{\rho_{TV}(\sigma)} - Q^{*\rho_{TV}(\sigma)}\right\|_\infty^2\right] \\
&\leq 2\mathbb{E}\left[\left\|\widehat{Q}_T^{\rho_{TV}(\sigma)} - \widehat{Q}^{*\rho_{TV}(\sigma)}\right\|_\infty^2\right] + 2\mathbb{E}\left[\left\|\widehat{Q}^{*\rho_{TV}(\sigma)} - Q^{*\rho_{TV}(\sigma)}\right\|_\infty^2\right] \\
&\overset{(i)}{\leq} \frac{32e\log(|\mathcal{S}||\mathcal{A}|)}{1-\gamma} 16\left(r_{\max}^2 + \gamma^2\frac{r_{\max}^2}{(1-\gamma)^2}\right)\left(1 + (N_{\max}+1)\left(3C_{TV}^2 + 6\right)\right)\frac{4\log T}{(1-\gamma)^2 T} \\
&\quad + \frac{2r_{\max}^2}{(1-\gamma)^2 T} + \frac{2}{1-\gamma}\left(\left(r_{\max} + \frac{r_{\max}}{1-\gamma}\right)2^{-\frac{N_{\max}+1}{2}}\left(2^{-\frac{N_{\max}+1}{2}} + C_{TV}\right)\right)^2 \\
&\overset{(ii)}{\leq} \frac{32e\log(|\mathcal{S}||\mathcal{A}|)}{1-\gamma} 16\left(r_{\max}^2 + \gamma^2\frac{r_{\max}^2}{(1-\gamma)^2}\right)\left(1 + (N_{\max}+1)\left(3C_{TV}^2 + 6\right)\right)\frac{4\log T}{(1-\gamma)^2 T} \\
&\quad + \frac{2r_{\max}^2}{(1-\gamma)^2 T} + \frac{1}{1-\gamma}\left(\left(r_{\max} + \frac{r_{\max}}{1-\gamma}\right)(1 + C_{TV})\right)^2\frac{1}{T} \\
&= \widetilde{\mathcal{O}}\left(\frac{1}{(1-\gamma)^5 T}\right),
\end{aligned}
\tag{73}
$$

where $(i)$ follows from Lemma B.4 and Theorem 4.1. $(ii)$ follows from $2^{\frac{\log T}{\log 2}} \leq \frac{1}{T}$.

When $\mathbb{E}\left[\left\|\widehat{Q}_T^{\rho_{TV}(\sigma)} - Q^{*,\rho_{TV}(\sigma)}\right\|_\infty^2\right] \leq \epsilon^2$, the iteration $T \geq \widetilde{\mathcal{O}}\left((1-\gamma)^{-5}\epsilon^{-2}\right)$. When $\psi = \frac{1}{2}$, the expected sample size per iteration is

$$
1 + \sum_{n=0}^{N_{max}} 2^{n+1} P_N = 1 + \sum_{n=0}^{N_{max}} 2^{n+1}\frac{1}{2^{n+1}} = N_{\max} + 2 = \frac{2\log T}{\log 2} + 2.
$$

Above all, the total sample complexity is $\widetilde{\mathcal{O}}\left(|\mathcal{S}||\mathcal{A}|(1-\gamma)^{-5}\epsilon^{-2}\right)$.

This completes the proof. $\qquad\square$

# D $\chi^2$ DIVERGENCE UNCERTAINTY SET

Similar to the proof in the TV part, we can complete the proof. Here we show the detailed result.

**Theorem D.1** (Restatement of Theorem 4.1 specifically for $\chi^2$ distance)**.** *Set $\psi = \frac{1}{2}$, then for any $Q \in \mathbb{R}^{\mathcal{S}\times\mathcal{A}}, s \in \mathcal{S}, a \in \mathcal{A}$, the estimation bias can be bounded as:*

$$
\sup_{s,a}\left|\mathbb{E}\left[\widehat{\mathcal{T}}_{N_{\max}}^{\rho_{\chi^2}(\sigma)}(Q)(s,a)\right] - \mathcal{T}^{\rho_{\chi^2}(\sigma)}(Q)(s,a)\right| \leq \widetilde{\mathcal{O}}\left(2^{-\frac{N_{\max}}{2}}\right).
$$

*The variance can be bounded as:*

$$
Var\left(\widehat{\mathcal{T}}_{N_{\max}}^{\rho_{\chi^2}(\sigma)}(Q)(s,a)\right) \leq \widetilde{\mathcal{O}}\left(N_{\max}\right).
\tag{74}
$$

*Proof.* Firstly, we make error decomposition as follows:

$$\sup_{s,a} \left| \mathbb{E}\left[ \widehat{\mathcal{T}}_{N_{\max}}^{\rho_{\chi^2}(\sigma)}(Q)(s,a) \right] - \mathcal{T}^{\rho_{\chi^2}(\sigma)}(Q)(s,a) \right|$$

$$\overset{(i)}{=} \sup_{s,a} \left| \mathbb{E}\left[ g^{*\rho_{\chi^2}(\sigma)}(\hat{\mu}_{s,a,2^{N_{\max}+1}}, r_{s,a}) \right] + \gamma \mathbb{E}\left[ f^{*\rho_{\chi^2}(\sigma)}(\hat{p}_{s,a,2^{N_{\max}+1}}, V) \right] \right.$$

$$\left. - g^{*\rho_{\chi^2}(\sigma)}(\mu_{s,a}, r_{s,a}) - \gamma f^{*\rho_{\chi^2}(\sigma)}(p_{s,a}, V) \right|$$

$$\leq \sup_{s,a} \left| \mathbb{E}\left[ g^{*\rho_{\chi^2}(\sigma)}(\hat{\mu}_{s,a,2^{N_{\max}+1}}, r_{s,a}) \right] - g^{*\rho_{\chi^2}(\sigma)}(\mu_{s,a}, r_{s,a}) \right|$$

$$+ \gamma \sup_{s,a} \left| \mathbb{E}\left[ f^{*\rho_{\chi^2}(\sigma)}(\hat{p}_{s,a,2^{N_{\max}+1}}, V) \right] - f^{*\rho_{\chi^2}(\sigma)}(p_{s,a}, V) \right|$$

$$\leq \mathbb{E}\left[ \sup_{s,a} \left| g^{*\rho_{\chi^2}(\sigma)}(\hat{\mu}_{s,a,2^{N_{\max}+1}}, r_{s,a}) - g^{*\rho_{\chi^2}(\sigma)}(\mu_{s,a}, r_{s,a}) \right| \right]$$

$$+ \gamma \mathbb{E}\left[ \sup_{s,a} \left| f^{*\rho_{\chi^2}(\sigma)}(\hat{p}_{s,a,2^{N_{\max}+1}}, V) - f^{*\rho_{\chi^2}(\sigma)}(p_{s,a}, V) \right| \right], \tag{75}$$

where $(i)$ follows from Proposition B.2.

Then, for convenience, we bound the second term in Equation (75). The first term can be bounded similarly. By Lemma 2.1,

$$\left| f^{*\rho_{\chi^2}(\sigma)}(\hat{p}_{s,a,2^{N_{\max}+1}}, V) - f^{*\rho_{\chi^2}(\sigma)}(p_{s,a}, V) \right|$$

$$= \left| \max_{\alpha \geq 0} \left\{ \mathbb{E}_{p_{s,a}}\left[ (V(s'_{s,a}))_\alpha \right] - \sqrt{\sigma \text{Var}_{p_{s,a}}\left[ (V(s'_{s,a}))_\alpha \right]} \right\} \right.$$

$$\left. - \mathbb{E}\left[ \max_{\alpha \geq 0} \left\{ \mathbb{E}_{\hat{p}_{s,a,2^{N_{\max}+1}}}\left[ (V(s'_{s,a}))_\alpha \right] - \sqrt{\sigma \text{Var}_{\hat{p}_{s,a,2^{N_{\max}+1}}}\left[ (V'(s'_{s,a}))_\alpha \right]} \right\} \right] \right|$$

$$\leq \mathbb{E}\left[ \max_{\max_{s'_{s,a}} V(s'_{s,a}) \geq \alpha \geq 0} \left\{ \left| \mathbb{E}_{p_{s,a}}\left[ (V(s'_{s,a}))_\alpha \right] - E_{\hat{p}_{s,a,2^{N_{\max}+1}}}\left[ (V(s'_{s,a}))_\alpha \right] \right. \right. \right.$$

$$\left. \left. \left. + \sqrt{\sigma \text{Var}_{p_{s,a}}\left[ (V(s'_{s,a}))_\alpha \right]} - \sqrt{\sigma \text{Var}_{\hat{p}_{s,a,2^{N_{\max}+1}}}\left[ (V(s'_{s,a}))_\alpha \right]} \right| \right\} \right]. \tag{76}$$

According to Lemma 15 and its proof in [Shi et al., 2023], we have the following lemma.

**Lemma D.2.** *Consider the case of $\chi^2$ constraint uncertainty set $\mathcal{P}^{\chi^2}(\sigma)$ with uncertainty level $\sigma$, for any $\delta \in (0,1)$, one has with probability at least $1 - \delta$,*

$$\max_{0 \leq \alpha \leq \max_{s'_{s,a}} V(s'_{s,a})} \left| \mathbb{E}_{p_{s,a}}\left[ (V(s'_{s,a}))_\alpha \right] - \sqrt{\sigma Var_{p_{s,a}}\left[ (V(s'_{s,a}))_\alpha \right]} \right.$$

$$\left. - E_{\hat{p}_{s,a,N}}\left[ (V(s'_{s,a}))_\alpha \right] + \sqrt{\sigma Var_{\hat{p}_{s,a,2^{N_{\max}+1}}}\left[ (V'(s'_{s,a}))_\alpha \right]} \right| \leq 4\sqrt{\frac{2r_{\max}^2(1+\sigma)\log\left(\frac{24N}{\delta}\right)}{(1-\gamma)^2 N}}. \tag{77}$$

According to Lemma D.2, we can get that with probability at least $1 - \frac{2^{-N_{\max}-1}}{|\mathcal{S}||\mathcal{A}|}$, we have

$$\left| f^{*\rho_{\chi^2}(\sigma)}(\hat{p}_{s,a,2^{N_{\max}+1}}, V) - f^{*\rho_{\chi^2}(\sigma)}(p_{s,a}, V) \right|$$

$$\leq 4r_{\max}\sqrt{\frac{2(1+\sigma)\left(\log\left(24|\mathcal{S}||\mathcal{A}|\right) + 2(N_{\max}+1)\log 2\right)}{(1-\gamma)^2 2^{N_{\max}+1}}} = C_{\chi^2}\frac{r_{\max}}{1-\gamma} 2^{-\frac{N_{\max}+1}{2}}, \tag{78}$$

where $C_{\chi^2} = 4\sqrt{2(1+\sigma)\left(\log\left(24|\mathcal{S}||\mathcal{A}|\right) + 2(N_{\max}+1)\log 2\right)}$.

Then, according to the Bernoulli's inequality, we have that

$$\left(1 - \frac{2^{-N_{\max}-1}}{|\mathcal{S}||\mathcal{A}|}\right)^{|\mathcal{S}||\mathcal{A}|} \geq 1 - 2^{-N_{\max}-1}. \tag{79}$$

Therefore, with probability at least $1 - 2^{-N_{\max}-1}$, there exists

$$\sup_{s,a}\left|f^{*\rho_{\chi^2}(\sigma)}(\hat{p}_{s,a,2^{N_{\max}+1}}, V) - f^{*\rho_{\chi^2}(\sigma)}(p_{s,a}, V)\right| \leq C_{\chi^2}\frac{r_{\max}}{1-\gamma}2^{-\frac{N_{\max}+1}{2}}. \tag{80}$$

Otherwise, we have

$$\sup_{s,a}\left|\max_{\max_{s'_{s,a}}V(s'_{s,a})\geq\alpha\geq 0}\left\{\left|\mathbb{E}_{p_{s,a}}\left[(V(s'_{s,a}))_\alpha\right] - E_{\hat{p}_{s,a,2^{N_{\max}+1}}}\left[(V(s'_{s,a}))_\alpha\right]\right|\right.\right.$$
$$\left.\left. + \left|\sqrt{\sigma\mathrm{Var}_{p_{s,a}}\left[(V(s'_{s,a}))_\alpha\right]} - \sqrt{\sigma\mathrm{Var}_{\hat{p}_{s,a,2^{N_{\max}+1}}}\left[(V'(s'_{s,a}))_\alpha\right]}\right|\right\}\right|$$
$$\leq \sup_{s,a}\max_{s'_{s,a}}V(s'_{s,a}) + \sup_{s,a}\left|\max_{0\leq\alpha\leq\max_{s'_{s,a}}V(s'_{s,a})}\left|\sqrt{\sigma\mathrm{Var}_{p_{s,a}}\left[(V(s'_{s,a}))_\alpha\right]} - \sqrt{\sigma\mathrm{Var}_{\hat{p}_{s,a,2^{N_{\max}+1}}}\left[(V'(s'_{s,a}))_\alpha\right]}\right|\right|$$
$$\leq \frac{r_{\max}}{1-\gamma} + \sup_{s,a}\sqrt{\sigma\left(\max_{s'_{s,a}}V(s'_{s,a})\right)^2}$$
$$\leq (1+\sqrt{\sigma})\frac{r_{\max}}{1-\gamma}. \tag{81}$$

Plugging the above equations to Equation (90), we can conclude that

$$\mathbb{E}\left[\sup_{s,a}\left|f^{*\rho_{\chi^2}(\sigma)}(\hat{p}_{s,a,2^{N_{\max}+1}}, V) - f^{*\rho_{\chi^2}(\sigma)}(p_{s,a}, V)\right|\right] \overset{(i)}{\leq} C_{\chi^2}\frac{r_{\max}}{1-\gamma}2^{-\frac{N_{\max}+1}{2}} + (1+\sqrt{\sigma})\frac{r_{\max}}{1-\gamma}2^{-(N_{\max}+1)}$$
$$\leq \frac{r_{\max}}{1-\gamma}2^{-\frac{N_{\max}+1}{2}}\left((1+\sqrt{\sigma})2^{-\frac{N_{\max}+1}{2}} + C_{\chi^2}\right), \tag{82}$$

where $(i)$ follows from that $1 - 2^{-(N_{\max}+1)} \leq 1$.

Similarly, we can get the bound

$$\mathbb{E}\left[\sup_{s,a}\left|g^{*\rho_{\chi^2}(\sigma)}(\hat{\mu}_{s,a,2^{N_{\max}+1}}, r_{s,a}) - g^{*\rho_{\chi^2}(\sigma)}(\mu_{s,a}, r_{s,a})\right|\right] \leq r_{\max}2^{-\frac{N_{\max}+1}{2}}\left((1+\sqrt{\sigma})2^{-\frac{N_{\max}+1}{2}} + 3C_{\chi^2}\right). \tag{83}$$

Thus, we can get that

$$\sup_{s,a}\left|\mathbb{E}\left[\widehat{\mathcal{T}}_{N_{\max}}^{\rho_{\chi^2}(\sigma)}(Q)(s,a)\right] - \mathcal{T}^{\rho_{\chi^2}(\sigma)}(Q)(s,a)\right|$$
$$\leq \mathbb{E}\left[\sup_{s,a}\left|g^{*\rho_{\chi^2}(\sigma)}(\hat{\mu}_{s,a,2^{N_{\max}+1}}, r_{s,a}) - g^{*\rho_{\chi^2}(\sigma)}(\mu_{s,a}, r_{s,a})\right|\right]$$
$$+ \gamma\mathbb{E}\left[\sup_{s,a}\left|f^{*\rho_{\chi^2}(\sigma)}(\hat{p}_{s,a,2^{N_{\max}+1}}, V) - f^{*\rho_{\chi^2}(\sigma)}(p_{s,a}, V)\right|\right]$$
$$\leq \left(\frac{\gamma r_{\max}}{1-\gamma} + r_{\max}\right)2^{-\frac{N_{\max}+1}{2}}\left((1+\sqrt{\sigma})2^{-\frac{N_{\max}+1}{2}} + C_{\chi^2}\right). \tag{84}$$

**Variance:** Next, we consider the variance of the robust Bellman operator. Firstly, we make error decomposition of the robust Bellman operator variance.

$$\mathrm{Var}\left(\widehat{\mathcal{T}}_{N_{\max}}^{\rho_{\chi^2}(\sigma)}(Q)(s,a)\right) = \mathrm{Var}\left(\widehat{r}^{\rho_{\chi^2}(\sigma)} + \gamma\widehat{v}^{\rho_{\chi^2}(\sigma)}(Q)(s,a)\right)$$
$$= \mathrm{Var}\left(\widehat{r}^{\rho_{\chi^2}(\sigma)}\right) + \gamma^2\mathrm{Var}\left(\widehat{v}^{\rho_{\chi^2}(\sigma)}(Q)(s,a)\right). \tag{85}$$

For convenience, we analyze the second term in the above equation. The first term can be bounded similarly.

$$\text{Var}\left(\widehat{v}^{\rho_{\chi^2}(\sigma)}(Q)(s,a)\right) = \mathbb{E}\left[\left(\widehat{v}^{\rho_{\chi^2}(\sigma)}(Q)(s,a)\right)^2\right] - \left(\mathbb{E}\left[\widehat{v}^{\rho_{\chi^2}(\sigma)}(Q)(s,a)\right]\right)^2 \le \mathbb{E}\left[\left(\widehat{v}^{\rho_{\chi^2}(\sigma)}(Q)(s,a)\right)^2\right]. \quad (86)$$

Next, according to the Equations (13) and (14), now we compute the expectation of $N_2$ and write a detailed explanation of the variance as follows:

$$\mathbb{E}\left[\left(\widehat{v}^{\rho_{\chi^2}(\sigma)}(Q)(s,a)\right)^2\right] = \mathbb{E}\left[\left(V(s'_{s,a,0}) + \frac{\delta^{\rho_{\chi^2}(\sigma)}_{s,a,N_2}(Q)(s,a)}{P_{N_2}}\right)^2\right]$$

$$\le 2\mathbb{E}\left[V(s'_{s,a,0})^2\right] + 2\mathbb{E}\left[\left(\frac{\delta^{\rho_{\chi^2}(\sigma)}_{s,a,N_2}(Q)(s,a)}{P_{N_2}}\right)^2\right]$$

$$\le \frac{2r^2_{\max}}{(1-\gamma)^2} + 2\sum_{N=0}^{N_{\max}}\mathbb{E}\left[\left(\frac{\delta^{\rho_{\chi^2}(\sigma)}_{s,a,N_2}(Q)(s,a)}{P_{N_2}}\Big|N_2=N\right)^2\right]P_N$$

$$\le \frac{2r^2_{\max}}{(1-\gamma)^2} + 2\sum_{N=0}^{N_{\max}}\frac{\mathbb{E}\left[(\delta^{\rho_{\chi^2}(\sigma)}_{s,a,N}(Q)(s,a))^2\right]}{P_N}. \quad (87)$$

Next, we bound the term $\left|\delta^{\rho_{\chi^2}(\sigma)}_{s,a,N}(Q)(s,a)\right|$,

$$\left|\delta^{\rho_{\chi^2}(\sigma)}_{s,a,N}(Q)(s,a)\right|$$

$$= \left|\sup_{\alpha\ge0}\left\{f^{\rho_{\chi^2}(\sigma)}(\widehat{p}_{s,a,2^{N+1}},\alpha,V)\right\} - \frac{1}{2}\sup_{\alpha\ge0}\left\{f^{\rho_{\chi^2}(\sigma)}(\widehat{p}^E_{s,a,2^N},\alpha,V)\right\} - \frac{1}{2}\sup_{\alpha\ge0}\left\{f^{\rho_{\chi^2}(\sigma)}(\widehat{p}^O_{s,a,2^N},\alpha,V)\right\}\right|. \quad (88)$$

We make an error decomposition as follows:

$$\left|\delta^{\rho_{\chi^2}(\sigma)}_{s,a,N}(Q)(s,a)\right|^2$$

$$= \left|\sup_{\alpha\ge0}\left\{f^{\rho_{\chi^2}(\sigma)}(\widehat{p}_{s,a,2^{N+1}},\alpha,V)\right\} - \frac{1}{2}\sup_{\alpha\ge0}\left\{f^{\rho_{\chi^2}(\sigma)}(\widehat{p}^E_{s,a,2^N},\alpha,V)\right\} - \frac{1}{2}\sup_{\alpha\ge0}\left\{f^{\rho_{\chi^2}(\sigma)}(\widehat{p}^O_{s,a,2^N},\alpha,V)\right\}\right|^2$$

$$\le 3\left|\sup_{\alpha\ge0}\left\{f^{\rho_{\chi^2}(\sigma)}(\widehat{p}_{s,a,2^{N+1}},\alpha,V)\right\} - \sup_{\alpha\ge0}\left\{f^{\rho_{\chi^2}(\sigma)}(p_{s,a},\alpha,V)\right\}\right|^2$$

$$+ \frac{3}{4}\left|\sup_{\alpha\ge0}\left\{f^{\rho_{\chi^2}(\sigma)}(\widehat{p}^E_{s,a,2^N},\alpha,V)\right\} - \sup_{\alpha\ge0}\left\{f^{\rho_{\chi^2}(\sigma)}(p_{s,a},\alpha,V)\right\}\right|^2$$

$$+ \frac{3}{4}\left|\sup_{\alpha\ge0}\left\{f^{\rho_{\chi^2}(\sigma)}(\widehat{p}^O_{s,a,2^N},\alpha,V)\right\} - \sup_{\alpha\ge0}\left\{f^{\rho_{\chi^2}(\sigma)}(p_{s,a},\alpha,V)\right\}\right|^2. \quad (89)$$

According to Lemma D.2, we can get that with probability $1 - 2^{-N_{\max}-1}$, we have

$$\left|f^{*\rho_{\chi^2}(\sigma)}(\hat{p}_{s,a,2^{N_{\max}+1}},V) - f^{*\rho_{\chi^2}(\sigma)}(p_{s,a},V)\right|$$

$$\le 4\sqrt{\frac{2r^2_{\max}(1+\sigma)\left(\log\left(24|\mathcal{S}||\mathcal{A}|\right) + 2(N_{\max}+1)\log 2\right)}{(1-\gamma)^2 2^{N_{\max}+1}}} = C_{\chi^2}\frac{r_{\max}}{1-\gamma}2^{-\frac{N_{\max}+1}{2}}, \quad (90)$$

where $C_{\chi^2} = 4\sqrt{2(1+\sigma)\left(\log\left(24|\mathcal{S}||\mathcal{A}|\right) + 2(N_{\max}+1)\log 2\right)}$. Otherwise, we have

$$
\mathbb{E}\Bigg[\max_{s'_{s,a}} \max_{V(s'_{s,a})\geq\alpha\geq 0}\bigg\{\Big|\mathbb{E}_{p_{s,a}}\left[(V(s'_{s,a}))_\alpha\right] - E_{\hat{p}_{s,a,2^{N_{\max}+1}}}\left[(V(s'_{s,a}))_\alpha\right]\Big|
$$

$$
+ \left|\sqrt{\sigma\mathrm{Var}_{p_{s,a}}\left[(V(s'_{s,a}))_\alpha\right]} - \sqrt{\sigma\mathrm{Var}_{\hat{p}_{s,a,2^{N_{\max}+1}}}\left[(V'(s'_{s,a}))_\alpha\right]}\right|\bigg\}\Bigg]
$$

$$
\leq \max_{s'_{s,a}} V(s'_{s,a}) + \max_{0\leq\alpha\leq\max_{s'_{s,a}}V(s'_{s,a})}\left|\sqrt{\sigma\mathrm{Var}_{p_{s,a}}\left[(V(s'_{s,a}))_\alpha\right]} - \sqrt{\sigma\mathrm{Var}_{\hat{p}_{s,a,2^{N_{\max}+1}}}\left[(V'(s'_{s,a}))_\alpha\right]}\right|
$$

$$
\leq \max_{s'_{s,a}} V(s'_{s,a}) + \sqrt{\sigma\left(\max_{s'_{s,a}} V(s'_{s,a})\right)^2}
$$

$$
\leq (1+\sqrt{\sigma})\frac{r_{\max}}{1-\gamma}. \tag{91}
$$

Then, combined with the analysis in Equations (90) and (91) and the fact that for any events $A, B, C$, $\mathbb{P}(A \cap B \cap C) \geq 1 - \mathbb{P}(\neg A) - \mathbb{P}(\neg B) - \mathbb{P}(\neg C)$, we can conclude that with probability at least $1 - 3*2^{-N}$

$$
\left|\delta^{\rho_{\chi^2}(\sigma)}_{s,a,N}(Q)(s,a)\right|^2 \leq 3\left(C_{\chi^2}\frac{r_{\max}}{1-\gamma}2^{-\frac{N+1}{2}}\right)^2 + \frac{3}{4}\left(C_{\chi^2}\frac{r_{\max}}{1-\gamma}2^{-\frac{N}{2}}\right)^2 + \frac{3}{4}\left(C_{\chi^2}\frac{r_{\max}}{1-\gamma}2^{-\frac{N}{2}}\right)^2
$$

$$
= 3\frac{C_{\chi^2}^2 r_{\max}^2}{(1-\gamma)^2}2^{-(N+1)}, \tag{92}
$$

Since $0 \leq \sup_{\alpha\geq 0}\left\{f^{\rho_{\chi^2}(\sigma)}(q, \alpha, V)\right\} \leq (1+\sqrt{\sigma})\frac{r_{\max}}{1-\gamma}$ for any distribution $q$, with probability at most $3*2^{-N}$ we have that

$$
\left|\delta^{\rho_{\chi^2}(\sigma)}_{s,a,N}(Q)(s,a)\right|^2 \leq \left((1+\sqrt{\sigma})\frac{r_{\max}}{1-\gamma}\right)^2. \tag{93}
$$

Above all, we can get that

$$
\mathbb{E}\left[\left|\delta^{\rho_{\chi^2}(\sigma)}_{s,a,N}(Q)(s,a)\right|^2\right] \leq \frac{9}{2}\frac{C_{\chi^2}^2 r_{\max}^2}{(1-\gamma)^2}2^{-(N+1)} + \left(\frac{r_{\max}}{1-\gamma}\right)^2 3*2^{-N} \leq \left(3C_{\chi^2}^2 + 6(1+\sqrt{\sigma})^2\right)\left(\frac{r_{\max}}{1-\gamma}\right)^2 2^{-N-1}. \tag{94}
$$

Then, plug Equation (94) in Equation (87), we can get the boundary of variance of robust Bellman operator as follows:

$$
\mathrm{Var}\left(\widehat{v}^{\rho_{\chi^2}(\sigma)}(Q)(s,a)\right) \leq \frac{2r_{\max}^2}{(1-\gamma)^2} + 2\sum_{N=0}^{N_{\max}}\frac{\mathbb{E}\left[(\delta^{\rho_{\chi^2}(\sigma)}_{s,a,N}(Q)(s,a))^2\right]}{P_N}
$$

$$
\leq \frac{2r_{\max}^2}{(1-\gamma)^2} + \frac{2r_{\max}^2}{(1-\gamma)^2}\left(3C_{\chi^2}^2 + 6(1+\sqrt{\sigma})^2\right)\sum_{N=0}^{N_{\max}}\frac{2^{-N-1}}{P_N}
$$

$$
\overset{(a)}{=} \frac{2r_{\max}^2}{(1-\gamma)^2}\left(1 + \left(3C_{\chi^2}^2 + 6(1+\sqrt{\sigma})^2\right)(N_{\max}+1)\right), \tag{95}
$$

where $(a)$ follows from that $P_N = \psi(1-\psi)^N = 2^{-N-1}$.

Similarly, we can get the bound of the variance $\mathrm{Var}\left(\widehat{r}^{\rho_{\chi^2}(\sigma)}\right)$ as follows:

$$
\mathrm{Var}\left(\widehat{r}^{\rho_{\chi^2}(\sigma)}\right) \leq 2r_{\max}^2\left(1 + \left(3C_{\chi^2}^2 + 6(1+\sqrt{\sigma})^2\right)(N_{\max}+1)\right). \tag{96}
$$

Hence, we can get the robust Bellman operator variance bound:

$$
\begin{aligned}
\mathrm{Var}\left(\widehat{\mathcal{T}}_{N_{\max}}^{\rho_{\chi^2}(\sigma)}(Q)(s,a)\right) &= \mathrm{Var}\left(\widehat{r}^{\rho_{\chi^2}(\sigma)}\right) + \gamma^2 \mathrm{Var}\left(\widehat{v}^{\rho_{\chi^2}(\sigma)}(Q)(s,a)\right) \\
&\leq \left(2r_{\max}^2 + \gamma^2 \frac{2r_{\max}^2}{(1-\gamma)^2}\right)\left(1 + \left(3C_{\chi^2}^2 + 6(1+\sqrt{\sigma})^2\right)(N_{\max}+1)\right).
\end{aligned}
\tag{97}
$$

This completes the proof. $\qquad\square$

**Lemma D.3.** *For any fixed $Q \in \mathbb{R}^{|\mathcal{S}||\mathcal{A}|}, s \in \mathcal{S}, a \in \mathcal{A}$, the infinite norm of robust Bellman operator can be bounded as:*

$$
\mathbb{E}\left[\left\|\widehat{\mathcal{T}}_{N_{\max}}^{\rho_{\chi^2}(\sigma)}(Q)(s,a)\right\|_\infty^2\right] \leq \widetilde{\mathcal{O}}(N_{\max}).
\tag{98}
$$

*Proof of Lemma D.3.* We then consider the expectation of infinite norm of robust Bellman operator. Set $\psi = \frac{1}{2}$, and then we make an error decomposition as follows

$$
\mathbb{E}\left[\left\|\widehat{\mathcal{T}}_{N_{\max}}^{\rho_{\chi^2}(\sigma)}(Q)\right\|_\infty^2\right] \leq 4r_{\max}^2 + 4\gamma^2 \frac{r_{\max}^2}{(1-\gamma)^2} + 4\mathbb{E}\left[\sum_{N_1=0}^{N_{\max}} \sup_{s,a} \frac{\left(\delta_{s,a,N_1}^{r,\rho(\sigma)}\right)^2}{2^{-N_1-1}} + \gamma^2 \sum_{N_2=0}^{N_{\max}} \sup_{s,a} \frac{\left(\delta_{s,a,N_2}^{\rho_{\chi^2}(\sigma)}(Q)\right)^2}{2^{-N_2-1}}\right].
\tag{99}
$$

For convenience, we analyze the last term in the above equation. Consider the term $\sup_{s,a}\left|\delta_{s,a,N}^{\rho_{\chi^2}(\sigma)}(Q)(s,a)\right|^2$, we make an error decomposition as follows:

$$
\begin{aligned}
\sup_{s,a}\left|\delta_{s,a,N}^{\rho_{\chi^2}(\sigma)}(Q)(s,a)\right|^2 &= \sup_{s,a}\left|\sup_{\alpha\geq 0}\left\{f^{\rho_{\chi^2}(\sigma)}(\widehat{p}_{s,a,2^{N+1}},\alpha,V)\right\}\right. \\
&\qquad \left. - \frac{1}{2}\sup_{\alpha\geq 0}\left\{f^{\rho_{\chi^2}(\sigma)}(\widehat{p}_{s,a,2^N}^E,\alpha,V)\right\} - \frac{1}{2}\sup_{\alpha\geq 0}\left\{f^{\rho_{\chi^2}(\sigma)}(\widehat{p}_{s,a,2^N}^O,\alpha,V)\right\}\right|^2 \\
&\leq 3\sup_{s,a}\left|\sup_{\alpha\geq 0}\left\{f^{\rho_{\chi^2}(\sigma)}(\widehat{p}_{s,a,2^{N+1}},\alpha,V)\right\} - \sup_{\alpha\geq 0}\left\{f^{\rho_{\chi^2}(\sigma)}(p_{s,a},\alpha,V)\right\}\right|^2 \\
&\quad + \frac{3}{4}\sup_{s,a}\left|\sup_{\alpha\geq 0}\left\{f^{\rho_{\chi^2}(\sigma)}(p_{s,a,2^N}^E,\alpha,V)\right\} - \sup_{\alpha\geq 0}\left\{f^{\rho_{\chi^2}(\sigma)}(p_{s,a},\alpha,V)\right\}\right|^2 \\
&\quad + \frac{3}{4}\sup_{s,a}\left|\sup_{\alpha\geq 0}\left\{f^{\rho_{\chi^2}(\sigma)}(\widehat{p}_{s,a,2^N}^O,\alpha,V)\right\} - \sup_{\alpha\geq 0}\left\{f^{\rho_{\chi^2}(\sigma)}(p_{s,a},\alpha,V)\right\}\right|^2.
\end{aligned}
\tag{100}
$$

Then, combined with the analysis in Equations (80) and (81) and the fact $\mathbb{P}(A \cap B \cap C) \geq 1 - \mathbb{P}(\neg A) - \mathbb{P}(\neg B) - \mathbb{P}(\neg C)$, we can conclude that for any $N \geq 0$, with probability at least $1 - 3 * 2^{-N}$

$$
\begin{aligned}
\sup_{s,a}\left|\delta_{s,a,N}^{\rho_{\chi^2}(\sigma)}(Q)(s,a)\right|^2 &\leq 3\left(C_{\chi^2}\frac{r_{\max}}{1-\gamma}2^{-\frac{N+1}{2}}\right)^2 + \frac{3}{4}\left(C_{\chi^2}\frac{r_{\max}}{1-\gamma}2^{-\frac{N}{2}}\right)^2 + \frac{3}{4}\left(C_{\chi^2}\frac{r_{\max}}{1-\gamma}2^{-\frac{N}{2}}\right)^2 \\
&\leq 6\frac{C_{\chi^2}^2 r_{\max}^2}{(1-\gamma)^2}2^{-(N+1)},
\end{aligned}
\tag{101}
$$

Since $0 \leq \sup_{\alpha\geq 0}\left\{f^{\rho_{\chi^2}(\sigma)}(q,\alpha,V)\right\} \leq \frac{r_{\max}}{1-\gamma}$ for any distribution $q$, with probability at most $3 * 2^{-N}$ we have that

$$
\sup_{s,a}\left|\delta_{s,a,N}^{\rho_{\chi^2}(\sigma)}(Q)(s,a)\right|^2 \leq \left(\frac{r_{\max}}{1-\gamma}\right)^2.
\tag{102}
$$

Above all, we can get that

$$\mathbb{E}\left[\sup_{s,a}\left|\delta_{s,a,N}^{\rho_{\chi^2}(\sigma)}(Q)(s,a)\right|^2\right] \le 6\frac{C_{\chi^2}^2 r_{\max}^2}{(1-\gamma)^2}2^{-(N+1)} + \left(\frac{r_{\max}}{1-\gamma}\right)^2 3 * 2^{-N}$$

$$\le \left(6C_{\chi^2}^2 + 6\right)\left(\frac{r_{\max}}{1-\gamma}\right)^2 2^{-N-1}. \tag{103}$$

Besides, we can get

$$\mathbb{E}\left[\sup_{s,a}\frac{\left|\delta_{s,a,N}^{\rho_{\chi^2}(\sigma)}(Q)(s,a)\right|^2}{P_N}\right] = \left(6C_{\chi^2}^2 + 6\right)\left(\frac{r_{\max}}{1-\gamma}\right)^2. \tag{104}$$

Then, plug Equation (104) in Equation (99), we can get the bound of expectation of infinite norm as follows:

$$\mathbb{E}\left[\sum_{N_2=0}^{N_{\max}}\sup_{s,a}\frac{\left(\delta_{s,a,N_2}^{\rho_{\chi^2}(\sigma)}(Q)\right)^2}{2^{-N_2-1}}\right] \le \sum_{N_2=0}^{N_{\max}}\left(6C_{\chi^2}^2 + 6\right)\left(\frac{r_{\max}}{1-\gamma}\right)^2 = (N_{\max}+1)\left(6C_{\chi^2}^2+6\right)\left(\frac{r_{\max}}{1-\gamma}\right)^2. \tag{105}$$

Similarly, we can get the bound as follows:

$$\mathbb{E}\left[\sum_{N_1=0}^{N_{\max}}\sup_{s,a}\frac{\left(\delta_{s,a,N_1}^{r,\rho(\sigma)}\right)^2}{2^{-N_1-1}}\right] \le \sum_{N_1=0}^{N_{\max}}\left(6C_{\chi^2}^2+6\right)r_{\max}^2 = (N_{\max}+1)\left(6C_{\chi^2}^2+6\right)r_{\max}^2. \tag{106}$$

Hence, combining Equation (99) and the above equations, we can get the robust Bellman operator infinite norm bound:

$$\mathbb{E}\left[\left\|\widehat{\mathcal{T}}_{N_{\max}}^{\rho_{\chi^2}(\sigma)}(Q)\right\|_\infty^2\right] \le 4\left(1 + (N_{\max}+1)\left(6C_{\chi^2}^2+6\right)\right)\left(r_{\max}^2 + \gamma^2\left(\frac{r_{\max}}{1-\gamma}\right)^2\right). \tag{107}$$

This completes the proof. □

**Theorem D.4** (Sample Complexity with $\chi^2$ Distance). *Set $N_{\max} = \frac{2\log T}{\log 2}$ and the stepsize as*

$$\beta_t = \beta = \frac{2\log T}{(1-\gamma)T}.$$

*Then the output of Algorithm 1 satisfies that:*

$$\mathbb{E}\left[\left\|\widehat{Q}_T^{\rho_{\chi^2}(\sigma)} - Q^{*\rho_{\chi^2}(\sigma)}\right\|_\infty^2\right] \le \frac{\log T}{T^2}\frac{r_{\max}^2}{(1-\gamma)^2} + \frac{r_{\max}^2\log T}{2(1-\gamma)^2 T}\left(1+\frac{\gamma^2}{(1-\gamma)^2}\right)\left(1+\left(6C_{\chi^2}^2+6\right)(N_{\max}+1)\right)$$

$$+ r_{\max}^2\left(\left(1+\frac{\gamma}{1-\gamma}\right)\left(1+3C_{\chi^2}\right)\right)^2\frac{1}{T}$$

$$\le \widetilde{\mathcal{O}}\left(\frac{1}{(1-\gamma)^5 T}\right). \tag{108}$$

*To ensure*

$$\mathbb{E}\left[\left\|\widehat{Q}_T^{\rho_{\chi^2}(\sigma)} - Q^{*\rho_{\chi^2}(\sigma)}\right\|_\infty^2\right] \le \epsilon^2,$$

*the expected total sample complexity $N^{\rho_{\chi^2}(\sigma)}(\epsilon)$ is,*

$$N^{\rho_{\chi^2}(\sigma)}(\epsilon) = |\mathcal{S}||\mathcal{A}|N_{\max}T \ge \widetilde{\mathcal{O}}\left(\frac{|\mathcal{S}||\mathcal{A}|}{(1-\gamma)^5\epsilon^2}\right).$$

*Proof.* We consider the stochastic iteration

$$\widehat{Q}_{t+1}^{\rho_{\chi^2}(\sigma)} = \widehat{Q}_t^{\rho_{\chi^2}(\sigma)} + \beta_t \left( \bar{\mathcal{T}}_{N_{\max}}^{\rho_{\chi^2}(\sigma)} \left( Q_t^{\rho_{\chi^2}(\sigma)} \right) - \widehat{Q}_t^{\rho_{\chi^2}(\sigma)} + W_t \right), \tag{109}$$

where $W_t = \widehat{\mathcal{T}}_{N_{\max}}^{\rho_{\chi^2}(\sigma)} \left( Q_t^{\rho_{\chi^2}(\sigma)} \right) - \bar{\mathcal{T}}_{N_{\max}}^{\rho_{\chi^2}(\sigma)} \left( Q_t^{\rho_{\chi^2}(\sigma)} \right).$

Define the filtration $\mathcal{F}_t = \left\{ Q_0^{\rho_{\chi^2}(\sigma)}, W_0, ..., Q_{t-1}^{\rho_{\chi^2}(\sigma)}, W_{t-1}, Q_t^{\rho_{\chi^2}(\sigma)} \right\}$. Then, by Theorem 4.1, we can get that

$$\mathbb{E}\left[W_t | \mathcal{F}_t\right] = 0, \tag{110}$$

and by Lemma D.3, we can get that

$$\mathbb{E}\left[ \|W_t\|_\infty^2 | \mathcal{F}_t \right] \leq \mathbb{E}\left[ \sup_{s,a} \left| \widehat{\mathcal{T}}_{N_{\max}}^{\rho_{\chi^2}(\sigma)} \left( Q_t^{\rho_{\chi^2}(\sigma)} \right)(s,a) - \bar{\mathcal{T}}_{N_{\max}}^{\rho_{\chi^2}(\sigma)} \left( Q_t^{\rho_{\chi^2}(\sigma)} \right)(s,a) \right|^2 | \mathcal{F}_t \right]$$

$$\leq 2\mathbb{E}\left[ \sup_{s,a} \left| \widehat{\mathcal{T}}_{N_{\max}}^{\rho_{\chi^2}(\sigma)} \left( Q_t^{\rho_{\chi^2}(\sigma)} \right)(s,a) \right|^2 + \sup_{s,a} \left| \bar{\mathcal{T}}_{N_{\max}}^{\rho_{\chi^2}(\sigma)} \left( Q_t^{\rho_{\chi^2}(\sigma)} \right)(s,a) \right|^2 | \mathcal{F}_t \right]$$

$$\overset{(a)}{\leq} 4\mathbb{E}\left[ \sup_{s,a} 2 \left| \widehat{\mathcal{T}}_{N_{\max}}^{\rho_{\chi^2}(\sigma)} \left( Q_t^{\rho_{\chi^2}(\sigma)} \right)(s,a) \right|^2 | \mathcal{F}_t \right]$$

$$\overset{(b)}{\leq} 16 \left( 1 + (N_{\max}+1) \left( 6C_{\chi^2}^2 + 6 \right) \right) \left( r_{\max}^2 + \gamma^2 \left( \frac{r_{\max}}{1-\gamma} \right)^2 \right), \tag{111}$$

where $(a)$ follows from that

$$\mathbb{E}\left[ \sup_{s,a} \left| \bar{\mathcal{T}}_{N_{\max}}^{\rho_{\chi^2}(\sigma)} \left( Q_t^{\rho_{\chi^2}(\sigma)} \right)(s,a) \right|^2 | \mathcal{F}_t \right] = \mathbb{E}\left[ \sup_{s,a} \left| \mathbb{E}\left[ \widehat{\mathcal{T}}_{N_{\max}}^{\rho_{\chi^2}(\sigma)} \left( Q_t^{\rho_{\chi^2}(\sigma)} \right)(s,a) \right] \right|^2 | \mathcal{F}_t \right]$$

$$\leq \mathbb{E}\left[ \sup_{s,a} \mathbb{E}\left[ \left| \widehat{\mathcal{T}}_{N_{\max}}^{\rho_{\chi^2}(\sigma)} \left( Q_t^{\rho_{\chi^2}(\sigma)} \right)(s,a) \right|^2 \right] | \mathcal{F}_t \right]$$

$$\leq \mathbb{E}\left[ \sup_{s,a} \left| \widehat{\mathcal{T}}_{N_{\max}}^{\rho_{\chi^2}(\sigma)} \left( Q_t^{\rho_{\chi^2}(\sigma)} \right)(s,a) \right|^2 | \mathcal{F}_t \right], \tag{112}$$

and $(b)$ follows from Lemma D.3.

According to Equation (23), we have that

$$\widehat{Q}^{*\rho_{\chi^2}(\sigma)}(s,a) = \bar{\mathcal{T}}_{N_{\max}}^{\rho(\sigma)}(\widehat{Q}^{*\rho_{\chi^2}(\sigma)})(s,a) = \mathbb{E}\left[ \widehat{\mathcal{T}}_{N_{\max}}^{\rho(\sigma)}(\widehat{Q}^{*\rho_{\chi^2}(\sigma)})(s,a) \right].$$

Then, apply Lemma B.5 [Chen et al., 2020], set the constant stepsize $\beta_t = \beta = \frac{2\log T}{(1-\gamma)T}$ and $T$ is large enough s.t.

$$\beta = \frac{2\log T}{(1-\gamma)T} \leq \frac{(1-\gamma)^2}{128e\log(|\mathcal{S}||\mathcal{A}|)}.$$

We can conclude that

$$\mathbb{E}\left[ \left\| \widehat{Q}_T^{\rho_{\chi^2}(\sigma)} - \widehat{Q}^{*\rho_{\chi^2}(\sigma)} \right\|_\infty^2 \right] \overset{(i)}{\leq} \frac{3}{2} \left\| \widehat{Q}_0^{\rho_{\chi^2}(\sigma)} - \widehat{Q}^{*\rho_{\chi^2}(\sigma)} \right\|_\infty^2 \prod_{j=0}^{T-1} \left( 1 - \frac{1-\gamma}{2}\beta_t \right)$$

$$+ \frac{16e\log(|\mathcal{S}||\mathcal{A}|)}{1-\gamma} 16 \left( 1 + (N_{\max}+1)\left(6C_{\chi^2}^2 + 6\right) \right) \left( r_{\max}^2 + \gamma^2 \left( \frac{r_{\max}}{1-\gamma} \right)^2 \right) \sum_{i=0}^{T-1} \beta_i^2 \prod_{t=i+1}^{T-1} (1 - \frac{1-\gamma}{2}\beta_t)$$

$$\overset{(ii)}{\leq} \frac{3}{2} \frac{r_{\max}^2}{(1-\gamma)^2} \frac{1}{T} + \frac{16e\log(|\mathcal{S}||\mathcal{A}|)}{1-\gamma} 16 \left( 1 + (N_{\max}+1)\left(6C_{\chi^2}^2 + 6\right) \right) \left( r_{\max}^2 + \gamma^2 \left( \frac{r_{\max}}{1-\gamma} \right)^2 \right) \frac{4\log T}{(1-\gamma)^2 T}, \tag{113}$$

where $(i)$ follows from the Lemma B.5. $(ii)$ follows from $(1 - (1-\gamma)\beta/2)^T \leq \frac{1}{T}$.

Set $N_{\max} = \frac{2\log T}{\log 2}$. Then, we make the decomposition and get the bound of $\mathbb{E}\left[\left\|\widehat{Q}_T^{\rho_{\chi^2}(\sigma)} - Q^{*\rho_{\chi^2}(\sigma)}\right\|_\infty^2\right]$ as follows

$$
\mathbb{E}\left[\left\|\widehat{Q}_T^{\rho_{\chi^2}(\sigma)} - Q^{*\rho_{\chi^2}(\sigma)}\right\|_\infty^2\right] \le 2\mathbb{E}\left[\left\|\widehat{Q}_T^{\rho_{\chi^2}(\sigma)} - \widehat{Q}^{*\rho_{\chi^2}(\sigma)}\right\|_\infty^2\right] + 2\mathbb{E}\left[\left\|\widehat{Q}^{*\rho_{\chi^2}(\sigma)} - Q^{*\rho_{\chi^2}(\sigma)}\right\|_\infty^2\right]
$$

$$
\overset{(i)}{\le} \frac{2r_{\max}^2}{(1-\gamma)^2 T} + \frac{32e\log(|\mathcal{S}||\mathcal{A}|)}{1-\gamma} 16\left(1 + (N_{\max}+1)\left(6C_{\chi^2}^2 + 6\right)\right)\left(r_{\max}^2 + \gamma^2\left(\frac{r_{\max}}{1-\gamma}\right)^2\right)\frac{4\log T}{(1-\gamma)^2 T}
$$

$$
+ \frac{2}{1-\gamma}\left(\left(r_{\max} + \frac{r_{\max}}{1-\gamma}\right)2^{-\frac{N_{\max}+1}{2}}\left(2^{-\frac{N_{\max}+1}{2}} + 3C_{\chi^2}\right)\right)^2
$$

$$
\overset{(ii)}{\le} \frac{2r_{\max}^2}{(1-\gamma)^2 T} + \frac{32e\log(|\mathcal{S}||\mathcal{A}|)}{1-\gamma} 16\left(1 + (N_{\max}+1)\left(6C_{\chi^2}^2 + 6\right)\right)\left(r_{\max}^2 + \gamma^2\left(\frac{r_{\max}}{1-\gamma}\right)^2\right)\frac{4\log T}{(1-\gamma)^2 T}
$$

$$
+ 2\frac{2}{1-\gamma}\left(\left(r_{\max} + \frac{r_{\max}}{1-\gamma}\right)\left(1 + 3C_{\chi^2}\right)\right)^2\frac{1}{T}
$$

$$
= \widetilde{\mathcal{O}}\left(\frac{1}{(1-\gamma)^5 T}\right), \tag{114}
$$

where $(i)$ follows from Lemma B.4 and Theorem 4.1. $(ii)$ follows from $2^{\frac{\log T}{\log 2}} \le \frac{1}{T}$.

When $\mathbb{E}\left[\left\|\widehat{Q}_T^{\rho_{\chi^2}(\sigma)} - Q^{*,\rho_{\chi^2}(\sigma)}\right\|_\infty^2\right] \le \epsilon^2$, the iteration $T \ge \widetilde{\mathcal{O}}\left((1-\gamma)^{-5}\epsilon^{-2}\right)$. When $\psi = \frac{1}{2}$, the expected sample size per iteration is

$$
1 + \sum_{n=0}^{N_{max}} 2^{n+1}P_N = 1 + \sum_{n=0}^{N_{max}} 2^{n+1}\frac{1}{2^{n+1}} = N_{\max} + 2 = \frac{2\log T}{\log 2} + 2.
$$

Above all, the total sample complexity is $\widetilde{\mathcal{O}}\left(|\mathcal{S}||\mathcal{A}|(1-\gamma)^{-5}\epsilon^{-2}\right)$.

This completes the proof. $\qquad\square$

# E   KL DIVERGENCE UNCERTAINTY SET

In this section, we provide the proof of Theorems 4.1 and 4.4 specifically for KL distance.

**Theorem E.1** (Restatement of Theorem 4.1 specifically for KL distance). *Consider the case of KL constraint uncertainty set with uncertainty level $\sigma$ i.e. $\mathcal{P}^{KL}(\sigma)$ and $\mathcal{R}^{KL}(\sigma)$, set $\psi = \frac{1}{2}$, for any $Q \in \mathbb{R}^{\mathcal{S}\times\mathcal{A}}, s \in \mathcal{S}, a \in \mathcal{A}$, the estimation bias can be bounded as:*

$$
\sup_{s,a}\left|\mathbb{E}\left[\widehat{\mathcal{T}}_{N_{\max}}^{\rho_{KL}(\sigma)}(Q)(s,a)\right] - \mathcal{T}^{\rho_{KL}(\sigma)}(Q)(s,a)\right| \le \widetilde{\mathcal{O}}\left(2^{-\frac{N_{\max}}{2}}\right),
$$

*and the variation can be bounded as:*

$$
Var\left(\widehat{\mathcal{T}}_{N_{\max}}^{\rho_{KL}(\sigma)}(Q)(s,a)\right) \le \widetilde{\mathcal{O}}\left(N_{\max}\right). \tag{115}
$$

*Proof.* Firstly, we make error decomposition as follows:

$$\sup_{s,a} \left| \mathbb{E}\left[ \widehat{\mathcal{T}}_{N_{\max}}^{\rho_{\chi^2}(\sigma)}(Q)(s,a) \right] - \mathcal{T}^{\rho_{\chi^2}(\sigma)}(Q)(s,a) \right|$$

$$\overset{(i)}{=} \sup_{s,a} \left| \mathbb{E}\left[ g^{*\rho_{KL}(\sigma)}(\hat{\mu}_{s,a,2^{N_{\max}+1}}, r_{s,a}) \right] + \gamma \mathbb{E}\left[ f^{*\rho_{KL}(\sigma)}(\hat{p}_{s,a,2^{N_{\max}+1}}, V) \right] \right.$$

$$\left. - g^{*\rho_{KL}(\sigma)}(\mu_{s,a}, r_{s,a}) - \gamma f^{*\rho_{KL}(\sigma)}(p_{s,a}, V) \right|$$

$$\leq \sup_{s,a} \left| \mathbb{E}\left[ g^{*\rho_{KL}(\sigma)}(\hat{\mu}_{s,a,2^{N_{\max}+1}}, r_{s,a}) \right] - g^{*\rho_{KL}(\sigma)}(\mu_{s,a}, r_{s,a}) \right|$$

$$+ \gamma \sup_{s,a} \left| \mathbb{E}\left[ f^{*\rho_{KL}(\sigma)}(\hat{p}_{s,a,2^{N_{\max}+1}}, V) \right] - f^{*\rho_{KL}(\sigma)}(p_{s,a}, V) \right|$$

$$\leq \mathbb{E}\left[ \sup_{s,a} \left| g^{*\rho_{KL}(\sigma)}(\hat{\mu}_{s,a,2^{N_{\max}+1}}, r_{s,a}) - g^{*\rho_{KL}(\sigma)}(\mu_{s,a}, r_{s,a}) \right| \right]$$

$$+ \gamma \mathbb{E}\left[ \sup_{s,a} \left| f^{*\rho_{KL}(\sigma)}(\hat{p}_{s,a,2^{N_{\max}+1}}, V) - f^{*\rho_{KL}(\sigma)}(p_{s,a}, V) \right| \right], \tag{116}$$

where $(i)$ follows from Proposition B.2.

Then, for convenience, we bound the second term in Equation (116). The first term can be bounded similarly. By Section 2.3,

$$\left| f^{*\rho_{KL}(\sigma)}(p_{s,a}, V) - f^{*\rho_{KL}(\sigma)}(\hat{p}_{s,a,2^{N_{\max}+1}}, V^*) \right|$$

$$= \left| \max_{\alpha \geq 0} \left\{ -\alpha \log\left( \mathbb{E}_{p_{s,a}}\left[ exp\left( -\frac{V(s'_{s,a})}{\alpha} \right) \right] \right) - \alpha\sigma \right\} \right.$$

$$\left. - \max_{\alpha \geq 0} \left\{ -\alpha \log\left( \mathbb{E}_{\hat{p}_{s,a,2^{N_{\max}+1}}}\left[ exp\left( -\frac{V(s'_{s,a})}{\alpha} \right) \right] \right) - \alpha\sigma \right\} \right|$$

$$\overset{(i)}{\leq} \max_{0 \leq \alpha \leq \frac{r_{\max}}{(1-\gamma)\sigma}} \left| -\alpha \log\left( \mathbb{E}_{p_{s,a}}\left[ exp\left( -\frac{V(s'_{s,a})}{\alpha} \right) \right] \right) \right.$$

$$\left. + \alpha \log\left( \mathbb{E}_{\hat{p}_{s,a,2^{N_{\max}+1}}}\left[ exp\left( -\frac{V(s'_{s,a})}{\alpha} \right) \right] \right) \right|$$

$$\leq \max_{0 \leq \alpha \leq \frac{r_{\max}}{(1-\gamma)\sigma}} \left| \alpha \log\left( \frac{\mathbb{E}_{\hat{p}_{s,a,2^{N_{\max}+1}}}\left[ exp\left( -\frac{V(s'_{s,a})}{\alpha} \right) \right]}{\mathbb{E}_{p_{s,a}}\left[ exp\left( -\frac{V(s'_{s,a})}{\alpha} \right) \right]} \right) \right|$$

$$\leq \max_{0 \leq \alpha \leq \frac{r_{\max}}{(1-\gamma)\sigma}} \left| \alpha \log\left( \frac{\mathbb{E}_{\hat{p}_{s,a,2^{N_{\max}+1}}}\left[ exp\left( -\frac{V(s'_{s,a})}{\alpha} \right) \right] - \mathbb{E}_{p_{s,a}}\left[ exp\left( -\frac{V(s'_{s,a})}{\alpha} \right) \right]}{\mathbb{E}_{p_{s,a}}\left[ exp\left( -\frac{V(s'_{s,a})}{\alpha} \right) \right]} + 1 \right) \right|, \tag{117}$$

where $(i)$ follows from $\alpha^{*\rho_{KL}(\sigma)}(p,V) \leq \frac{\max_{s:p(s)\neq 0} V(s)}{\sigma} \leq \frac{\max_{s,a} Q(s,a)}{\sigma} \leq \frac{r_{\max}}{(1-\gamma)\sigma}$ by [Hu and Hong, 2013].

Noting that $\hat{p}_{s,a,2^{N_{\max}+1}}(s'_{s,a})$ is absolutely continuous on $p_{s,a}(s'_{s,a})$, then by Hoeffding's inequality we have

$$\mathbb{P}\left( \max_{s'_{s,a}} \left| \frac{\hat{p}_{s,a,2^{N_{\max}+1}}(s'_{s,a}) - p_{s,a}(s'_{s,a})}{p_{s,a}(s'_{s,a})} \right| \geq \sqrt{\frac{1}{2^{N_{\max}+1}p_{\wedge}^2} \log \frac{2|\mathcal{S}|}{\tau}} \right) \leq \tau. \tag{118}$$

Set $\tau = \frac{2^{-(N_{\max}+1)}}{|\mathcal{S}||\mathcal{A}|}$. With probability at least $1 - \frac{2^{-(N_{\max}+1)}}{|\mathcal{S}||\mathcal{A}|}$, we have that

$$\left| \frac{\mathbb{E}_{\hat{p}_{s,a,2^{N_{\max}+1}}}\left[ exp\left( -\frac{V(s'_{s,a})}{\alpha} \right) \right] - \mathbb{E}_{p_{s,a}}\left[ exp\left( -\frac{V(s'_{s,a})}{\alpha} \right) \right]}{\mathbb{E}_{p_{s,a}}\left[ exp\left( -\frac{V(s'_{s,a})}{\alpha} \right) \right]} \right|$$

$$\overset{(i)}{\le} \max_{s'_{s,a}} \left| \frac{\hat{p}_{s,a,2^{N_{\max}+1}}(s'_{s,a}) - p_{s,a}(s'_{s,a})}{p_{s,a}(s'_{s,a})} \right|$$

$$\le \sqrt{\frac{N_{\max}}{2^{N_{\max}+1}p_\wedge^2} \log\left(2|\mathcal{S}|^2|\mathcal{A}|\right)}, \tag{119}$$

where $(i)$ follows from the fact that

$$\left| \sum p_i x_i \right| = \left| \sum \frac{p_i}{q_i} q_i x_i \right| \le \left| \sum q_i x_i \right| \max_i \left| \frac{p_i}{q_i} \right|. \tag{120}$$

Note that if we set $\frac{N_{\max}}{p_\wedge^2} \log\left(2|\mathcal{S}|^2|\mathcal{A}|\right) \le \frac{1}{4} 2^{N_{\max}+1}$, it holds that $\sqrt{\frac{N_{\max}}{2^{N_{\max}+1}p_\wedge^2} \log\left(2|\mathcal{S}|^2|\mathcal{A}|\right)} \le \frac{1}{2}$. Then, combined with Equation (117), we can conclude that

$$\left| f^{*\rho_{KL}(\sigma)}\left(p_{s,a}, V(s'_{s,a})\right) - f^{*\rho_{KL}(\sigma)}\left(\hat{p}_{s,a,2^{N_{\max}+1}}, V^*(s'_{s,a})\right) \right|$$

$$\le \frac{r_{\max}}{(1-\gamma)\sigma} \max_{0 \le \alpha \le \frac{r_{\max}}{(1-\gamma)\sigma}} \left| \log\left( \frac{\mathbb{E}_{\hat{p}_{s,a,2^{N_{\max}+1}}}\left[exp\left(-\frac{V}{\alpha}\right)\right] - \mathbb{E}_{p_{s,a}}\left[exp\left(-\frac{V}{\alpha}\right)\right]}{\mathbb{E}_{p_{s,a}}\left[exp\left(-\frac{V(s'_{s,a})}{\alpha}\right)\right]} + 1 \right) \right|$$

$$\overset{(i)}{\le} \frac{r_{\max}}{(1-\gamma)\sigma} \max_{0 \le \alpha \le \frac{r_{\max}}{(1-\gamma)\sigma}} 2 \left| \frac{\mathbb{E}_{\hat{p}_{s,a,2^{N_{\max}+1}}}\left[exp\left(-\frac{V(s'_{s,a})}{\alpha}\right)\right] - \mathbb{E}_{p_{s,a}}\left[exp\left(-\frac{V(s'_{s,a})}{\alpha}\right)\right]}{\mathbb{E}_{p_{s,a}}\left[exp\left(-\frac{V(s'_{s,a})}{\alpha}\right)\right]} \right|$$

$$\le \frac{2r_{\max}}{(1-\gamma)\sigma} \max_{s'_{s,a}} \left| \frac{\hat{p}_{s,a,2^{N_{\max}+1}}(s'_{s,a}) - p_{s,a}(s'_{s,a})}{p_{s,a}(s'_{s,a})} \right| \le \frac{2r_{\max}}{(1-\gamma)\sigma} \sqrt{\frac{N_{\max}}{2^{N_{\max}+1}p_\wedge^2} \log\left(2|\mathcal{S}|^2|\mathcal{A}|\right)}, \tag{121}$$

where $(i)$ follows from that $|\log(x+1)| \le 2|x|$ for $|x| \le \frac{1}{2}$. Then, according to the Bernoulli's inequality, we have that

$$\left(1 - \frac{2^{-N_{\max}-1}}{|\mathcal{S}||\mathcal{A}|}\right)^{|\mathcal{S}||\mathcal{A}|} \ge 1 - 2^{-N_{\max}-1}. \tag{122}$$

Therefore, with probability at least $1 - 2^{-N_{\max}-1}$, there exists

$$\sup_{s,a} \left| f^{*\rho_{KL}(\sigma)}(\hat{p}_{s,a,2^{N_{\max}+1}}, V) - f^{*\rho_{KL}(\sigma)}(p_{s,a}, V) \right| \le \frac{2r_{\max}}{(1-\gamma)\sigma} \sqrt{\frac{N_{\max}}{2^{N_{\max}+1}p_\wedge^2} \log\left(2|\mathcal{S}|^2|\mathcal{A}|\right)}. \tag{123}$$

Otherwise, with probability at most $2^{-(N_{\max}+1)}$, we can conclude that

$$\sup_{s,a} \left| f^{*\rho_{KL}(\sigma)}\left(p_{s,a}, V\right) - f^{*\rho_{KL}(\sigma)}\left(\hat{p}_{s,a,2^{N_{\max}+1}}, V^*\right) \right| \le \max_{s'_{s,a}} V(s'_{s,a}) \le \frac{r_{\max}}{1-\gamma}. \tag{124}$$

Then, consider the expectation, we can get

$$\mathbb{E}\left[ \sup_{s,a} \left| f^{*\rho_{KL}(\sigma)}\left(p_{s,a}, V\right) - f^{*\rho_{KL}(\sigma)}\left(\hat{p}_{s,a,2^{N_{\max}+1}}, V^*\right) \right| \right]$$

$$\le \frac{2r_{\max}}{(1-\gamma)\sigma} \sqrt{\frac{N_{\max}}{2^{N_{\max}+1}p_\wedge^2} \log\left(2|\mathcal{S}|^2|\mathcal{A}|\right)} + 2^{-(N_{\max}+1)} \frac{r_{\max}}{1-\gamma}$$

$$\le \frac{2r_{\max}}{(1-\gamma)\sigma} \sqrt{N_{\max} \log\left(2|\mathcal{S}|^2|\mathcal{A}|\right)} \frac{1}{p_\wedge 2^{\frac{N_{\max}+1}{2}}} + 2^{-(N_{\max}+1)} \frac{r_{\max}}{1-\gamma}, \tag{125}$$

where we set $C_{KL} = 2\sqrt{N_{\max} \log\left(2|\mathcal{S}|^2|\mathcal{A}|\right)}$, then

$$\mathbb{E}\left[ \sup_{s,a} \left| f^{*\rho_{KL}(\sigma)}\left(p_{s,a}, V\right) - f^{*\rho_{KL}(\sigma)}\left(\hat{p}_{s,a,2^{N_{\max}+1}}, V^*\right) \right| \right]$$

$$\le \frac{r_{\max}}{(1-\gamma)\sigma} C_{KL} \frac{1}{p_\wedge 2^{\frac{N_{\max}+1}{2}}} + \frac{r_{\max}}{1-\gamma} 2^{-(N_{\max}+1)}. \tag{126}$$

Similarly, we can get the bound

$$\mathbb{E}\left[\sup_{s,a}\left|g^{*\rho_{KL}(\sigma)}(\hat{\mu}_{s,a,2^{N_{\max}+1}}, r_{s,a}) - g^{*\rho_{KL}(\sigma)}(\mu_{s,a}, r_{s,a})\right|\right]$$
$$\leq \frac{r_{\max}C_{KL}}{\sigma p_\wedge}2^{-\frac{N_{\max}+1}{2}} + r_{\max}2^{-(N_{\max}+1)}. \tag{127}$$

Thus, we can get that

$$\sup_{s,a}\left|\mathbb{E}\left[\widehat{\mathcal{T}}_{N_{\max}}^{\rho_{KL}(\sigma)}(Q)(s,a)\right] - \mathcal{T}^{\rho_{KL}(\sigma)}(Q)(s,a)\right|$$

$$\leq \mathbb{E}\left[\sup_{s,a}\left|g^{*\rho_{KL}(\sigma)}(\hat{\mu}_{s,a,2^{N_{\max}+1}}, r_{s,a}) - g^{*\rho_{KL}(\sigma)}(\mu_{s,a}, r_{s,a})\right|\right]$$
$$+ \gamma\mathbb{E}\left[\sup_{s,a}\left|f^{*\rho_{KL}(\sigma)}(\hat{p}_{s,a,2^{N_{\max}+1}}, V) - f^{*\rho_{KL}(\sigma)}(p_{s,a}, V)\right|\right]$$
$$\leq \left(\frac{\gamma r_{\max}}{1-\gamma} + r_{\max}\right)2^{-\frac{N_{\max}+1}{2}}\left(2^{-\frac{N_{\max}+1}{2}} + \frac{C_{KL}}{\sigma p_\wedge}\right). \tag{128}$$

**Variance:** Next, we consider the variance of the robust Bellman operator. Firstly, we make error decomposition of the robust Bellman operator variance.

$$\mathrm{Var}\left(\widehat{\mathcal{T}}_{N_{\max}}^{\rho_{KL}(\sigma)}(Q)(s,a)\right) = \mathrm{Var}\left(\widehat{r}^{\rho_{KL}(\sigma)} + \gamma\widehat{v}^{\rho_{KL}(\sigma)}(Q)(s,a)\right)$$
$$= \mathrm{Var}\left(\widehat{r}^{\rho_{KL}(\sigma)}\right) + \gamma^2\mathrm{Var}\left(\widehat{v}^{\rho_{KL}(\sigma)}(Q)(s,a)\right). \tag{129}$$

For convenience, we analyze the second term in the above equation. The first term can be bounded similarly.

$$\mathrm{Var}\left(\widehat{v}^{\rho_{KL}(\sigma)}(Q)(s,a)\right) = \mathbb{E}\left[\left(\widehat{v}^{\rho_{KL}(\sigma)}(Q)(s,a)\right)^2\right] - \left(\mathbb{E}\left[\widehat{v}^{\rho_{KL}(\sigma)}(Q)(s,a)\right]\right)^2$$
$$\leq \mathbb{E}\left[\left(\widehat{v}^{\rho_{KL}(\sigma)}(Q)(s,a)\right)^2\right]. \tag{130}$$

Next, according to the Equations (13) and (14), now we compute the expectation of $N_2$ and write a detailed explanation of the variance as follows:

$$\mathbb{E}\left[\left(\widehat{v}^{\rho_{KL}(\sigma)}(Q)(s,a)\right)^2\right] = \mathbb{E}\left[\left(V(s'_{s,a,0}) + \frac{\delta_{s,a,N_2}^{\rho_{KL}(\sigma)}(Q)(s,a)}{P_{N_2}}\right)^2\right]$$

$$\leq 2\mathbb{E}\left[V(s'_{s,a,0})^2\right] + 2\mathbb{E}\left[\left(\frac{\delta_{s,a,N_2}^{\rho_{KL}(\sigma)}(Q)(s,a)}{P_{N_2}}\right)^2\right]$$

$$\leq \frac{2r_{\max}^2}{(1-\gamma)^2\sigma^2} + 2\sum_{N=0}^{N_{\max}}\mathbb{E}\left[\left(\frac{\delta_{s,a,N_2}^{\rho_{KL}(\sigma)}(Q)(s,a)}{P_{N_2}}\Big|N_2=N\right)^2\right]P_N$$

$$\leq \frac{2r_{\max}^2}{(1-\gamma)^2\sigma^2} + 2\sum_{N=0}^{N_{\max}}\frac{\mathbb{E}\left[(\delta_{s,a,N}^{\rho_{KL}(\sigma)}(Q)(s,a))^2\right]}{P_N}. \tag{131}$$

Next, we bound the term $\left|\delta_{s,a,N}^{\rho_{KL}(\sigma)}(Q)(s,a)\right|$,

$$\left|\delta_{s,a,N}^{\rho_{KL}(\sigma)}(Q)(s,a)\right| = \left|\sup_{\alpha\geq 0}\left\{f^{\rho_{KL}(\sigma)}(\widehat{p}_{s,a,2^{N+1}}, \alpha, V)\right\}\right.$$

$$\left. - \frac{1}{2}\sup_{\alpha\geq 0}\left\{f^{\rho_{KL}(\sigma)}(\widehat{p}_{s,a,2^N}^E, \alpha, V)\right\} - \frac{1}{2}\sup_{\alpha\geq 0}\left\{f^{\rho_{KL}(\sigma)}(\widehat{p}_{s,a,2^N}^O, \alpha, V)\right\}\right|. \tag{132}$$

Then, we make an error decomposition as follows:

$$
\begin{aligned}
\left|\delta_{s,a,N}^{\rho_{KL}(\sigma)}(Q)(s,a)\right|^2 &= \left| \sup_{\alpha \geq 0} \left\{ f^{\rho_{KL}(\sigma)}(\widehat{p}_{s,a,2^{N+1}}, \alpha, V) \right\} \right. \\
&\quad \left. - \frac{1}{2} \sup_{\alpha \geq 0} \left\{ f^{\rho_{KL}(\sigma)}(\widehat{p}_{s,a,2^N}^E, \alpha, V) \right\} - \frac{1}{2} \sup_{\alpha \geq 0} \left\{ f^{\rho_{KL}(\sigma)}(\widehat{p}_{s,a,2^N}^O, \alpha, V) \right\} \right|^2 \\
&\leq 3 \left| \sup_{\alpha \geq 0} \left\{ f^{\rho_{KL}(\sigma)}(\widehat{p}_{s,a,2^{N+1}}, \alpha, V) \right\} - \sup_{\alpha \geq 0} \left\{ f^{\rho_{KL}(\sigma)}(p_{s,a}, \alpha, V) \right\} \right|^2 \\
&\quad + \frac{3}{4} \left| \sup_{\alpha \geq 0} \left\{ f^{\rho_{KL}(\sigma)}(\widehat{p}_{s,a,2^N}^E, \alpha, V) \right\} - \sup_{\alpha \geq 0} \left\{ f^{\rho_{KL}(\sigma)}(\widehat{p}_{s,a}, \alpha, V) \right\} \right|^2 \\
&\quad + \frac{3}{4} \left| \sup_{\alpha \geq 0} \left\{ f^{\rho_{KL}(\sigma)}(\widehat{p}_{s,a,2^N}^O, \alpha, V) \right\} - \sup_{\alpha \geq 0} \left\{ f^{\rho_{KL}(\sigma)}(p_{s,a}, \alpha, V) \right\} \right|^2 .
\end{aligned}
\tag{133}
$$

**Case 1:** Combined with the analysis in Equations (121) and (124), we can conclude that when $N \leq \frac{\log(1+p_\wedge^2 \log(2|\mathcal{S}|^2|\mathcal{A}|)\log T)}{\log 2}$, we bound the term $\frac{\mathbb{E}\left[\left|\delta_{s,a,N}^{\rho_{KL}(\sigma)}(Q)(s,a)\right|^2\right]}{P_N}$ as follows,

$$
\left|\delta_{s,a,N}^{\rho_{KL}(\sigma)}(Q)(s,a)\right|^2 \leq \left(\frac{r_{\max}}{1-\gamma}\right)^2, \qquad \frac{1}{P_N} = 2^N \leq 1 + p_\wedge^{-2} \log(2|\mathcal{S}|^2|\mathcal{A}|)\log T.
\tag{134}
$$

Hence, we have that

$$
\frac{\mathbb{E}\left[\left|\delta_{s,a,N}^{\rho_{KL}(\sigma)}(Q)(s,a)\right|^2\right]}{P_N} \leq \left(\frac{r_{\max}}{1-\gamma}\right)^2 \left(1 + p_\wedge^{-2} \log(2|\mathcal{S}|^2|\mathcal{A}|)\log T\right).
\tag{135}
$$

**Case 2:** When $N > \frac{\log(1+p_\wedge^2 \log(2|\mathcal{S}|^2|\mathcal{A}|)\log T)}{\log 2}$, consider the fact $\mathbb{P}(A \cap B \cap C) \geq 1 - \mathbb{P}(\neg A) - \mathbb{P}(\neg B) - \mathbb{P}(\neg C)$, by Equation (121), with probability at least $1 - 3 * 2^{-N}$

$$
\begin{aligned}
\left|\delta_{s,a,N}^{\rho_{KL}(\sigma)}(Q)(s,a)\right|^2 &\leq 3 \left(C_{KL} \frac{r_{\max}}{p_\wedge(1-\gamma)\sigma} 2^{-\frac{N+1}{2}}\right)^2 \\
&\quad + \frac{3}{4} \left(C_{KL} \frac{r_{\max}}{p_\wedge(1-\gamma)\sigma} 2^{-\frac{N}{2}}\right)^2 + \frac{3}{4} \left(C_{KL} \frac{r_{\max}}{p_\wedge(1-\gamma)\sigma} 2^{-\frac{N}{2}}\right)^2 \\
&= 3 \frac{C_{KL}^2 r_{\max}^2}{p_\wedge^2(1-\gamma)^2\sigma^2} 2^{-(N+1)},
\end{aligned}
\tag{136}
$$

Since $0 \leq \sup_{\alpha \geq 0} \left\{ f^{\rho_{KL}(\sigma)}(q, \alpha, V) \right\} \leq \frac{r_{\max}}{1-\gamma}$ for any distribution $q$, with probability at most $3 * 2^{-N}$ we have that

$$
\left|\delta_{s,a,N}^{\rho_{KL}(\sigma)}(Q)(s,a)\right|^2 \leq \left(\frac{r_{\max}}{1-\gamma}\right)^2.
\tag{137}
$$

Above all, we can get that

$$
\mathbb{E}\left[\left|\delta_{s,a,N}^{\rho_{KL}(\sigma)}(Q)(s,a)\right|^2\right] \leq 3 \frac{C_{KL}^2 r_{\max}^2}{p_\wedge^2(1-\gamma)^2\sigma^2} 2^{-(N+1)} + \left(\frac{r_{\max}}{1-\gamma}\right)^2 3 * 2^{-N}.
\tag{138}
$$

Combined with **Case 1** and **Case 2**, when $\psi = \frac{1}{2}$, $P_N = 2^{-N-1}$. Then, we have that

$$\mathbb{E}\left[\frac{\left|\delta_{s,a,N}^{\rho_{KL}(\sigma)}(Q)(s,a)\right|^2}{P_N}\right]$$

$$\leq \left(\frac{r_{\max}}{1-\gamma}\right)^2 \left(1 + p_\wedge^{-2}\log(2|\mathcal{S}|^2|\mathcal{A}|)\log T\right) + \frac{3}{2}\frac{C_{KL}^2 r_{\max}^2}{p_\wedge^2(1-\gamma)^2\sigma^2} + 3\left(\frac{r_{\max}}{1-\gamma}\right)^2. \tag{139}$$

Then, by Equation (139), we can get the boundary of variance of the robust Bellman operator as follows:

$$\mathrm{Var}\left(\widehat{v}^{\rho_{KL}(\sigma)}(Q)(s,a)\right)$$

$$\leq \frac{2r_{\max}^2}{(1-\gamma)^2\sigma^2} + 2\sum_{N=0}^{N_{\max}} \frac{\mathbb{E}\left[(\delta_{s,a,N}^{\rho_{KL}(\sigma)}(Q)(s,a))^2\right]}{P_N}$$

$$\leq \frac{2r_{\max}^2}{(1-\gamma)^2\sigma^2} + 4\sum_{N=0}^{N_{\max}} \left(\frac{r_{\max}}{1-\gamma}\right)^2 \left(4 + p_\wedge^{-2}\log(2|\mathcal{S}|^2|\mathcal{A}|)\log T + \frac{3}{2}\frac{C_{KL}^2}{p_\wedge^2\sigma^2}\right)$$

$$\leq \left(2 + 4(N_{\max} + 1)\left(4 + \log(2|\mathcal{S}|^2|\mathcal{A}|\log T)(1-\gamma) + \frac{3C_{KL}^2}{2}\right)\right)\frac{r_{\max}^2}{p_\wedge^2(1-\gamma)^2\sigma^2}, \tag{140}$$

Set $C_{\mathrm{var}} = 2 + 4(N_{\max} + 1)\left(4 + \log(2|\mathcal{S}|\log T)(1-\gamma) + \frac{3C_{KL}^2}{2}\right)$, then

$$\mathrm{Var}\left(\widehat{v}^{\rho_{KL}(\sigma)}(Q)(s,a)\right) \leq C_{\mathrm{var}}\frac{r_{\max}^2}{p_\wedge^2(1-\gamma)^2\sigma^2}.$$

Similarly, we can get the boundary of the variance $\mathrm{Var}\left(\widehat{r}^{\rho_{KL}(\sigma)}\right)$ as follows:

$$\mathrm{Var}\left(\widehat{r}^{\rho_{KL}(\sigma)}\right) \leq C_{\mathrm{var}}\frac{r_{\max}^2}{p_\wedge^2\sigma^2}. \tag{141}$$

Hence, we can get the robust Bellman operator variance bound:

$$\mathrm{Var}\left(\widehat{\mathcal{T}}_{N_{\max}}^{\rho_{KL}(\sigma)}(Q)(s,a)\right) = \mathrm{Var}\left(\widehat{r}^{\rho_{KL}(\sigma)}\right) + \gamma^2\mathrm{Var}\left(\widehat{v}^{\rho_{KL}(\sigma)}(Q)(s,a)\right)$$

$$\leq \frac{C_{\mathrm{var}}}{p_\wedge^2\sigma^2}\left(r_{\max}^2 + \frac{\gamma^2 r_{\max}^2}{(1-\gamma)^2}\right). \tag{142}$$

This completes the proof. $\qquad\square$

**Lemma E.2.** *For any fixed $Q \in \mathbb{R}^{|\mathcal{S}||\mathcal{A}|}, s \in \mathcal{S}, a \in \mathcal{A}$, the infinite norm of robust Bellman operator can be bounded as:*

$$\mathbb{E}\left[\left\|\widehat{\mathcal{T}}_{N_{\max}}^{\rho_{KL}(\sigma)}(Q)(s,a)\right\|_\infty^2\right] \leq \widetilde{\mathcal{O}}\left(N_{\max}\right). \tag{143}$$

*Proof of Lemma E.2.* We then consider the expectation of infinite norm of robust Bellman operator. Set $\psi = \frac{1}{2}$, and then we make an error decomposition as follows

$$\mathbb{E}\left[\left\|\widehat{\mathcal{T}}_{N_{\max}}^{\rho_{KL}(\sigma)}(Q)\right\|_\infty^2\right] \leq 4r_{\max}^2 + 4\gamma^2\frac{r_{\max}^2}{(1-\gamma)^2} + 4\mathbb{E}\left[\sum_{N_1=0}^{N_{\max}}\sup_{s,a}\frac{\left(\delta_{s,a,N_1}^{r,\rho(\sigma)}\right)^2}{2^{-N_1-1}} + \gamma^2\sum_{N_2=0}^{N_{\max}}\sup_{s,a}\frac{\left(\delta_{s,a,N_2}^{\rho_{KL}(\sigma)}(Q)\right)^2}{2^{-N_2-1}}\right]. \tag{144}$$

For convenience, we analyze the last term in the above equation. Consider the term $\sup_{s,a} \left| \delta_{s,a,N}^{\rho_{KL}(\sigma)}(Q)(s,a) \right|^2$, we make an error decomposition as follows:

$$\sup_{s,a} \left| \delta_{s,a,N}^{\rho_{KL}(\sigma)}(Q)(s,a) \right|^2 = \sup_{s,a} \left| \sup_{\alpha \geq 0} \left\{ f^{\rho_{KL}(\sigma)}(\widehat{p}_{s,a,2^{N+1}}, \alpha, V) \right\} \right.$$

$$\left. - \frac{1}{2} \sup_{\alpha \geq 0} \left\{ f^{\rho_{KL}(\sigma)}(\widehat{p}_{s,a,2^N}^E, \alpha, V) \right\} - \frac{1}{2} \sup_{\alpha \geq 0} \left\{ f^{\rho_{KL}(\sigma)}(\widehat{p}_{s,a,2^N}^O, \alpha, V) \right\} \right|^2$$

$$\leq 3 \sup_{s,a} \left| \sup_{\alpha \geq 0} \left\{ f^{\rho_{KL}(\sigma)}(\widehat{p}_{s,a,2^{N+1}}, \alpha, V) \right\} - \sup_{\alpha \geq 0} \left\{ f^{\rho_{KL}(\sigma)}(p_{s,a}, \alpha, V) \right\} \right|^2$$

$$+ \frac{3}{4} \sup_{s,a} \left| \sup_{\alpha \geq 0} \left\{ f^{\rho_{KL}(\sigma)}(\widehat{p}_{s,a,2^N}^E, \alpha, V) \right\} - \sup_{\alpha \geq 0} \left\{ f^{\rho_{KL}(\sigma)}(p_{s,a}, \alpha, V) \right\} \right|^2$$

$$+ \frac{3}{4} \sup_{s,a} \left| \sup_{\alpha \geq 0} \left\{ f^{\rho_{KL}(\sigma)}(\widehat{p}_{s,a,2^N}^O, \alpha, V) \right\} - \sup_{\alpha \geq 0} \left\{ f^{\rho_{KL}(\sigma)}(p_{s,a}, \alpha, V) \right\} \right|^2. \tag{145}$$

**Case 1:** When $N \leq \frac{\log(1 + p_\wedge^2 \log(2|\mathcal{S}|^2|\mathcal{A}|) \log T)}{\log 2}$, we can get

$$\sup_{s,a} \left| \delta_{s,a,N}^{\rho_{KL}(\sigma)}(Q)(s,a) \right|^2 \leq \left( \frac{r_{\max}}{1-\gamma} \right)^2, \frac{1}{P_N} = 2^N \leq 1 + \frac{\log(2|\mathcal{S}|^2|\mathcal{A}|) \log T}{p_\wedge^2}. \tag{146}$$

Therefore, we have that

$$\frac{\sup_{s,a} \left| \delta_{s,a,N}^{\rho_{KL}(\sigma)}(Q)(s,a) \right|^2}{P_N} \leq \left( \frac{r_{\max}}{1-\gamma} \right)^2 \left( 1 + \frac{\log(2|\mathcal{S}|^2|\mathcal{A}|) \log T}{p_\wedge^2} \right). \tag{147}$$

**Case 2:** When $N > \frac{\log(1 + p_\wedge^2 \log(2|\mathcal{S}|^2|\mathcal{A}|) \log T)}{\log 2}$, combined with the analysis in Equations (122) and (123) and the fact $\mathbb{P}(A \cap B \cap C) \geq 1 - \mathbb{P}(\neg A) - \mathbb{P}(\neg B) - \mathbb{P}(\neg C)$, we can conclude that for any $N \geq 0$, with probability at least $1 - 3 * 2^{-N}$

$$\sup_{s,a} \left| \delta_{s,a,N}^{\rho_{KL}(\sigma)}(Q)(s,a) \right|^2 \leq 3 \left( \frac{C_{KL} r_{\max}}{\sigma(1-\gamma)p_\wedge} 2^{-\frac{N+1}{2}} \right)^2 + \frac{3}{4} \left( \frac{C_{KL} r_{\max}}{\sigma(1-\gamma)p_\wedge} 2^{-\frac{N}{2}} \right)^2 + \frac{3}{4} \left( \frac{C_{KL} r_{\max}}{\sigma(1-\gamma)p_\wedge} 2^{-\frac{N}{2}} \right)^2$$

$$\leq 6 \frac{C_{KL}^2 r_{\max}^2}{\sigma^2(1-\gamma)^2 p_\wedge^2} 2^{-(N+1)}, \tag{148}$$

Since $0 \leq \sup_{\alpha \geq 0} \left\{ f^{\rho_{KL}(\sigma)}(q, \alpha, V) \right\} \leq \frac{r_{\max}}{1-\gamma}$ for any distribution $q$, with probability at most $3 * 2^{-N}$ we have that

$$\sup_{s,a} \left| \delta_{s,a,N}^{\rho_{KL}(\sigma)}(Q)(s,a) \right|^2 \leq \left( \frac{r_{\max}}{1-\gamma} \right)^2. \tag{149}$$

Above all, we can get that

$$\mathbb{E} \left[ \sup_{s,a} \left| \delta_{s,a,N}^{\rho_{KL}(\sigma)}(Q)(s,a) \right|^2 \right] \leq 6 \frac{C_{KL}^2 r_{\max}^2}{\sigma^2(1-\gamma)^2 p_\wedge^2} 2^{-(N+1)} + \left( \frac{r_{\max}}{1-\gamma} \right)^2 3 * 2^{-N}$$

$$\leq 2 \left( 3 \frac{C_{KL}^2}{\sigma^2 p_\wedge^2} + 3 \right) \left( \frac{r_{\max}}{1-\gamma} \right)^2 2^{-N}. \tag{150}$$

Combined with **Case 1** and **Case 2**, we can get

$$\mathbb{E} \left[ \sup_{s,a} \frac{\left| \delta_{s,a,N}^{\rho_{KL}(\sigma)}(Q)(s,a) \right|^2}{P_N} \right] = \left( 6 \frac{C_{KL}^2}{\sigma^2 p_\wedge^2} + 7 + \frac{\log(2|\mathcal{S}|^2|\mathcal{A}|) \log T}{p_\wedge^2} \right) \left( \frac{r_{\max}}{1-\gamma} \right)^2. \tag{151}$$

Then, plug Equation (151) in Equation (144), we can get the bound of expectation of infinite norm as follows:

$$\mathbb{E}\left[\sum_{N_2=0}^{N_{\max}}\sup_{s,a}\frac{\left(\delta_{s,a,N_2}^{\rho_{KL}(\sigma)}(Q)\right)^2}{2^{-N_2-1}}\right] \leq \sum_{N_2=0}^{N_{\max}}\left(6\frac{C_{KL}^2}{\sigma^2 p_\wedge^2}+7+\frac{\log(2|\mathcal{S}|^2|\mathcal{A}|)\log T}{p_\wedge^2}\right)\left(\frac{r_{\max}}{1-\gamma}\right)^2$$

$$=(N_{\max}+1)\left(6\frac{C_{KL}^2}{\sigma^2 p_\wedge^2}+7+\frac{\log(2|\mathcal{S}|^2|\mathcal{A}|)\log T}{p_\wedge^2}\right)\left(\frac{r_{\max}}{1-\gamma}\right)^2. \tag{152}$$

Similarly, we can get the bound as follows:

$$\mathbb{E}\left[\sum_{N_1=0}^{N_{\max}}\sup_{s,a}\frac{\left(\delta_{s,a,N_1}^{r,\rho(\sigma)}\right)^2}{2^{-N_1-1}}\right] \leq \sum_{N_1=0}^{N_{\max}}\left(6\frac{C_{KL}^2}{\sigma^2 p_\wedge^2}+7+\frac{\log(2|\mathcal{S}|^2|\mathcal{A}|)\log T}{p_\wedge^2}\right)r_{\max}^2$$

$$=(N_{\max}+1)\left(6\frac{C_{KL}^2}{\sigma^2 p_\wedge^2}+7+\frac{\log(2|\mathcal{S}|^2|\mathcal{A}|)\log T}{p_\wedge^2}\right)r_{\max}^2. \tag{153}$$

Hence, combining Equation (144) and the above equations, we can get the robust Bellman operator infinite norm bound:

$$\mathbb{E}\left[\left\|\widehat{\mathcal{T}}_{N_{\max}}^{\rho_{KL}(\sigma)}(Q)\right\|_\infty^2\right] \leq 4\left(1+(N_{\max}+1)\left(6\frac{C_{KL}^2}{\sigma^2 p_\wedge^2}+7+\frac{\log(2|\mathcal{S}|^2|\mathcal{A}|)\log T}{p_\wedge^2}\right)\right)\left(r_{\max}^2+\gamma^2\left(\frac{r_{\max}}{1-\gamma}\right)^2\right). \tag{154}$$

This completes the proof. $\square$

**Theorem E.3** (Restatement of Theorem 4.4). *If we set $\psi=\frac{1}{2}$ and the stepsize as*

$$\beta_t = \beta = \frac{\log T}{(1-\gamma)T}.$$

*Then the output of Algorithm 1 satisfies that:*

$$\mathbb{E}\left[\left\|\widehat{Q}_T^{\rho_{KL}(\sigma)}-Q^{*\rho_{KL}(\sigma)}\right\|_\infty^2\right] \leq \widetilde{\mathcal{O}}\left(\frac{1}{p_\wedge^2(1-\gamma)^5 T}\right).$$

*To ensure*

$$\mathbb{E}\left[\left\|\widehat{Q}_T^{\rho_{KL}(\sigma)}-Q^{*\rho_{KL}(\sigma)}\right\|_\infty^2\right] \leq \epsilon^2,$$

*the expected total sample complexity $N^{\rho_{KL}(\sigma)}(\epsilon)$ is*

$$N^{\rho_{KL}(\sigma)}(\epsilon) = |\mathcal{S}||\mathcal{A}|N_{\max}T \geq \widetilde{\mathcal{O}}\left(\frac{|\mathcal{S}||\mathcal{A}|}{p_\wedge^2(1-\gamma)^5\epsilon^2}\right).$$

*Proof.* We consider the stochastic iteration that

$$\widehat{Q}_{t+1}^{\rho_{KL}(\sigma)} = \widehat{Q}_t^{\rho_{KL}(\sigma)} + \beta_t\left(\bar{\mathcal{T}}_{N_{\max}}^{\rho_{KL}(\sigma)}\left(Q_t^{\rho_{KL}(\sigma)}\right)-\widehat{Q}_t^{\rho_{KL}(\sigma)}+W_t\right), \tag{155}$$

where $W_t = \widehat{\mathcal{T}}_{N_{\max}}^{\rho_{KL}(\sigma)}\left(Q_t^{\rho_{KL}(\sigma)}\right)-\bar{\mathcal{T}}_{N_{\max}}^{\rho_{KL}(\sigma)}\left(Q_t^{\rho_{KL}(\sigma)}\right)$.

Define the filtration $\mathcal{F}_t = \left\{Q_0^{\rho_{KL}(\sigma)},W_0,...,Q_{t-1}^{\rho_{KL}(\sigma)},W_{t-1},Q_t^{\rho_{KL}(\sigma)}\right\}$. Then, by Theorem 4.4, we can get that

$$\mathbb{E}\left[W_t|\mathcal{F}_t\right] = 0, \tag{156}$$

and by Lemma E.2, we can get that

$$
\begin{aligned}
\mathbb{E}\left[\|W_t\|_\infty^2 \mid \mathcal{F}_t\right] &\leq \mathbb{E}\left[\sup_{s,a}\left|\widehat{\mathcal{T}}_{N_{\max}}^{\rho_{KL}(\sigma)}\left(Q_t^{\rho_{KL}(\sigma)}\right)(s,a) - \bar{\boldsymbol{\mathcal{T}}}_{N_{\max}}^{\rho_{KL}(\sigma)}\left(Q_t^{\rho_{KL}(\sigma)}\right)(s,a)\right|^2 \mid \mathcal{F}_t\right] \\
&\leq 2\mathbb{E}\left[\sup_{s,a}\left|\widehat{\mathcal{T}}_{N_{\max}}^{\rho_{KL}(\sigma)}\left(Q_t^{\rho_{KL}(\sigma)}\right)(s,a)\right|^2 + \sup_{s,a}\left|\bar{\boldsymbol{\mathcal{T}}}_{N_{\max}}^{\rho_{KL}(\sigma)}\left(Q_t^{\rho_{KL}(\sigma)}\right)(s,a)\right|^2 \mid \mathcal{F}_t\right] \\
&\overset{(a)}{\leq} 4\mathbb{E}\left[\sup_{s,a} 2\left|\widehat{\mathcal{T}}_{N_{\max}}^{\rho_{KL}(\sigma)}\left(Q_t^{\rho_{KL}(\sigma)}\right)(s,a)\right|^2 \mid \mathcal{F}_t\right] \\
&\overset{(b)}{\leq} 16\left(1 + (N_{\max}+1)\left(6\frac{C_{KL}^2}{\sigma^2 p_\wedge^2} + 7 + \frac{\log(2|\mathcal{S}|^2|\mathcal{A}|)\log T}{p_\wedge^2}\right)\right)\left(r_{\max}^2 + \gamma^2\left(\frac{r_{\max}}{1-\gamma}\right)^2\right), \quad (157)
\end{aligned}
$$

where $(a)$ follows from that

$$
\begin{aligned}
\mathbb{E}\left[\sup_{s,a}\left|\bar{\boldsymbol{\mathcal{T}}}_{N_{\max}}^{\rho_{KL}(\sigma)}\left(Q_t^{\rho_{KL}(\sigma)}\right)(s,a)\right|^2 \mid \mathcal{F}_t\right] &= \mathbb{E}\left[\sup_{s,a}\left|\mathbb{E}\left[\widehat{\mathcal{T}}_{N_{\max}}^{\rho_{KL}(\sigma)}\left(Q_t^{\rho_{KL}(\sigma)}\right)(s,a)\right]\right|^2 \mid \mathcal{F}_t\right] \\
&\leq \mathbb{E}\left[\sup_{s,a}\mathbb{E}\left[\left|\widehat{\mathcal{T}}_{N_{\max}}^{\rho_{KL}(\sigma)}\left(Q_t^{\rho_{KL}(\sigma)}\right)(s,a)\right|^2\right] \mid \mathcal{F}_t\right] \\
&\leq \mathbb{E}\left[\sup_{s,a}\left|\widehat{\mathcal{T}}_{N_{\max}}^{\rho_{KL}(\sigma)}\left(Q_t^{\rho_{KL}(\sigma)}\right)(s,a)\right|^2 \mid \mathcal{F}_t\right], \quad (158)
\end{aligned}
$$

and $(b)$ follows from Lemma E.2.

According to Equation (23), we have that

$$
\widehat{Q}^{*\rho_{KL}(\sigma)}(s,a) = \bar{\boldsymbol{\mathcal{T}}}_{N_{\max}}^{\rho(\sigma)}(\widehat{Q}^{*\rho_{KL}(\sigma)})(s,a) = \mathbb{E}\left[\widehat{\mathcal{T}}_{N_{\max}}^{\rho(\sigma)}(\widehat{Q}^{*\rho_{KL}(\sigma)})(s,a)\right].
$$

Then, apply Lemma B.5 [Chen et al., 2020], set the constant stepsize $\beta_t = \beta = \frac{2\log T}{(1-\gamma)T}$ and $T$ large enough s.t.

$$
\beta = \frac{2\log T}{(1-\gamma)T} \leq \frac{(1-\gamma)^2}{128e\log(|\mathcal{S}||\mathcal{A}|)}.
$$

We can conclude that

$$
\begin{aligned}
&\mathbb{E}\left[\left\|\widehat{Q}_T^{\rho_{KL}(\sigma)} - \widehat{Q}^{*\rho_{KL}(\sigma)}\right\|_\infty^2\right] \\
&\overset{(i)}{\leq} \frac{3}{2}\left\|\widehat{Q}_0^{\rho_{KL}(\sigma)} - \widehat{Q}^{*\rho_{KL}(\sigma)}\right\|_\infty^2 \prod_{j=0}^{T-1}\left(1 - \frac{1-\gamma}{2}\beta_t\right) + \frac{16e\log(|\mathcal{S}||\mathcal{A}|)}{1-\gamma}16r_{\max}^2\left(1 + \frac{\gamma^2}{(1-\gamma)^2}\right) \\
&\quad \cdot\left(1 + (N_{\max}+1)\left(6\frac{C_{KL}^2}{\sigma^2 p_\wedge^2} + 7 + \frac{\log(2|\mathcal{S}|^2|\mathcal{A}|)\log T}{p_\wedge^2}\right)\right)\sum_{i=0}^{T-1}\beta_i^2\prod_{t=i+1}^{T-1}(1 - \frac{1-\gamma}{2}\beta_t) \\
&\overset{(ii)}{\leq} \frac{3}{2}\frac{r_{\max}^2}{(1-\gamma)^2}\frac{1}{T} + \frac{16e\log(|\mathcal{S}||\mathcal{A}|)}{1-\gamma}16r_{\max}^2\left(1 + (N_{\max}+1)\left(6\frac{C_{KL}^2}{\sigma^2 p_\wedge^2} + 7 + \frac{\log(2|\mathcal{S}|^2|\mathcal{A}|)\log T}{p_\wedge^2}\right)\right) \\
&\quad \cdot\left(1 + \frac{\gamma^2}{(1-\gamma)^2}\right)\frac{4\log T}{(1-\gamma)^2 T}, \quad (159)
\end{aligned}
$$

where $(i)$ follows from the Lemma B.5. $(ii)$ follows from $(1 - (1-\gamma)\beta/2)^T \leq \frac{1}{T}$.

Set $N_{\max} = \frac{2\log T}{\log 2}$. Then, we make the decomposition and get the bound of $\mathbb{E}\left[\left\|\widehat{Q}_T^{\rho_{KL}(\sigma)} - Q^{*\rho_{KL}(\sigma)}\right\|_\infty^2\right]$ as follows

$$
\begin{aligned}
&\mathbb{E}\left[\left\|\widehat{Q}_T^{\rho_{KL}(\sigma)} - Q^{*\rho_{KL}(\sigma)}\right\|_\infty^2\right] \\
&\leq 2\mathbb{E}\left[\left\|\widehat{Q}_T^{\rho_{KL}(\sigma)} - \widehat{Q}^{*\rho_{KL}(\sigma)}\right\|_\infty^2\right] + 2\mathbb{E}\left[\left\|\widehat{Q}^{*\rho_{KL}(\sigma)} - Q^{*\rho_{KL}(\sigma)}\right\|_\infty^2\right]
\end{aligned}
$$

$$
\overset{(i)}{\leq} \frac{2r_{\max}^2}{(1-\gamma)^2 T} + \frac{32e\log(|\mathcal{S}||\mathcal{A}|)}{1-\gamma} 16r_{\max}^2 \left(1 + (N_{\max}+1)\left(6\frac{C_{KL}^2}{\sigma^2 p_\wedge^2} + 7 + \frac{\log(2|\mathcal{S}|^2|\mathcal{A}|)\log T}{p_\wedge^2}\right)\right)
$$
$$
\cdot \left(1 + \frac{\gamma^2}{(1-\gamma)^2}\right) \frac{4\log T}{(1-\gamma)^2 T} + \frac{2}{1-\gamma}\left(\left(\frac{\gamma r_{\max}}{1-\gamma} + r_{\max}\right) 2^{-\frac{N_{\max}+1}{2}}\left(2^{-\frac{N_{\max}+1}{2}} + \frac{C_{KL}}{\sigma p_\wedge}\right)\right)^2
$$
$$
\overset{(ii)}{\leq} \frac{2r_{\max}^2}{(1-\gamma)^2 T} + \frac{32e\log(|\mathcal{S}||\mathcal{A}|)}{1-\gamma} 16r_{\max}^2 \left(1 + (N_{\max}+1)\left(6\frac{C_{KL}^2}{\sigma^2 p_\wedge^2} + 7 + \frac{\log(2|\mathcal{S}|^2|\mathcal{A}|)\log T}{p_\wedge^2}\right)\right)
$$
$$
\cdot \left(1 + \frac{\gamma^2}{(1-\gamma)^2}\right) \frac{4\log T}{(1-\gamma)^2 T} + \frac{2}{1-\gamma}\left(\left(\frac{\gamma r_{\max}}{1-\gamma} + r_{\max}\right)\left(\frac{1}{T} + \frac{C_{KL}}{\sigma p_\wedge}\right)\right)^2 \frac{1}{T}
$$
$$
= \widetilde{\mathcal{O}}\left(\frac{1}{(1-\gamma)^5 p_\wedge^2 \sigma^2 T}\right), \tag{160}
$$

where $(i)$ follows from Lemma B.4 and Theorem 4.1. $(ii)$ follows from $2^{\frac{\log T}{\log 2}} \leq \frac{1}{T}$.

When $\mathbb{E}\left[\left\|\widehat{Q}_T^{\rho_{KL}(\sigma)} - Q^{*,\rho_{KL}(\sigma)}\right\|_\infty^2\right] \leq \epsilon^2$, the iteration $T \geq \widetilde{\mathcal{O}}\left((1-\gamma)^{-5}\epsilon^{-2}p_\wedge^{-2}\sigma^{-2}\right)$. When $\psi = \frac{1}{2}$, the expected sample size per iteration is $N_{\max} + 2$. Above all, the total sample complexity is $\widetilde{\mathcal{O}}\left(|\mathcal{S}||\mathcal{A}|(1-\gamma)^{-5}\epsilon^{-2}p_\wedge^{-2}\sigma^{-2}\right)$. This completes the proof. $\qquad\square$

# F  PROOF OF LEMMAS AND PROPOSITIONS

*Proof of Lemma B.4.*

$$
\left\|\widehat{Q}^{*\rho(\sigma)} - Q^{*\rho(\sigma)}\right\|_\infty
$$
$$
= \left\|\bar{\mathcal{T}}_{N_{\max}}^{\rho(\sigma)}\left(\widehat{Q}^{*\rho(\sigma)}\right) - \mathcal{T}^{\rho(\sigma)}\left(Q^{*\rho(\sigma)}\right)\right\|_\infty
$$
$$
\leq \left\|\bar{\mathcal{T}}_{N_{\max}}^{\rho(\sigma)}\left(\widehat{Q}^{*\rho(\sigma)}\right) - \bar{\mathcal{T}}_{\max}^{\rho(\sigma)}\left(Q^{*\rho(\sigma)}\right)\right\|_\infty + \left\|\bar{\mathcal{T}}_{N_{\max}}^{\rho(\sigma)}\left(Q^{*\rho(\sigma)}\right) - \mathcal{T}^{\rho(\sigma)}\left(Q^{*\rho(\sigma)}\right)\right\|_\infty
$$
$$
\overset{(i)}{\leq} \gamma\left\|\widehat{Q}^{*\rho(\sigma)} - Q^{*\rho(\sigma)}\right\|_\infty + \left\|\bar{\mathcal{T}}_{N_{\max}}^{\rho(\sigma)}\left(Q^{*\rho(\sigma)}\right) - \mathcal{T}^{\rho(\sigma)}\left(Q^{*\rho(\sigma)}\right)\right\|_\infty, \tag{161}
$$

where $(i)$ follows from Proposition B.3. $\qquad\square$

*Proof of Proposition B.2.* Here we recall the definition of $\delta_{s,a,N_2}^{\rho(\sigma)}(Q)$ that

$$
\delta_{s,a,N_2}^{\rho(\sigma)}(Q) := \sup_{\alpha \geq 0}\left\{f^{\rho(\sigma)}(\widehat{p}_{s,a,2^{N_2+1}}, \alpha, V)\right\}
$$
$$
- \frac{1}{2}\sup_{\alpha \geq 0}\left\{f^{\rho(\sigma)}(\widehat{p}_{s,a,2^{N_2}}^E, \alpha, V)\right\} - \frac{1}{2}\sup_{\alpha \geq 0}\left\{f^{\rho(\sigma)}(\widehat{p}_{s,a,2^{N_2}}^O, \alpha, V)\right\}. \tag{162}
$$

Then, we recall that

$$
f^{*\rho(\sigma)}(\hat{p}_{s,a,n}, V) := \sup_{\alpha \geq 0}\left\{f^{\rho(\sigma)}(\hat{p}_{s,a,n}, \alpha, V)\right\}. \tag{163}
$$

Thus, we can get that

$$
\mathbb{E}\left[\delta_{s,a,N_2}^{Q,\rho(\sigma)}\big|N_2\right] = \mathbb{E}\left[f^{*\rho(\sigma)}(\hat{p}_{s,a,2^{N_2+1}}, V)\big|N_2\right]
$$
$$
- \frac{1}{2}\mathbb{E}\left[f^{*\rho(\sigma)}(\hat{p}_{s,a,2^{N_2+1}}^O, V)\big|N_2\right] - \frac{1}{2}\mathbb{E}\left[f^{*\rho(\sigma)}(\hat{p}_{s,a,2^{N_2+1}}^E, V)\big|N_2\right]
$$
$$
= E\left[f^{*\rho(\sigma)}(\hat{p}_{s,a,2^{N_2+1}}, V)\big|N_2\right] - E\left[f^{*\rho(\sigma)}(\hat{p}_{s,a,2^{N_2}}, V)\big|N_2\right]. \tag{164}
$$

Take the expectation of the random variable $N_2 \sim \text{Geo}(\psi)$, we can obtain that

$$\mathbb{E}\left[\widehat{v}^{\rho(\sigma)}(Q(s,a))\right]$$

$$= \mathbb{E}\left[V(s'_{s,a,0}) + \frac{\delta^{Q,\rho(\sigma)}_{s,a,N_2}}{P_{N_2}}\right]$$

$$= \mathbb{E}[V(s'_{s,a,0})] + \mathbb{E}\left[\frac{\delta^{Q,\rho(\sigma)}_{s,a,N_2}}{P_{N_2}}\right]$$

$$\overset{(i)}{=} \mathbb{E}[V(s'_{s,a,0})] + \sum_{N=0}^{N_{\max}} \mathbb{E}\left[\frac{\delta^{Q,\rho(\sigma)}_{s,a,N_2}}{P_{N_2}}\Big| N_2 = N\right]\mathbb{P}(N) + \sum_{N=N_{\max}+1}^{\infty} \mathbb{E}\left[\frac{\delta^{Q,\rho(\sigma)}_{s,a,N_2}}{P_{N_2}}\Big| N_2 = N\right]\mathbb{P}(N)$$

$$\overset{(ii)}{=} E\left[f^{*\rho(\sigma)}(\hat{p}_{s,a,2^0}, V)\right] + \sum_{N=0}^{N_{\max}} \mathbb{E}\left[\delta_N^{Q,\rho(\sigma)}\right]$$

$$\overset{(iii)}{=} E\left[f^{*\rho(\sigma)}(\hat{p}_{s,a,2^0}, V)\right] + \sum_{N=0}^{N_{\max}} E\left[f^{*\rho(\sigma)}(\hat{p}_{s,a,2^{N+1}}, V)\right] - E\left[f^{*\rho(\sigma)}(\hat{p}_{s,a,2^N}, V)\right]$$

$$= E\left[f^{*\rho(\sigma)}(\hat{p}_{s,a,2^{N_{\max}+1}}, V)\right], \tag{165}$$

where $(i)$ and $(ii)$ follows from the Equation (13); $(iii)$ follows from Definition B.1.

This completes the proof. $\qquad\square$

*Proof of Proposition B.3.* For any $Q, Q' \in \mathbb{R}^{|\mathcal{S}||\mathcal{A}|}$, we have that

$$\bar{\mathcal{T}}^{\rho(\sigma)}_{N_{\max}}(Q)(s,a) - \bar{\mathcal{T}}^{\rho(\sigma)}_{N_{\max}}(Q')(s,a)$$

$$= \mathbb{E}\left[g^{*\rho(\sigma)}(\hat{\mu}_{s,a,2^{N_{\max}+1}}, r_{s,a}) + \gamma f^{*\rho(\sigma)}(\hat{p}_{s,a,2^{N_{\max}+1}}, V)\right]$$

$$\quad - \mathbb{E}\left[g^{*\rho(\sigma)}(\hat{\mu}_{s,a,2^{N_{\max}+1}}, r_{s,a}) + \gamma f^{*\rho(\sigma)}(\hat{p}_{s,a,2^{N_{\max}+1}}, V')\right]$$

$$= \gamma\left(\mathbb{E}\left[f^{*\rho(\sigma)}(\hat{p}_{s,a,2^{N_{\max}+1}}, V)\right] - \mathbb{E}\left[f^{*\rho(\sigma)}(\hat{p}_{s,a,2^{N_{\max}+1}}, V')\right]\right)$$

$$= \gamma\mathbb{E}_{\hat{p}_{s,a,2^{N_{\max}+1}}}\left[\inf_{\rho(q,\hat{p}_{s,a,2^{N_{\max}+1}})\leq\sigma}\mathbb{E}_q[V'(s'_{s,a})] - \inf_{\rho(q,\hat{p}_{s,a,2^{N_{\max}+1}})\leq\sigma}\mathbb{E}_q[V(s'_{s,a})]\right]$$

$$= \gamma\mathbb{E}_{\hat{p}_{s,a,2^{N_{\max}+1}}}\left[\inf_{\rho(q,\hat{p}_{s,a,2^{N_{\max}+1}})\leq\sigma}\mathbb{E}_q[\max_{a'}Q'(s'_{s,a}, a')]\right.$$

$$\left. - \inf_{\rho(q,\hat{p}_{s,a,2^{N_{\max}+1}})\leq\sigma}\mathbb{E}_q[\max_{a'}Q(s'_{s,a}, a')]\right] \tag{166}$$

Hence, consider the infinite norm of both sides Equation (166), we can get

$$\left\|\bar{\mathcal{T}}^{\rho(\sigma)}_{N_{\max}}(Q) - \bar{\mathcal{T}}^{\rho(\sigma)}_{N_{\max}}(Q')\right\|_\infty$$

$$\leq \max_{s,a}\left|\bar{\mathcal{T}}^{\rho(\sigma)}_{N_{\max}}(Q)(s,a) - \bar{\mathcal{T}}^{\rho(\sigma)}_{N_{\max}}(Q')(s,a)\right|$$

$$= \gamma\max_{s,a}\left|\mathbb{E}_{\hat{p}_{s,a,2^{N_{\max}+1}}}\left[\inf_{\rho(q,\hat{p}_{s,a,2^{N_{\max}+1}})\leq\sigma}\mathbb{E}_q[\max_{a'}Q'(s'_{s,a}, a')]\right.\right.$$

$$\left.\left. - \inf_{\rho(q,\hat{p}_{s,a,2^{N_{\max}+1}})\leq\sigma}\mathbb{E}_q[\max_{a'}Q(s'_{s,a}, a')]\right]\right|$$

$$\leq \gamma\max_{s,a}\left|\mathbb{E}_{\hat{p}_{s,a,2^{N_{\max}+1}}}\left[\sup_{\rho(q,\hat{p}_{s,a,2^{N_{\max}+1}})\leq\sigma}\mathbb{E}_q[\max_{a'}Q'(s'_{s,a}, a') - \max_{a'}Q(s'_{s,a}, a')]\right]\right|$$

$$\leq \gamma \max_{s,a} \max_{s'_{s,a}} \left| \max_{a'} Q'(s'_{s,a}, a') - \max_{a'} Q(s'_{s,a}, a') \right|$$

$$\leq \gamma \max_{s'} \max_{a'} |Q(s', a') - Q'(s', a')|$$

$$= \gamma \|Q - Q'\|_\infty. \tag{167}$$

$\square$

*Proof of Lemma C.2.* [Shi et al., 2023] Firstly, for a fixed $\alpha$, by Bernstein's inequality, we has that with probability at least $1 - \delta$,

$$\left| \mathbb{E}_{p_{s,a}} \left[ (V(s'_{s,a}))_\alpha \right] - \mathbb{E}_{\hat{p}_{s,a,N}} \left[ (V(s'_{s,a}))_\alpha \right] \right| \leq \sqrt{\frac{2 \log\left(\frac{2}{\delta}\right)}{N}} \sqrt{\mathrm{Var}_{p_{s,a}}((V(s'_{s,a}))_\alpha)} + \frac{2 r_{\max} \log\left(\frac{2}{\delta}\right)}{3N(1-\gamma)}. \tag{168}$$

Then, the term $\max_{0 \leq \alpha \leq \max_{s'_{s,a}} V(s'_{s,a})} \left| \mathbb{E}_{p_{s,a}} \left[ (V(s'_{s,a}))_\alpha \right] - \mathbb{E}_{\hat{p}_{s,a,N}} \left[ (V(s'_{s,a}))_\alpha \right] \right|$ is 1-Lipschitz w.r.t. $\alpha$ for any $V$ obeying $\|V\|_\infty \leq \frac{r_{\max}}{1-\gamma}$. In addition, we construct an $\epsilon_1$-net $N_{\epsilon_1}$ over $[0, \frac{r_{\max}}{1-\gamma}]$ whose size satisfies $|N_{\epsilon_1}| \leq \frac{3 r_{\max}}{\epsilon_1 (1-\gamma)}$ [Vershynin, 2018]. By union bound and Equation (168), with probability at least $1 - \delta$, we have that for all $\alpha \in N_{\epsilon_1}$,

$$\left| \mathbb{E}_{p_{s,a}} \left[ (V(s'_{s,a}))_\alpha \right] - \mathbb{E}_{\hat{p}_{s,a,N}} \left[ (V(s'_{s,a}))_\alpha \right] \right| \leq \sqrt{\frac{2 \log\left(\frac{2|N_{\epsilon_1}|}{\delta}\right)}{N}} \sqrt{\mathrm{Var}_{p_{s,a}}(V)} + \frac{2 r_{\max} \log\left(\frac{2|N_{\epsilon_1}|}{\delta}\right)}{3N(1-\gamma)}. \tag{169}$$

Then, we have that

$$\max_{0 \leq \alpha \leq \max_{s'_{s,a}} V(s'_{s,a})} \left| \mathbb{E}_{p_{s,a}} \left[ (V(s'_{s,a}))_\alpha \right] - \mathbb{E}_{\hat{p}_{s,a,N}} \left[ (V(s'_{s,a}))_\alpha \right] \right|$$

$$\overset{(a)}{\leq} \epsilon_1 + \sup_{\alpha \in N_{\epsilon_1}} \left| \mathbb{E}_{p_{s,a}} \left[ (V(s'_{s,a}))_\alpha \right] - \mathbb{E}_{\hat{p}_{s,a,N}} \left[ (V(s'_{s,a}))_\alpha \right] \right|$$

$$\overset{(b)}{\leq} \epsilon_1 + \sqrt{\frac{2 \log\left(\frac{2|N_{\epsilon_1}|}{\delta}\right)}{N}} \sqrt{\mathrm{Var}_{p_{s,a}}(V)} + \frac{2 r_{\max} \log\left(\frac{2|N_{\epsilon_1}|}{\delta}\right)}{3N(1-\gamma)}$$

$$\overset{(c)}{\leq} \sqrt{\frac{2 \log\left(\frac{2|N_{\epsilon_1}|}{\delta}\right)}{N}} \sqrt{\mathrm{Var}_{p_{s,a}}(V)} + \frac{2 r_{\max} \log\left(\frac{|N_{\epsilon_1}|}{\delta}\right)}{N(1-\gamma)}$$

$$\overset{(d)}{\leq} 2\sqrt{\frac{\log\left(\frac{2N}{\delta}\right)}{N}} \|V\|_\infty + \frac{2 r_{\max} \log\left(\frac{1}{\delta}\right)}{N(1-\gamma)}$$

$$\overset{(e)}{\leq} 3 r_{\max} \sqrt{\frac{\log\left(\frac{2N}{\delta}\right)}{(1-\gamma)^2 N}}, \tag{170}$$

where $(a)$ follows from that the parameter $\alpha^* = \arg\max_\alpha \left| \mathbb{E}_{p_{s,a}} \left[ (V(s'_{s,a}))_\alpha \right] - \mathbb{E}_{\hat{p}_{s,a,N}} \left[ (V(s'_{s,a}))_\alpha \right] \right|$ falls into a $\epsilon_1$ balls centered around some point inside $N_{\epsilon_1}$. $(b)$ follows from Equation (168). $(c)$ follows from taking $\epsilon_1 = \frac{r_{\max} \log\left(\frac{2|N_{\epsilon_1}|}{\delta}\right)}{3N(1-\gamma)}$. $(d)$ follows from that $|N_{\epsilon_1}| \leq \frac{3}{\epsilon_1(1-\gamma)} \leq 9N$. $(e)$ follows from the fact that $\|V\|_\infty \leq \frac{r_{\max}}{1-\gamma}$ and $N \geq \log\left(\frac{18N}{\delta}\right)$. This completes the proof. $\square$

*Proof of Lemma D.2.* [Shi et al., 2023] Firstly, we do error decomposition as follows

$$\max_{0\leq\alpha\leq\max_{s'_{s,a}}V(s'_{s,a})}\left|\mathbb{E}_{p_{s,a}}\left[(V(s'_{s,a}))_\alpha\right]-\sqrt{\sigma\text{Var}_{p_{s,a}}((V(s'_{s,a}))_\alpha)}-\mathbb{E}_{p_{s,a}}\left[(V(s'_{s,a}))_\alpha\right]+\sqrt{\sigma\text{Var}_{\hat{p}_{s,a,N}}((V(s'_{s,a}))_\alpha)}\right|$$

$$\leq\max_{0\leq\alpha\leq\max_{s'_{s,a}}V(s'_{s,a})}\left|\mathbb{E}_{p_{s,a}}\left[(V(s'_{s,a}))_\alpha\right]-\mathbb{E}_{\hat{p}_{s,a,N}}\left[(V(s'_{s,a}))_\alpha\right]\right|$$

$$+\max_{0\leq\alpha\leq\max_{s'_{s,a}}V(s'_{s,a})}\sqrt{\sigma}\left|\sqrt{\text{Var}_{p_{s,a}}((V(s'_{s,a}))_\alpha)}-\sqrt{\text{Var}_{\hat{p}_{s,a,N}}((V(s'_{s,a}))_\alpha)}\right|. \tag{171}$$

Then, consider the first terms in Equation (171). By Bernstein's inequality, for fixed $\alpha$, with probability at least $1-\delta$, we have that

$$\max_{0\leq\alpha\leq\max_{s'_{s,a}}V(s'_{s,a})}\left|\mathbb{E}_{p_{s,a}}\left[(V(s'_{s,a}))_\alpha\right]-\mathbb{E}_{\hat{p}_{s,a,N}}\left[(V(s'_{s,a}))_\alpha\right]\right|\leq 2r_{\max}\sqrt{\frac{\log\left(\frac{2N}{\delta}\right)}{(1-\gamma)^2N}}. \tag{172}$$

Next, consider the second term in Equation (171). According to the Lemma 6 in [Panaganti and Kalathil, 2022], with probability at least $1-\delta$, we have that

$$\left|\sqrt{\text{Var}_{p_{s,a}}((V(s'_{s,a}))_\alpha)}-\sqrt{\text{Var}_{\hat{p}_{s,a,N}}((V(s'_{s,a}))_\alpha)}\right|\leq\sqrt{\frac{2\log\left(\frac{2}{\delta}\right)}{(1-\gamma)^2N}}. \tag{173}$$

Next, we prove the Lipschitz property of the above term.

$$\left|\sqrt{\text{Var}_{p_{s,a}}((V(s'_{s,a}))_{\alpha_1})}-\sqrt{\text{Var}_{\hat{p}_{s,a,N}}((V(s'_{s,a}))_{\alpha_1})}\right|-\left|\sqrt{\text{Var}_{p_{s,a}}((V(s'_{s,a}))_{\alpha_2})}-\sqrt{\text{Var}_{\hat{p}_{s,a,N}}((V(s'_{s,a}))_{\alpha_2})}\right|$$

$$\leq\left|\sqrt{\text{Var}_{p_{s,a}}((V(s'_{s,a}))_{\alpha_1})}-\sqrt{\text{Var}_{\hat{p}_{s,a,N}}((V(s'_{s,a}))_{\alpha_1})}-\sqrt{\text{Var}_{p_{s,a}}((V(s'_{s,a}))_{\alpha_2})}+\sqrt{\text{Var}_{\hat{p}_{s,a,N}}((V(s'_{s,a}))_{\alpha_2})}\right|$$

$$\leq\left|\sqrt{\text{Var}_{p_{s,a}}((V(s'_{s,a}))_{\alpha_1})}-\sqrt{\text{Var}_{p_{s,a}}((V(s'_{s,a}))_{\alpha_2})}\right|+\left|\sqrt{\text{Var}_{\hat{p}_{s,a,N}}((V(s'_{s,a}))_{\alpha_1})}-\sqrt{\text{Var}_{\hat{p}_{s,a,N}}((V(s'_{s,a}))_{\alpha_2})}\right|$$

$$\overset{(a)}{\leq}\sqrt{\left|\text{Var}_{p_{s,a}}((V(s'_{s,a}))_{\alpha_1})-\text{Var}_{p_{s,a}}((V(s'_{s,a}))_{\alpha_2})\right|}+\sqrt{\left|\text{Var}_{\hat{p}_{s,a,N}}((V(s'_{s,a}))_{\alpha_1})-\text{Var}_{\hat{p}_{s,a,N}}((V(s'_{s,a}))_{\alpha_2})\right|}$$

$$\overset{(b)}{\leq}2\sqrt{2(\alpha_1+\alpha_2)\left|\alpha_1-\alpha_2\right|}\leq 4\sqrt{\frac{r_{\max}\left|\alpha_1-\alpha_2\right|}{1-\gamma}}, \tag{174}$$

where $(a)$ follows from $|\sqrt{x}-\sqrt{y}|\leq\sqrt{|x-y|}$ and $(b)$ follows from that

$$\left|\text{Var}_{p_{s,a}}((V(s'_{s,a}))_{\alpha_1})-\text{Var}_{p_{s,a}}((V(s'_{s,a}))_{\alpha_2})\right|$$

$$=\left|\mathbb{E}_{p_{s,a}}\left[((V(s'_{s,a}))_{\alpha_1})^2\right]-\left(\mathbb{E}_{p_{s,a}}\left[(V(s'_{s,a}))_{\alpha_1}\right]\right)^2-\mathbb{E}_{p_{s,a}}\left[((V(s'_{s,a}))_{\alpha_2})^2\right]+\left(\mathbb{E}_{p_{s,a}}\left[(V(s'_{s,a}))_{\alpha_2}\right]\right)^2\right|$$

$$\leq\left|\mathbb{E}_{p_{s,a}}\left[((V(s'_{s,a}))_{\alpha_1})^2-((V(s'_{s,a}))_{\alpha_2})^2\right]\right|+\left|\left(\mathbb{E}_{p_{s,a}}\left[(V(s'_{s,a}))_{\alpha_1}\right]\right)^2-\left(\mathbb{E}_{p_{s,a}}\left[(V(s'_{s,a}))_{\alpha_2}\right]\right)^2\right|$$

$$\overset{(a)}{\leq}2(\alpha_1+\alpha_2)|\alpha_1-\alpha_2|, \tag{175}$$

where $(a)$ follows from $(V(s'_{s,a}))_\alpha\leq\alpha$.

To prove the union bound, we also construct an $\epsilon_2$-net $N_{\epsilon_2}$ over $\left[0,\frac{r_{\max}}{1-\gamma}\right]$ [Vershynin, 2018]. With probability at least $1-\delta$, we have that

$$\max_{0\leq\alpha\leq\max_{s'_{s,a}}V(s'_{s,a})}\sqrt{\sigma}\left|\sqrt{\text{Var}_{p_{s,a}}((V(s'_{s,a}))_\alpha)}-\sqrt{\text{Var}_{\hat{p}_{s,a,N}}((V(s'_{s,a}))_\alpha)}\right|$$

$$\overset{(a)}{\leq}4\sqrt{\frac{r_{\max}\epsilon_2}{1-\gamma}}+\sup_{\alpha\in N_{\epsilon_2}}\left|\sqrt{\text{Var}_{p_{s,a}}((V(s'_{s,a}))_\alpha)}-\sqrt{\text{Var}_{\hat{p}_{s,a,N}}((V(s'_{s,a}))_\alpha)}\right|$$

$$\overset{(b)}{\leq}4\sqrt{\frac{r_{\max}\epsilon_2}{1-\gamma}}+r_{\max}\sqrt{\frac{2\log\left(\frac{2|N_{\epsilon_2}|}{\delta}\right)}{(1-\gamma)^2N}}$$

$$\overset{(c)}{\leq} 2r_{\max} \sqrt{\frac{2 \log\left(\frac{2|N_{\epsilon_2}|}{\delta}\right)}{(1-\gamma)^2 N}}$$

$$\overset{(d)}{\leq} 2r_{\max} \sqrt{\frac{2 \log\left(\frac{24N}{\delta}\right)}{(1-\gamma)^2 N}}, \tag{176}$$

where $(a)$ follows from the property of $N_{\epsilon_2}$. $(b)$ follows from Equation (174). $(c)$ follows from taking $\epsilon_2 = \frac{r_{\max} \log\left(\frac{2|N_{\epsilon_2}|}{\delta}\right)}{8N(1-\gamma)}$. $(d)$ follows from that $|N_{\epsilon_2}| \leq \frac{3}{\epsilon_2(1-\gamma)} \leq 24N$.

$\square$

This completes the proof.

