# OpenReview forum: "Model-Free Robust Reinforcement Learning with Sample Complexity Analysis"
_auai.org/UAI/2024/Conference — UAI 2024 poster_

### Official Review · Reviewer_dDey · 2024-03-22

**Q2-1 Originality-Novelty:** 3
**Q2-2 Correctness-Technical Quality:** 4
**Q2-5 Clarity Of Writing:** 4

**Q10 Ethical Concerns:**

No.

**Q1 Summary And Contributions:**

The paper introduces a Distributionally Robust Reinforcement Learning (DR-RL) algorithm using Model-Level Monte Carlo (MLMC) estimators to address model uncertainty in reinforcement learning tasks. Its contributions include developing the model-free DR-RL approach, which provides finite sample complexity analyses for uncertainty sets defined by total variation, Chi-square divergence, and KL divergence. This approach improves sample complexity over existing models for these uncertainty sets.

**Q2-3 Extent To Which Claims Are Supported By Evidence:**

3: Good: the main claims are supported by convincing evidence (in the form of adequate experimental evaluation, proofs, (pseudo-)code, references, assumptions).

**Q2-4 Reproducibility:**

3: Good: key resources (e.g. proofs, code, data) are available and key details (e.g. proofs, experimental setup) are sufficiently well-described for competent researchers to confidently reproduce the main results.

**Q3 Main Strengths:**

- The proposed method deals with three types of uncertainty sets, highlighting its versatility and broader applicability.
- It enhances sample complexity for handling uncertainties.
- Overall, the paper provides a thorough explanation of the methodologies and their implications, along with a justification for the proposed approach, making it accessible and convincing.

**Q4 Main Weakness:**

- Lack of baseline comparison in experiments.
- The empirical evaluation is limited to a simple task setup, raising questions about performance in real-world scale problems.

**Q5 Detailed Comments To The Authors:**

- The paper could benefit from a more extensive comparison with baseline models, especially in demonstrating how the proposed methods converge on the same problems, to highlight its advantages more clearly.
- Can you provide specific examples or case studies where your proposed algorithm has been or could be successfully applied to solve real-world-level tasks? How does the algorithm address the challenges specific to these applications?
- The introduction of bias through the threshold design is acknowledged as a trade-off for achieving finite sample complexity. Could you discuss further how this bias impacts the learned policy's overall performance and reliability and compareit to the bias present in other methods?

**Q9 Complying With Reviewing Instructions:**

Yes

---

> ### Author Rebuttal · Authors · 2024-04-05
>
> **Weaknesses 1, 2 \& Comments 1: lack of baseline comparison in experiments; performance of T-MLMC algorithm in real-world scale problems**
>
> Thanks for your comments. We have the optimal robust value functions as the baseline, but we will include comparison with other robust RL algorithms. Since we mainly consider tabular setting so we only adopt simple experiments. Extending our T-MLMC to large scale problems, although doable, can results in large computational cost. We will add experiments on large-scale problems and explore potential theoretical extensions.
>
> We have updated the results. In the link https://anonymous.4open.science/r/UAI_278_results_-B68A/ , you will find the results for the FrozenLake environment, Gambler's game, Large-scale environment, and the recycling robot. The results also include comparisons between T-MLMC, the model-based method, and vanilla MLMC.
>
> In these figures, the y-axis represents the expected cumulative reward $J^\pi= \mathbb{E}[V^\pi_{\mathbf{P}_0,\mathbf{R}_0}(s)]$.
>
> For the results where T-MLMC is compared with the model-based method and vanilla MLMC, the x-axis represents the number of samples. We set the model-based method's sampling at $2^{N_{\max}+1}$ samples per step, where both T-MLMC and the model-based method will converge to the same $\epsilon$-accurate optimal policy.  From these results, we can observe that our T-MLMC algorithm is computationally economical, which is also an advantage of model-free methods.
>
>
> Furthermore, additional simulation results demonstrate the convergence of our T-MLMC. When choosing the suitable $N_{\max}$ the bias introduced by the T-MLMC estimator will not lead to instability of the system. The algorithm will converge to the $\epsilon$-accurate optimal robust policy.
>
> **Comments 2: provide specific examples or case studies where your proposed algorithm has been or could be successfully applied to solve real-world-level tasks? How does the algorithm address the challenges specific to these applications?**
>
> Thanks for your comments. We currently apply our T-MLMC algorithm to design the policy of recycling robot problem (Example 3.3 [1]). A mobile robot running on a rechargeable battery aims to collect empty soda cans. It has 2 battery levels: low and high. The robot can either 1) search for empty cans; 2) remain stationary and wait for someone to bring it a can; 3) go back to its home base to recharge. Under low (high) battery level, the robot finds an empty can with probabilities $\alpha$ ($\beta$), and remains at the same battery level.  If the robot goes out to  search but finds nothing, it will run out of its battery and can only be carried back by human.  We introduce model uncertainty to the probabilities $\alpha,\beta$ of finding an empty can if the robot chooses the action `search'. We implement our algorithm under this problem.
>
> The challenge here is that the collected samples can be limited, and it may result in inaccurate estimation and sub-optimal performance. Our T-MLMC algorithm ensures that with limited sample, we will obtain the optimal policy under the problem. Compared to previous methods with strong assumption or infinite sample numbers, our algorithm can be implemented in practice and has better capability. In the link https://anonymous.4open.science/r/UAI_278_results_-B68A/ , you will find the results for the recycling robot. In the results of recycling robot, The perturbation describes the uncertainty level of the test environment.
>
>
> Besides, we also plan to extend our T-MLMC algorithm to solve the stability problems in wireless connections.
>
>
> [1]Sutton R S, Barto A G. Reinforcement learning: An introduction[M]. MIT press, 2018.
>
> **Comments 3: discuss further how this bias impacts the learned policy's overall performance and reliability and compare it to the bias present in other methods**
>
> The bias in the operator estimation can be made small by choosing $N_{\max}=\log1/\epsilon$, and its effect on the performance of the learned policy can be bounded. Namely, our algorithm is shown to converge to the **optimal** robust policy, and hence has the best performance. Moreover, the sample complexity/convergence rate of our algorithm is better, meaning that with the same amount of data, our algorithm can obtain a better policy.
>
> The trade-off is between the bias and sample complexity/assumptions. Namely, to ensure the estimator is unbiased, either infinite samples [Liu et al., 2022] or restrictive assumptions [Wang et al., 2023 ab] are required; Yet if we admit bias, we no longer need any assumptions or infinite data. However, both approaches will have similar performance, as they both converge to the optimal policy.

---

### Official Review · Reviewer_6UQA · 2024-03-24

**Q2-1 Originality-Novelty:** 2
**Q2-2 Correctness-Technical Quality:** 3
**Q2-5 Clarity Of Writing:** 3

**Q1 Summary And Contributions:**

This paper studies model-free distributionally robust RL. It proposes a T-MLMC algorithm, and proves finite sample guarantees under three types of uncertainty sets.

**Q2-3 Extent To Which Claims Are Supported By Evidence:**

3: Good: the main claims are supported by convincing evidence (in the form of adequate experimental evaluation, proofs, (pseudo-)code, references, assumptions).

**Q2-4 Reproducibility:**

3: Good: key resources (e.g. proofs, code, data) are available and key details (e.g. proofs, experimental setup) are sufficiently well-described for competent researchers to confidently reproduce the main results.

**Q3 Main Strengths:**

The paper proposes to use a threshold-MLMC technique to address a challenge in the existing literature using MLMC and studies the sample complexity under commonly used uncertainty sets.

**Q4 Main Weakness:**

The contribution of this paper is somehow incremental. The only novel idea is incorporating a threshold into the MLMC method to control the sample size. While the reviewer may lack a corresponding background in MLMC and is unsure whether the threshold technique is entirely novel in the field of MLMC, similar ideas utilizing thresholds are common in bandit and reinforcement learning literature.

There is room for improvement in the current presentation of this paper. For example, some results are incorrectly cited, some notation is used without definition, some arguments are ambiguous.

**Q5 Detailed Comments To The Authors:**

In Table 1 and Table 2, the two results of Shi et at., 2023 actually depend on the uncertainty level $\sigma$ in both TV and $\chi^2$ setting, which are omitted in your citation.

In the contribution part, you claim that your result on KL “significantly enhancing previous findings for the KL divergence model”, while in Table 3, your result is worse by $1/(1-\gamma)$. This point is not clear until the end of section 4.

In the related work section, you might also need to include some function approximation results in DR-RL, e.g., Liu & Xu, Distributionally Robust Off-Dynamics Reinforcement Learning: Provable Efficiency with Linear Function Approximation. (AISTATS 2024)

Why do the results in Theorem 4.1 not depend on the range $r_\max$ and $1/(1-\gamma)$?

The $P_{N_1}$ in (12) is not defined.

How to estimate the empirical reward distribution $\hat{\mu}$? For example, using some non-parametric estimators?

Typo: the sample complexity in Theorem 4.2 should be ‘’greater than or equal to’’.

The argument “We note that for model-free algorithms or general stochastic approximation algorithms [Li et al., 2020, 2021], the tightest dependence on $(1 − \gamma)$ is also $O((1 − \gamma)^{−5})$, which implies the tightness of our complexity result.” is problematic. As far as the reviewer knows, Li et al, 2021 achieve $O((1 − \gamma)^{−4})$ and $O((1 − \gamma)^{−3})$ sample complexities.

The argument “Compared to the model-based methods, our complexity presents an additional $O((1 − \gamma)^{−1})$- order dependence, which is common in model-free algorithms.” Please provide evidence or citations. The model-free algorithms for offline and online setting match the lower bounds [1][2].

As for the discussion of the restriction of precious paper, it seems that even when $\sigma=\infty$, $p_{^}\geq O(1)$ is reasonable. Why “This assumption significantly limits the applicability of their results”?

**Q9 Complying With Reviewing Instructions:**

Yes

---

> ### Author Rebuttal · Authors · 2024-04-05
>
> **Contribution of this paper**
>
> The design of the threshold in MLMC is new and novel, as it fundamentally changes the original motivation behind the MLMC. Specifically, the MLMC is designed to construct an **unbiased** estimator, with the price of requiring infinite number of samples or restrictive assumptions. To bypass these issues, we design this threshold in MLMC that enables us to control the number of samples, resulting a biased estimator. We further develop a novel analysis of biased stochastic approximation framework, showing that although it is biased, our algorithm still converges to the optimal policy with a reduced sample complexity. This motivation of design is hence different from MLMC, and enables us to obtain stronger results.
>
> We also want to highlight that we further developed studies for two more uncertainty sets that are not previously studied. It is worth noting that such an extension is not possible using previous approaches. The method in [Wang et al., 2023b] relies on the smoothness of the dual-form of the DRO problem with KL-divergence uncertainty set, which does not hold for TV and CS uncertainty sets. Whereas, our analysis does not rely on it and can be adapted under all uncertainty sets.
>
> **Comments 1:** Since we mainly want to highlight the dependence on other parameters, e.g., $(1-\gamma, p_\wedge, S , A) $. we omit the uncertainty level in the table. We add them in the revised version.
>
> **Comments 2:** Our comparison is mainly with vanilla approaches. The improvements of our results are two-folds: we get rid of any additional assumptions; And we have a better complexity compared to the vanilla algorithm. Their better result relies on the variance reduction technique, which is a standard technique to reduce sample complexity and can be viewed as a modification of vanilla algorithms, hence we do not focus on it. We will make it more clear in paper.
>
> **Comments 3:**  Since we mainly focus on the tabular setting we omit the large body of function approximation researches. We will include a part of  discussion on these works. [Tamar et al., 2014] firstly extend the robust MDP to linear function approximation and provide asymptotic convergence. Subsequently, numerous studies have explored robust MDPs with linear function approximation, including works by [Panaganti et al, 2022],  [Badrinath et al., 2021], [Ma et al., 2022], [Blanchet et al., 2023], [Zhou et al., 2023],and [Liu, \& Xu, 2024]. These studies have proposed related algorithms and theoretical results.
>
> **Comments 4:** The explicate dependence on $r_{\max}$ and $\frac{1}{1-\gamma}$ can be found in eq. (44),(55) for TV, (68),(79) for $\chi^2$); eq. (98),(112) for KL. We will revise the paper accordingly.
>
> **Comments 5:** $P\_{N_1}$ denotes the probability of $N=N_1$ where $N$ is the random level number generated from the geometry distribution.
>
> **Comments 6:** In our work, we assume the reward distribution is discrete. Therefore, we estimate $\hat{\mu}$ using the empirical distribution, i.e., $\hat{\mu}=\frac{\sum_i \textbf{1}_{R=r_i}}{N}$. In practice, when the reward is continuous, we can use k nearest neighbor or kernel density estimation to estimate the reward distribution.
>
> **Comments 7:** We will correct this typo in revised version.
>
> **Comments 8:** We will modify our discussion accordingly. Our results match the state-of-the-art in model-based approaches for $\chi^2$ and KL uncertainty sets in terms of $1-\gamma$, but presents a gap between the model-based results under TV case. Comparing our analysis with [Shi et al., 2022], we prove that applying a tighter variance analysis improves our TV results to $\mathcal{O}((1-\gamma)^{-4})$. We will update the results in revised version.  Additionally, applying variance reduction to our T-MLMC algorithm could achieve the same complexity as [Shi et al., 2022], which we aim to explore in the future.
>
> **Comments 9:** We apologize for the misleading statement here. Take the standard non-robust RL problem as an example. The lower bound/optimal complexity for model-based RL is $\mathcal{O}((1-\gamma)^{-3})$ [1], whereas the model-free Q-learning presents  $\mathcal{O}((1-\gamma)^{-4})$ [Li et al., 2021] (without variance reduction technique), and variance reduced Q-learning achieves the minimax lower bound. From this aspect, the vanilla model-free algorithm has an additional dependence on $1-\gamma$, which is expected in the robust setting.
>
> [1] Li et al. Breaking the sample size barrier in model-based reinforcement learning with a generative model.
>
> **Comments 10:** In [Wang et. al., 2023ab],  the assumption is required that
> $\frac{1}{2}p_\wedge \geq 1- e^{-\sigma}$. It implies that $\frac{1}{2}p_\wedge \geq 1- e^{-\sigma}\geq \frac{\sigma}{2}$, the radius of uncertainty set has to be very small if $p_\wedge$ is small.
>
> Moreover, since there is no information about $p_\wedge$ in practice, it can be challenging to design an uncertainty set satisfying the assumption.

---

### Official Review · Reviewer_j69A · 2024-03-27

**Q2-1 Originality-Novelty:** 2
**Q2-2 Correctness-Technical Quality:** 3
**Q2-5 Clarity Of Writing:** 3

**Q1 Summary And Contributions:**

This paper proposes a model-free DR-RL algorithm leveraging the Multi-level Monte Carlo (MLMC) technique. The proposed algorithm
can be applied to uncertainty sets defined by total variation, Chi-square divergence, and KL divergence. The authors further establish the tightest sample complexity for all three uncertainty models.

**Q2-3 Extent To Which Claims Are Supported By Evidence:**

3: Good: the main claims are supported by convincing evidence (in the form of adequate experimental evaluation, proofs, (pseudo-)code, references, assumptions).

**Q2-4 Reproducibility:**

3: Good: key resources (e.g. proofs, code, data) are available and key details (e.g. proofs, experimental setup) are sufficiently well-described for competent researchers to confidently reproduce the main results.

**Q3 Main Strengths:**

1. The paper is well-written and easy to follow.
2. The theoretical results are new and tightest. And the proofs are sound.

**Q4 Main Weakness:**

1. It seems the proposed algorithm has some similarities with (Wang et al, 2023a) and (Wang et al, 2023b). The only difference is the number of samples in each iteration. However, can the authors explain the key differences and challenges in the analysis so that the technical contribution of the proposed algorithm is clear?

2. I understand that this is a theory-oriented paper. However, I wonder whether the proposed algorithm can work in real-life DR-RL problems using carefully designed sample number N_max. I suggest the authors explore this question by an empirical study.

**Q5 Detailed Comments To The Authors:**

It seems (Wang et al, 2023b) have a lower sample complexity under the KL uncertainty set thanks to the variance reduce technique. I wonder if the same idea of (Wang et al, 2023b) can be extended to the other two uncertainty sets easily. If so, is it possible to have lower sample complexities in those settings? If not, can the authors explain the challenge and why the proposed algorithm can work in all settings?

**Q9 Complying With Reviewing Instructions:**

Yes

---

> ### Author Rebuttal · Authors · 2024-04-05
>
> **Weaknesses 1: explain the key differences and challenges in the analysis**
>
> Our algorithm enjoys a distinct threshold design, however, the motivations of algorithm designs are different. In [Wang et al, 2023b], they need restrictive assumptions to ensure their MLMC estimator is **unbiased** and **requires finite samples**; In our algorithm, we introduce a threshold to remove the assumption, at the price of a **biased** estimator. This introduces a great challenge in the analysis, since the accumulative error from the bias needs to be analyzed. Our technical contribution lies in the analysis of our biased stochastic approximation framework.
>
> We also want to highlight that we further developed studies for two more uncertainty sets that are not previously studied. It is worth noting that such an extension is not possible using previous approaches. The method in [Wang et al., 2023ab] relies on the smoothness of the dual-form of the DRO problem with KL-divergence uncertainty set. Therefore, an extra assumption is made in  [Wang et al., 2023ab] to ensure its smoothness. However, the dual functions for TV and $\chi^2$ uncertainty set are not smooth, implying no direct extension from KL-case. Whereas, our analysis does not rely on such a smoothness property, and can be adapted under all uncertainty sets.
>
> **Weaknesses 2: provide empirical study applying our T-MLMC algorithm  in real-life DR-RL problems**
>
> Our experiments are provided in the appendix due to space limitation.
>
> We further provide an experiment on a real-life problem: recycling robot problem (Example 3.3 [1]. A mobile robot running on a rechargeable battery aims to collect empty soda cans. It has 2 battery levels: low and high. The robot can either 1) search for empty cans; 2) remain stationary and wait for someone to bring it a can; 3) go back to its home base to recharge. Under low (high) battery level, the robot finds an empty can with probabilities $\alpha$ ($\beta$), and remains at the same battery level.  If the robot goes out to  search but finds nothing, it will run out of its battery and can only be carried back by human.  We introduce model uncertainty to the probabilities $\alpha,\beta$ of finding an empty can if the robot chooses the action `search'. We implement our algorithm under this problem.
>
> We have updated the results. In the link https://anonymous.4open.science/r/UAI_278_results_-B68A/ , you will find the results for the FrozenLake environment, Gambler's game, and the recycling robot.
>
> In these figures, the y-axis represents the expected cumulative reward $J^\pi= \mathbb{E}[V^\pi_{\mathbf{P}_0,\mathbf{R}_0}(s)]$.
>
> For the results where T-MLMC is compared with the model-based method and vanilla MLMC, the x-axis represents the number of samples. We set the model-based method's sampling at $2^{N_{\max}+1}$ samples per step, where both T-MLMC and the model-based method will converge to the same $\epsilon$-accurate optimal policy.  From these results, we can observe that our T-MLMC algorithm is computationally economical, which is also an advantage of model-free methods.
>
>
> Furthermore, additional simulation results demonstrate the convergence of our T-MLMC. When choosing the suitable $N_{\max}$ the bias introduced by the T-MLMC estimator will not lead to instability of the system. The algorithm will converge to the $\epsilon$-accurate optimal robust policy.
>
> **Questions: Whether [Wang et al., 2023b] can be extended to the other two uncertainty sets easily?  If so, is it possible to have lower sample complexities in those settings? If not, can the authors explain the challenge and why the proposed algorithm can work in all settings?**
>
> The analysis method in [Wang et al., 2023b] relies on the smoothness of the dual-form of the DRO problem with KL-divergence defined uncertainty.
> Therefore, an extra assumption is made in  [Wang et al., 2023b] to ensure its smoothness (eq. (3)). However, the dual functions for TV and $\chi^2$ uncertainty sets are not smooth, implying that direct extension of  [Wang et al, 2023b] is not feasible.
> Whereas, our analysis does not rely on such a smoothness assumption, and can be applied under all uncertainty sets.
>
>
> The lower complexity in [Wang et al., 2023b] is due to the variance reduction technique. It is expected that such technique can also be applied in our algorithm design, but it requires more detailed analysis and is left for future work.
>
> [1]Sutton R S, Barto A G. Reinforcement learning: An introduction[M]. MIT press, 2018.

---

### Official Review · Reviewer_XEXK · 2024-03-28

**Q2-1 Originality-Novelty:** 2
**Q2-2 Correctness-Technical Quality:** 3
**Q2-5 Clarity Of Writing:** 3

**Q1 Summary And Contributions:**

This paper studies the problem of Distributionally Robust Reinforcement Learning (DR-RL) using model-free RL methods. In particular, the authors propose threshold-MLMC (T-MLMC, Algorithm 1), which is based on Q-learning and deals with uncertainty sets defined by total variation, Chi-square divergence, and KL divergence, achieving $O\Big( \frac{ |\mathcal{S}| \  | \mathcal{A} |  }{ (1 - \gamma)^5  \epsilon^2 } \Big)$ sample complexity (Tables 1-3).

On the technical level, to address the biased updating as reviewed in Sec. 3, Algorithm 1 uses uses Multi-level Monte Carlo (MLMC) technique (Liu et al., 2022, Wang et al., 2023c), which requires infinitely many samples to construct estimator. Therefore, the major innovation of this paper is to modify the MLMC by designing a threshold on the level number. The sample complexity results are obtained by choosing the threshold to be $O(\log{T})$.

**Q2-3 Extent To Which Claims Are Supported By Evidence:**

3: Good: the main claims are supported by convincing evidence (in the form of adequate experimental evaluation, proofs, (pseudo-)code, references, assumptions).

**Q2-4 Reproducibility:**

3: Good: key resources (e.g. proofs, code, data) are available and key details (e.g. proofs, experimental setup) are sufficiently well-described for competent researchers to confidently reproduce the main results.

**Q3 Main Strengths:**

1. Strong sample complexity results on DR-RL problem using model-free RL methods.
2. Using threshold in MLMC seems a novel idea.
3. Results contain multiple uncertainty sets defined by total variation, Chi-square divergence, and KL divergence.

**Q4 Main Weakness:**

1. Results improves marginally comparing with existing methods of Wang et al., 2023 ab in terms of improved constant dependences.

**Q5 Detailed Comments To The Authors:**

1. As noted in Sec. 3, the T-MLMC estimator also requires infinitely many samples (as $N_\max \to \infty$) to reduce biases. What is the intuition of its improvement over the original MLMC estimator (which requires infinite expected amount of samples as well)?
2. Below Eq. (12), why is the calculation separated according to even and odd indices?

**Q9 Complying With Reviewing Instructions:**

Yes

---

> ### Author Rebuttal · Authors · 2024-04-05
>
> **Weakness: Results improve marginally comparing with existing methods of Wang et al., 2023 ab in terms of improved constant dependences.**
>
> We first highlight that we additionally provide results for two more uncertainty sets: total variation and Chi-square, compared to [Wang et al., 2023 ab]. The analyses of which are highly different from the one for KL divergence in [Wang et al., 2023 ab].
>
> Compared with previous works on KL uncertainty set, besides the sample complexity, our major improvement is that we get rid of the restrictive assumption they made, which is $$\frac{1}{2}p_\wedge \geq 1- e^{-\sigma},$$ where the $p_\wedge$ is the  minimum positive entry of the nominal transition kernel.
> We can easily construct an uncertainty set such that this assumption does not hold. And therefore, this condition restricts the applicability of their approaches, and in this paper our results hold without the need of such assumption.
>
> **Questions 1: As noted in Sec. 3, the T-MLMC estimator also requires infinitely many samples (as $N_{\max}\to \infty$) to reduce biases. What is the intuition of its improvement over the original MLMC estimator (which requires infinite expected amount of samples as well)?**
>
> In our T-MLMC design, we do not require $N_{\max}\to \infty$. Instead, we set it to be a fixed number of order $N\_{\max}=\mathcal{O}(\log1/\epsilon)$, where $\epsilon$ is the desired accuracy. With our design, the sample required at each step is at most $\mathcal{O}(N\_{\max} \cdot 2^{\frac{N\_{\max}}{2}})$ and is hence finite.
>
> On the other hand, the expected number of sample required for vanilla MLMC in [Liu, et. al., 2022] is $\mathbb{E}_{N\sim \textbf{Geom}(0.5)}[N*2^N]=\infty$.
>
> As an intuition of such an improvement, the reduction in sample complexity comes with the price of bias. Namely when using the threshold, the resulting estimator is no longer unbiased. But we can show that even with the bias, our algorithm still obtains the optimal policy, with a finite sample complexity.
>
> **Questions 2: Below Eq. (12), why is the calculation separated according to even and odd indices**
>
> There is no requirement for the indices of the sampled data to be of any specific type. As long as the dataset is partitioned to two sub-collections with the same sizes. We adopt the even-odd separation as in the standard MLMC approach, but it can be directly generalized to other partitions.

---

### Official Review · Reviewer_5cr9 · 2024-03-29

**Q2-1 Originality-Novelty:** 3
**Q2-2 Correctness-Technical Quality:** 4
**Q2-5 Clarity Of Writing:** 3

**Q1 Summary And Contributions:**

This paper proposes a model-free algorithm based on the multi-level Monte Carlo (MLMC) technique to solve the problem of distributionally robust RL. As compared to previous approaches, the proposed algorithm incorporates a threshold for sampling, and thus ensures competitive sample complexity. In fact, the theoretical results show that, for TV-, $\chi^2$-divergence-, and KL-divergence-constrained $\texttt{sa}$-rectangular ambiguity sets, the proposed algorithm achieves the best known sample complexity for model-free algorithms. Numerical simulations are presented to show the practical performance of the algorithm.

**Q2-3 Extent To Which Claims Are Supported By Evidence:**

4: Excellent: all claims are supported by very convincing evidence (in the form of comprehensive experimental evaluation, rigorous mathematical proofs, detailed (pseudo-)code, precise references, well-motivated and realistic assumptions) and the authors deliver what they promise.

**Q2-4 Reproducibility:**

3: Good: key resources (e.g. proofs, code, data) are available and key details (e.g. proofs, experimental setup) are sufficiently well-described for competent researchers to confidently reproduce the main results.

**Q3 Main Strengths:**

* Overall, the paper is largely well-motivated, and also fits into the literature and on-going research.
* The literature review is quite comprehensive, with table comparing known sample complexities in multiple cases, which familiarizes the readers with the current progress and challenges in this direction.
* All theoretical results are concrete and rigorously written. The proof sketch in Section 5 is helpful for getting the main proof ideas.
* The proofs in the appendix are detailed (though indeed very technical so that I spend more-than-expected time on it...).
* The flow and writing of the paper is largely clear and reader-friendly.

**Q4 Main Weakness:**

* As a reader who is familiar with robust MDPs but has never heard of MLMC, the design ideas in Section 3 seem unnatural and hard to digest.
    + Though I have a rough feeling that splitting the even- and odd-indexed samples in MLMC is just *the* standard approach (after scanning some of the references), I'm still confused why this would work on a high level. Would splitting into more classes (say, modulo $k$ for a generic $k$) be even better? Why is parity of indices so special here?
    + Probably as a result of the previous point, the "correction terms" $\delta^{r, \rho(\sigma)}\_{s,a,N\_1}$ in eq. (12) and $\delta^{\rho(\sigma)}\_{s,a,N\_2}$ in eq. (13) seem very much under-motivated to me — they just come out of nowhere.
    + Is adding the $N_{\max}$ threshold the only change made by this paper? A few sentences comparing the current design against previous designs would be helpful.
    + As a personal suggestion, an introductory paragraph stating the high-level intuition behind MLMC would also be very helpful.
* The motivating argument that criticizes model-based algorithms for storage inefficiency sounds a little insufficient to me.
    + It's true that storing the estimated transition kernel requires $O(S^2 A)$ space, but storing the Q-function also requires $O(SA)$ space, which is not significantly better than the former when $\mathcal{S}$ is infinitely large.
    + Personally I don't think this issue can be resolved without leveraging the intrinsic low-rankness of MDPs and introducing some kind of representation, whether model-based or model-free approaches are used.
    + I appreciate the fact that Tables 1 through 3 honestly list the complexities of model-based algorithms, some of which turn out to be superior to the proposed model-free algorithm. That said, the paper largely avoids talking about the better complexities achieved by model-based algorithms.
    + I don't think [Panaganti & Kalathil, 2022] fall into the category of model-based algorithms. It does not assume knowledge of the nominal model, but rather, assume a known linear representation of value functions, which is very reasonable for reducing the complexity in my opinion.

**Q5 Detailed Comments To The Authors:**

Please refer to the previous parts.

**Q9 Complying With Reviewing Instructions:**

Yes

---

> ### Author Rebuttal · Authors · 2024-04-05
>
> **Weaknesses 1.a and b:**
>
> We appreciate your suggestions and will update the writing accordingly. Here we provide an intuitive explanation of MLMC algorithm. The idea is that for the maximum likelihood estimate $\hat{p}\_N$ using $N$ samples, when $N\rightarrow\infty$, the estimator is accurate: $\hat{p}\_\infty=p$. Thus, we have that $f(p)=f(\hat{p}\_\infty)$. It hence allows us to rewrite the function as $f(p)=f(\hat{p}\_\infty)=f(\hat{p}\_{2^0})+\sum\_{N=1}^\infty f(\hat{p}\_{2^N})- f(\hat{p}\_{2^{N-1}})=f(\hat{p}\_{2^0})+\sum\_{N=1}^\infty P\_N \frac{f(\hat{p}\_{2^{N+1}})- f(\hat{p}\_{2^{N}})}{P\_N}= f(\hat{p}\_{2^0})+\mathbb{E}\_{N\sim P\_N}[\frac{f(\hat{p}\_{2^{N+1}})- f(\hat{p}\_{2^N})}{P\_N}] $ for some distribution $P\_N$. It hence suffices to obtain an unbiased estimator of $\mathbb{E}\_{N\sim P\_N}[\frac{f(\hat{p}\_{2^{N+1}})- f(\hat{p}\_{2^N})}{P\_N}]$. Hence, we can straightforwardly sample $N\sim P\_N$ and construct the "correction term" $\delta^{r, \rho(\sigma)}\_{s,a,N}=\frac{f(\hat{p}\_{2^{N+1}})- f(\hat{p}\_{2^{N}})}{P\_{N}} $ to be an unbiased estimator of $\mathbb{E}\_{N\sim P\_N}[\frac{f(\hat{p}\_{2^{N+1}})- f(\hat{p}\_{2^N})}{P\_N}]$.
>
> **Weaknesses 1.c:**
>
> In algorithm design, the key modification compared to previous MLMC algorithms is the inclusion of a threshold, but it leads to substantial new development in the theoretical analysis. Previous MLMC algorithms necessitate infinite samples [Liu et al,2022] or rely on a strong assumption ($\frac{p_\wedge}{2}\geq 1-e^{-\sigma}$) [Wang et al, 2023ab] to ensure the MLMC is unbiased with bounded variance.
> Our threshold-based design circumvents these issues, albeit at the expense of introducing bias into the estimator.  We characterize the exponentially small probability of the difference between T-MLMC and MLMC occurs, and carefully analyze the deviation introduced by this bias. We showed that with a carefully designed threshold, the deviation is small and our biased algorithm converges to a close neighbour of the optimal policy, requiring fewer samples and no additional assumptions. This underscores the effectiveness of our design and its significant advantages.
>
> **Weaknesses 1.d:**
>
> We will add the discussion in Weaknesses 1. ab  and introductory paragraph in the revised version.
>
> **Weaknesses 2. a:**
>
> One major advantage of model-free method is that it can eliminate the need of explicitly estimating the empirical transition kernel and can be implemented in an online fashion, which saves a factor of $S$ in the memory cost.
> In many practical applications, the state space, though being discrete, can still be extremely large and such a reduction results in a great improvement in memory efficiency. There have been quite a few works delicate to find the optimal dependency of the complexity on $S$, as listed in [a,b] for online learning setting.
>
> [a] Li, Gen, et al. "Is Q-learning minimax optimal? a tight sample complexity analysis."
>
> [b] Zhang, et al. "Settling the sample complexity of online reinforcement learning."
>
> **Weaknesses 2. b and d:**
>
> Here we notice there are two different works [Panaganti \& Kalathil, 2022a] and [Panaganti \& Kalathil, 2022b]. In this paper, we cite the work [Panaganti \& Kalathil, 2022a] in tables, which is model-based for **tabular** robust MDPs. [Panaganti \& Kalathil, 2022b] provides studies for robust RL with **linear function approximation**. Since we focus on the tabular setting in this paper, and therefore, we did not discuss [Panaganti \& Kalathil, 2022b] in the table.
>
> We agree with the reviewer that additional techniques like function approximation or low-dimensional structure are required when tackling large-scale problems. In this paper, we start with the fundamental tabular setting, and aims to understand even in this fundamental setting, how to design the algorithm and further analyze their complexity. It is also of our future interest to extend our approach to large-scale problems using function approximation or low-dimensional structure.
>
> Panaganti, \& Kalathil, (2022a). Sample complexity of robust reinforcement learning with a generative model.
>
> Panaganti, et al. (2022b). Robust reinforcement learning using offline data.
>
> **Weaknesses 2.c:**
>
>
> We clarified in Section 4 that model-based approaches have greater complexity, which is acknowledged. Model-based methods are typically more sample-efficient than most model-free methods, except for a few model-free approaches that use variance reduction to match the complexity of model-based methods.
>
> However, the major benefits that model-free approaches offered is in terms of memory/space efficiency.  It makes the model-free approaches more appealing for scenarios where computational resources are limited or when working with large-scale problems.
>
> On the other hand, with variance reduction methods, our T-MLMC is expected to achieve the same sample complexity as model-based method. We leave it as a future interest.

---

### Meta-Review · Area_Chair_GbsX · 2024-04-21

The authors introduce multi-leve Monte Carlo (MLMC) for distributionally robust MDP, which leads to a novel model-free algorithm with competitive sample complexity, for TV-, chi-square, and KL- ambiguity sets.

All the reviewers acknolwdge the novelty in using MLMC for robust MDP. The major concern raised by the reviewers lies in the discussion to the related work, as the references provided by Reviewer 6UQA.

Please consider the suggestions to further improve the submission.

> Typo: In conclusion section, "we introduce a novel model-based T-MLMC algorithm tailored for finding the optimal robust policy in the DR-RL problem", I guess it should be "model-free".